# (Accelerated) Noise-adaptive
# Stochastic Heavy-Ball Momentum

**Anh Dang**                                                    *anh__dang@sfu.ca*
*Simon Fraser University*

**Reza Babanezhad**                                          *babanezhad@gmail.com*
*Samsung AI, Montreal*

**Sharan Vaswani**                                      *vaswani.sharan@gmail.com*
*Simon Fraser University*

**Reviewed on OpenReview:** *https://openreview.net/forum?id=0kxp1W8If0*

## Abstract

Stochastic heavy ball momentum (SHB) is commonly used to train machine learning models, and often provides empirical improvements over stochastic gradient descent. By primarily focusing on strongly-convex quadratics, we aim to better understand the theoretical advantage of SHB and subsequently improve the method. For strongly-convex quadratics, Kidambi et al. (2018) show that SHB (with a mini-batch of size 1) cannot attain accelerated convergence, and hence has no theoretical benefit over SGD. They conjecture that the practical gain of SHB is a by-product of using larger mini-batches. We first substantiate this claim by showing that SHB can attain an accelerated rate when the mini-batch size is larger than a threshold $b^*$ that depends on the condition number $\kappa$. Specifically, we prove that with the same step-size and momentum parameters as in the deterministic setting, SHB with a sufficiently large mini-batch size results in an $O\left(\exp(-T/\sqrt{\kappa}) + \sigma\right)$ convergence when measuring the distance to the optimal solution in the $\ell_2$ norm, where $T$ is the number of iterations and $\sigma^2$ is the variance in the stochastic gradients. We prove a lower-bound which demonstrates that a $\kappa$ dependence in $b^*$ is necessary. To ensure convergence to the minimizer, we design a noise-adaptive multi-stage algorithm that results in an $O\left(\exp\left(-T/\sqrt{\kappa}\right) + \frac{\sigma}{\sqrt{T}}\right)$ rate when measuring the distance to the optimal solution in the $\ell_2$ norm. We also consider the general smooth, strongly-convex setting and propose the first noise-adaptive SHB variant that converges to the minimizer at an $O(\exp(-T/\kappa) + \frac{\sigma^2}{T})$ rate when measuring the distance to the optimal solution in the squared $\ell_2$ norm. We empirically demonstrate the effectiveness of the proposed algorithms.

## 1 Introduction

Heavy ball (HB) or Polyak momentum (Polyak, 1964) has been extensively studied for minimizing smooth, strongly-convex quadratics in the deterministic setting. In this setting, HB converges to the minimizer at an accelerated linear rate (Polyak, 1964; Wang et al., 2021) meaning that for a problem with condition number $\kappa$ (see definition in Section 2), $T$ iterations of HB results in the optimal $O\left(\exp(-T/\sqrt{\kappa})\right)$ convergence. For general smooth, strongly-convex functions, Ghadimi et al. (2015) prove that HB converges to the minimizer at a linear but non-accelerated rate. In this setting, Wang et al. (2022) prove an accelerated linear rate for HB, but under very restrictive assumptions (e.g. one-dimensional problems or problems with a diagonal hessian). Recently, Goujaud et al. (2023) showed that HB (with any fixed step-size or momentum parameter) cannot achieve accelerated convergence on general (non-quadratic) strongly-convex problems, and consequently has no theoretical benefit over gradient descent (GD).

While there is a good theoretical understanding of HB in the deterministic setting, the current understanding of stochastic heavy ball momentum (SHB) is rather unsatisfactory. SHB is commonly used to train machine

learning models and often provides empirical improvements over stochastic gradient descent (SGD). Furthermore, it forms the basis of modern adaptive gradient methods such as Adam (Kingma & Ba, 2014). As such, it is important to better understand the theoretical advantage of SHB over SGD. Previous works (Defazio, 2020; You et al., 2019) have conjectured that the use of momentum for non-convex minimization can help reduce the variance resulting in faster convergence. Recently, Wang et al. (2023) analyze stochastic momentum in the regime where the gradient noise dominates, and demonstrate that in this regime, momentum has limited benefits with respect to both optimization and generalization. However, it is unclear whether momentum can provably help improve the convergence in other settings. In this paper, we primarily focus on the simple setting of minimizing strongly-convex quadratics, with the aim of better understanding the theoretical benefit of SHB and subsequently improving the method.

We first consider the general smooth, strongly-convex setting and aim to design an SHB variant that matches the theoretical convergence of SGD. In this setting, Sebbouh et al. (2020); Liu et al. (2020) use SHB with a constant step-size and momentum parameter, obtaining linear convergence to the neighborhood of the minimizer. In order to attain convergence to the solution, Sebbouh et al. (2020) use a sequence of constant-then-decreasing step-sizes to achieve an $O\left(\kappa^2/T^2 + \sigma^2/T\right)$ rate, where $\sigma^2$ is the variance in the stochastic gradients and the sub-optimality is measured in the squared $\ell_2$ norm. In contrast, in the same setting, SGD can attain an $O\left(\exp\left(-T/\kappa\right) + \sigma^2/T\right)$ convergence to the minimizer. To the best of our knowledge, in this setting, there is no variant of SHB that can converge to the minimizer at a rate matching SGD.

**Contribution 1: Noise-adaptive, non-accelerated convergence to the minimizer for smooth, strongly-convex functions.** In Section 3, we propose an SHB method that combines the averaging interpretation of SHB (Sebbouh et al., 2020) and the exponentially decreasing step-sizes (Li et al., 2021; Vaswani et al., 2022) to achieve an $O\left(\exp\left(-T/\kappa\right) + \sigma^2/T\right)$ convergence rate that matches the SGD rate. Importantly, the proposed algorithm is noise-adaptive meaning that it does not require the knowledge of $\sigma^2$, but recovers the non-accelerated linear convergence rate (matching Ghadimi et al. (2015)) when $\sigma = 0$. Moreover, the algorithm provides an adaptive way to set the momentum parameter, alleviating the need to tune this additional hyper-parameter.

Next, we focus on minimizing strongly-convex quadratics, and aim to analyze the conditions under which SHB is provably better than SGD. A number of works (Kidambi et al., 2018; Paquette & Paquette, 2021; Loizou & Richtárik, 2020; Bollapragada et al., 2023; Lee et al., 2022) have studied SHB for minimizing quadratics. In this setting, Kidambi et al. (2018) show that SHB (with batch-size 1 and any choice of step-size and momentum parameters) cannot attain an accelerated rate. They conjecture that the practical gain of SHB is a by-product of using larger mini-batches. Similarly, Paquette & Paquette (2021) demonstrate that SHB with small batch-sizes cannot obtain a faster rate than SGD. While Loizou & Richtárik (2020) prove an accelerated rate for SHB (for any batch-size) in the "$L1$ sense", this does not imply acceleration according to the standard sub-optimality metrics. Recently, Bollapragada et al. (2023); Lee et al. (2022) use results from random matrix theory to prove that SHB with a constant step-size and momentum can achieve an accelerated rate when the mini-batch size is sufficiently large. Compared to these works, we use the non-asymptotic analysis standard in the optimization literature, and prove stronger worst-case results.

**Contribution 2: Accelerated convergence to the neighborhood for quadratics.** Our result in Section 4.1 substantiates the claim by Kidambi et al. (2018). Specifically, for strongly-convex quadratics, we prove that SHB with a mini-batch size larger than a certain threshold $b^*$ (that depends on $\kappa$) and constant step-size and momentum parameters can achieve an $O\left(\exp(-T/\sqrt{\kappa}) + \sigma\right)$ non-asymptotic convergence up to a neighborhood of the solution where the sub-optimality is measured in the $\ell_2$ norm. For problems such as non-parametric regression (Belkin et al., 2019; Liang & Rakhlin, 2020) or feasible linear systems, where the interpolation property (Ma et al., 2018; Vaswani et al., 2019) is satisfied, $\sigma = 0$ and SHB with a large batch-size results in accelerated convergence to the minimizer.

**Contribution 3: Lower Bound for SHB.** Our result in Section 4.2 shows that there exist quadratics for which SHB (with a constant step-size and momentum) diverges when the mini-batch size is below a certain threshold. Moreover, the lower-bound demonstrates that a $\kappa$ dependence in $b^*$ is necessary.

The result in Section 4.1 only demonstrates convergence to the neighbourhood of the solution. Next, we aim to design an SHB algorithm that can achieve accelerated convergence to the minimizer.

**Contribution 4: Noise-adaptive, accelerated convergence to the minimizer for quadratics.** In Section 4.3, we design a multi-stage SHB method (Algorithm 1) and prove that for strongly-convex quadratics, Algorithm 1 (with a sufficiently large batch-size) converges to the minimizer at an accelerated $O\left(\exp\left(-T/\sqrt{\kappa}\right) + \frac{\sigma}{\sqrt{T}}\right)$ rate where the sub-optimality is measured in the $\ell_2$ norm. Algorithm 1 is noise-adaptive and has a similar structure as the algorithm proposed for incorporating Nesterov acceleration in the stochastic setting (Aybat et al., 2019). In comparison, both Bollapragada et al. (2023); Lee et al. (2022) only consider accelerated convergence to a neighbourhood of the minimizer. In concurrent work, Pan et al. (2024) make a stronger bounded variance assumption in order to analyze SHB for minimizing strongly-convex quadratics. They propose a similar multi-stage algorithm and under the bounded variance assumption, prove that it can converge to the minimizer at an accelerated rate for any mini-batch size. In Section 4.3, we argue that the bounded variance assumption is problematic even for simple quadratics and the algorithm in Pan et al. (2024) can diverge for small mini-batches (see Fig. 2).

In settings where $T \gg n$, the batch-size required by the multi-stage approach in Algorithm 1 can be quite large, affecting the practicality of the algorithm. In order to alleviate this issue, we design a two phase algorithm that combines the algorithmic ideas in Sections 3 and 4.1.

**Contribution 5: Partially accelerated convergence to the minimizer for quadratics.** In Section 4.4, we propose a two-phase algorithm (Algorithm 2) that uses a constant step-size and momentum in Phase 1, followed by an exponentially decreasing step-size and corresponding momentum in Phase 2. By adjusting the relative length of the two phases, we demonstrate that Algorithm 2 (with a sufficiently large batch-size) can obtain a partially accelerated rate.

**Contribution 6: Experimental Evaluation.** In Section 5, we empirically validate the effectiveness of the proposed algorithms on synthetic benchmarks. In particular, for strongly-convex quadratics, we demonstrate that SHB and its variants can attain an accelerated rate when the mini-batch size is larger than a threshold. While SHB with a constant step-size and momentum converges to a neighbourhood of the solution, Algorithms 1 and 2 are able to counteract the noise resulting in smaller sub-optimality.

## 2 Problem Formulation

We consider the unconstrained minimization of a finite-sum objective $f : \mathbb{R}^d \to \mathbb{R}$, $f(w) := \frac{1}{n} \sum_{i=1}^n f_i(w)$. For supervised learning, $n$ represents the number of training examples and $f_i$ is the loss of example $i$. Throughout, we assume that $f$ and each $f_i$ are differentiable and lower-bounded by $f^*$ and $f_i^*$, respectively. We also assume that each function $f_i$ is $L_i$-smooth, implying that $f$ is $L$-smooth with $L := \max_i L_i$. Furthermore, $f$ is considered to be $\mu$-strongly convex while each $f_i$ is convex[1]. We define $\kappa := \frac{L}{\mu}$ as the condition number of the problem, and denote $w^*$ to be the unique minimizer of the above problem. We primarily focus on strongly-convex quadratic objectives where $f_i(w) := \frac{1}{2} w^T A_i w - \langle d_i, w \rangle$ and $f(w) = \frac{1}{n} \sum_1^n f_i(w) = w^T A w - \langle d, w \rangle$, where $A_i$ are symmetric positive semi-definite matrices. Here, $L = \lambda_{\max}[A]$ and $\mu = \lambda_{\min}[A] > 0$, where $\lambda_{\max}$ and $\lambda_{\min}$ refer to the maximum and minimum eigenvalues.

In each iteration $k \in [T] := \{0, 1, .., T\}$, SHB samples a mini-batch $B_k$ ($b := |B_k|$) of examples and uses it to compute the stochastic gradient of the loss function. The mini-batch is formed by sampling without replacement. We denote $\nabla f_{ik}(w_k)$ to be the average stochastic gradient for the mini-batch $B_k$, meaning that $\nabla f_{ik}(w_k) := \frac{1}{b} \sum_{i \in B_k} \nabla f_i(w_k)$ and $\mathbb{E}[\nabla f_{ik}(w_k)|w_k] = \nabla f(w_k)$. At iteration $k$, SHB takes a descent step in the direction of $\nabla f_{ik}(w_k)$ together with a momentum term computed using the previous iterate. Specifically, the SHB update is given as:

$$w_{k+1} = w_k - \alpha_k \nabla f_{ik}(w_k) + \beta_k (w_k - w_{k-1}) \tag{1}$$

where $w_{k+1}$, $w_k$, and $w_{k-1}$ are the SHB iterates and $w_{-1} = w_0$; $\{\alpha_k\}_{k=0}^{T-1}$ and $\{\beta_k\}_{k=0}^{T-1}$ is the sequence of step-sizes and momentum parameters respectively. In the next section, we analyze the convergence of SHB for general smooth, strongly-convex functions.

---

[1]We include definitions of these properties in App. A.

## 3 Non-accelerated linear convergence for strongly-convex functions

We first consider the non-accelerated convergence of SHB in the general smooth, strongly-convex setting. Following Loizou et al. (2021); Vaswani et al. (2022), we define $\sigma^2 := \mathbb{E}_i[f^* - f_i^*]$ as the measure of stochasticity. We develop an SHB method that (i) converges to the minimizer at the $O\left(\exp\left(-T/\kappa\right) + \sigma^2/T\right)$ rate, (ii) is noise-adaptive in that it does not require the knowledge of $\sigma^2$ and (iii) does not require manual tuning the momentum parameter. In order to do so, we use an alternative form of the update (Sebbouh et al., 2020) that interprets SHB as a moving average of the iterates $z_k$ computed by stochastic gradient descent. Specifically, for $z_0 = w_0$,

$$w_{k+1} = \frac{\lambda_{k+1}}{\lambda_{k+1}+1}w_k + \frac{1}{\lambda_{k+1}+1}z_k \quad ; \quad z_k := z_{k-1} - \eta_k \nabla f_{i_k}(w_k), \tag{2}$$

where $\{\eta_k, \lambda_k\}$ are parameters to be determined theoretically. For any $\{\eta_k, \lambda_k\}$ sequence, if $\alpha_k = \frac{\eta_k}{1+\lambda_{k+1}}$, $\beta_k = \frac{\lambda_k}{1+\lambda_{k+1}}$, then the update in Eq. (2) is equivalent to the SHB update in Eq. (1) (Sebbouh et al., 2020, Theorem 2.1). The proposed SHB method combines the above averaging interpretation of SHB and exponentially decreasing step-sizes (Li et al., 2021; Vaswani et al., 2022) to achieve a noise-adaptive non-accelerated convergence rate. Specifically, following Li et al. (2021); Vaswani et al. (2022), we set $\eta_k = \upsilon \gamma_k$, where $\upsilon$ is the problem-dependent scaling term that captures the smoothness of the function and $\gamma_k$ is the problem-independent term that controls the decay of the step-size. By setting $\{\eta_k, \gamma_k\}$ appropriately, the following theorem (proved in App. B) shows that the proposed method converges to the minimizer at an $O\left(\exp\left(-T/\kappa\right) + \sigma^2/T\right)$ rate. In contrast, Sebbouh et al. (2020) use constant-then-decaying step-sizes to obtain a sub-optimal $O\left(\kappa^2/T^2 + \sigma^2/T\right)$ rate.

---

**Theorem 1.** For $L$-smooth, $\mu$ strongly-convex functions, SHB (Eq. (2)) with $\tau \geq 1$, $\upsilon = \frac{1}{4L}$, $\gamma = \left(\frac{\tau}{T}\right)^{1/T}$, $\gamma_k = \gamma^{k+1}$, $\eta_k = \upsilon \gamma_k$, $\eta = \eta_0$ and $\lambda_k := \frac{1-2\eta L}{\eta_k \mu}\left(1 - (1-\eta_k \mu)^k\right)$ converges as:

$$\mathbb{E}\left\|w_{T-1} - w^*\right\|^2 \leq C_4 \left\|w_0 - w^*\right\|^2 \exp\left(-\frac{T}{\kappa}\frac{\gamma}{4\ln(T/\tau)}\right) + C_4 C_5 \frac{\sigma^2}{T}$$

where $\zeta = \sqrt{\frac{n-b}{(n-1)b}}$ and $C_4, C_5$ are polynomial in $\kappa$ and poly-logarithmic in $T$.

---

This rate matches that of SGD with an exponentially decreasing step-size (Li et al., 2021; Vaswani et al., 2022). In the deterministic setting, when $b = n$, then by Lemma 7, $\zeta = 0$, and SHB matches the non-accelerated linear rate of GD and HB (Ghadimi et al., 2015). Non-parametric regression (Belkin et al., 2019; Liang & Rakhlin, 2020) or feasible linear systems (Loizou & Richtárik, 2020) satisfy the interpolation (Ma et al., 2018; Vaswani et al., 2019) property. For these problems, the model is able to completely interpolate the data meaning that the noise at the optimum vanishes and hence $\sigma = 0$. For this case, SHB matches the convergence rate of constant step-size SGD (Vaswani et al., 2019). Compared to Sebbouh et al. (2020), the rate in Theorem 1 has the same optimal $\tilde{O}(1/T)$ dependence on the variance term, but results in a worse dependence on the constants. On the other hand, our algorithm results in a better dependence on the bias term:

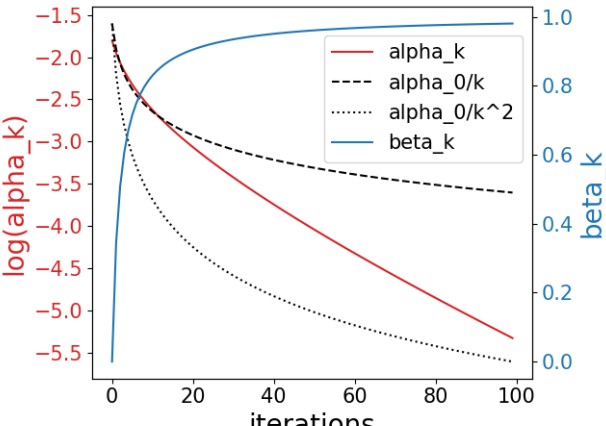

Figure 1: Variation in $\alpha_k := \frac{\eta_k}{1+\lambda_{k+1}}$ and $\beta_k := \frac{\lambda_k}{1+\lambda_{k+1}}$ where $\eta_k := \frac{(\tau/T)^{\frac{k+1}{T}}}{4L}$, $\lambda_k := \frac{1-2\eta_0 L}{\eta_k \mu}\left(1 - (1-\eta_k \mu)^k\right)$ for $T = 100$, $L = 10$, $\mu = 1$

$O(\exp(-T))$ in Theorem 1 versus the $O(1/T^2)$ rate obtained by Sebbouh et al. (2020). Consequently, when using the full dataset or under the interpolation setting, the rate in Theorem 1 recovers that of deterministic HB (Ghadimi et al., 2015), while that in Sebbouh et al. (2020) does not. For general strongly-convex

functions, Goujaud et al. (2023) prove that HB (with any step-size or momentum parameter) cannot achieve an accelerated convergence rate on general (non-quadratic and with dimension greater than 1) smooth, strongly-convex problems. Furthermore, we know that the variance term (depending on $\sigma^2$) cannot be decreased at a faster rate than $\Omega(1/T)$ (Nguyen et al., 2019). Hence, the above rate is the best-achievable for SHB in the general strongly-convex setting.

We reiterate that the method does not require knowledge of $\sigma^2$ and is hence noise-adaptive. Furthermore, all algorithm parameters are completely determined by the $\mu$, $L$ and $\gamma_k$ sequence. Hence, the resulting algorithm does not require manual tuning of the momentum. In Fig. 1, we show the variation of the $(\alpha_k, \beta_k)$ parameters, and observe that the method results in a more aggressive decrease in the step-size (compared to the standard $O(1/k)$ rate). This compensates for the increasing momentum parameter. The above theorem requires knowledge of $L$ and $\mu$ which can be difficult to obtain in practice. Hence in App. C, we consider the effect of misestimating $L$ and $\mu$ on the convergence rate of SHB. These are the first results that consider the effect of parameter misspecification for SHB.

Next, we focus on strongly-convex quadratics where SHB can obtain an accelerated convergence rate.

# 4   Accelerated linear convergence for strongly-convex quadratics

In this section, we focus on strongly-convex quadratics and in Section 4.1, we prove that SHB with a large batch-size attains accelerated linear convergence to a neighbourhood determined by the noise. In Section 4.2, we prove a corresponding lower-bound that demonstrates the necessity of a large batch-size to attain acceleration. The exponentially decreasing step-sizes in Section 3 are too conservative to obtain an accelerated rate to the minimizer. Consequently, in Section 4.3, we design a multi-stage SHB algorithm that achieves accelerated convergence to the minimizer. Finally, in Section 4.4, we design a two-phase SHB algorithm that has a simpler implementation, but can only attain partially accelerated rates.

## 4.1   Upper Bound for SHB

In the following theorem (proved in App. D), we show that for strongly-convex quadratics, SHB with a batch-size $b$ larger than a certain problem-dependent threshold $b^*$, constant step-size and momentum parameter converges to a neighbourhood of the solution at an accelerated linear rate. We note that the measure of suboptimality in this section is expressed as the norm, whereas in the previous section, it was represented as the squared norm. By Jensen's inequality, an upper-bound on $\mathbb{E}\|w_T - w^*\|^2$ implies an upper-bound on $\mathbb{E}\|w_T - w^*\|$.

**Theorem 2.** For $L$-smooth, $\mu$ strongly-convex quadratics, SHB (Eq. (1)) with $\alpha_k = \alpha = \frac{a}{L}$ for $a \leq 1$, $\beta_k = \beta = \left(1 - \frac{1}{2}\sqrt{\alpha\mu}\right)^2$, batch-size $b$ s.t. $b \geq b^* := n \max\left\{\frac{1}{1 + \frac{n-1}{C\kappa^2}}, \frac{1}{1 + \frac{(n-1)a}{3}}\right\}$ converges as:

$$\mathbb{E}\|w_T - w^*\| \leq \frac{6\sqrt{2}\sqrt{\kappa}}{\sqrt{a}} \exp\left(-\frac{\sqrt{a}\,T}{2\sqrt{\kappa}}\max\left\{\frac{3}{4}, 1 - 2\sqrt{\kappa}\sqrt{\zeta}\right\}\right)\|w_0 - w^*\| + \frac{12\sqrt{a}\chi}{\mu}\min\left\{1, \frac{\zeta}{\sqrt{a}}\right\}$$

where $\chi := \sqrt{\mathbb{E}\|\nabla f_i(w^*)\|^2}$, $\zeta = \sqrt{3\frac{n-b}{(n-1)b}}$ and $C := 3^5 2^6$.

The first term in the convergence rate represents the bias. Since $1 - 2\sqrt{\kappa}\sqrt{\zeta} > \frac{3}{4}$ when $b \geq b^*$, the initial sub-optimality $\|w_0 - w^*\|$ is forgotten at an accelerated linear rate proportional to $\exp(-T/\sqrt{\kappa})$. Moreover, since the bias term depends on $\zeta$, using a larger batch-size (above $b^*$) leads to a smaller $\zeta$ resulting in faster convergence. We note that $a$ is a constant independent of $T$ to avoid the dependence of $b^*$ on $T$. In the deterministic case, when $b = n$ and $\zeta = 0$, we recover the non-asymptotic accelerated convergence for HB (Wang et al., 2021). Similar to the deterministic case, the accelerated convergence requires a "warmup" number of iterations meaning that $T$ needs to be sufficiently large to ensure that $\exp\left(-\frac{T}{\sqrt{\kappa}}\frac{\sqrt{a}}{2}\max\left\{\frac{3}{4}, 1 - 2\sqrt{\kappa}\sqrt{\zeta}\right\}\right) \leq \frac{6\sqrt{2\kappa}}{\sqrt{a}}$. The second term represents the variance, and determines the size of the neighbourhood. The above theorem uses $\chi^2 = \mathbb{E}\|\nabla f_i(w^*)\|^2$ as the measure of stochasticity, where $\chi^2 \leq 2L\sigma^2$ because of the $L$-smoothness of the problem. Compared to constant step-size SGD that achieves

an $O(\exp(-T/\kappa) + \chi)$, SHB with a sufficiently large batch-size results in an accelerated $O(\exp(-T/\sqrt{\kappa}) + \chi)$ rate. We observe that if $\kappa$ is large, a larger batch-size is required to attain acceleration. Likewise, using a smaller step-size requires a proportionally larger batch-fraction to guarantee an accelerated rate. On the other hand, as $n$ increases, the relative batch-fraction (equal to $b/n$) required for acceleration is smaller. The proof of the above theorem relies on the non-asymptotic result for HB in the deterministic setting (Wang et al., 2021), coupled with an inductive argument over the iterations.

The above result substantiates the claim that the practical gain of SHB is a by-product of using larger mini-batches. In comparison to the above result, Loizou & Richtárik (2020) also prove an accelerated rate for SHB, but measure the sub-optimality in terms of $\|\mathbb{E}[w_T - w^*]\|$. This does not effectively model the problem's stochasticity and only shows convergence in expectation, a weaker form of convergence compared to our mean-square result. In contrast to Bollapragada et al. (2023, Theorem 3.1) which results in an $O\left(T \exp(-T/\sqrt{\kappa}) + \sigma \log(d)\right)$ rate where $d$ is the problem dimension, we obtain a faster convergence rate without an additional $T$ dependence in the bias term, nor an additional $\log(d)$ dependence in the variance term. In order to achieve an accelerated rate, our threshold $b^*$ scales as $O\left(\frac{1}{1/n + 1/\kappa^2}\right)$. When $n >> O(\kappa^2)$, our result implies that SHB with a nearly constant (independent of $n$) mini-batch size can attain accelerated convergence to a neighbourhood of the minimizer. In contrast, (Bollapragada et al., 2023, Theorem 4) require a batch-size of $\Omega(d\,\kappa^{3/2})$ to attain an accelerated rate in the worst-case. This condition is vacuous in the over-parameterized regime when $d > n$. Hence, compared to our result, Bollapragada et al. (2023) require a more stringent condition on the batch-size when $d > \sqrt{\kappa}$. On the other hand, Lee et al. (2022) provide an average-case analysis of SHB as $d, n \to \infty$, and prove an accelerated rate when $b \geq n\frac{\bar{\kappa}}{\sqrt{\bar{\kappa}}}$ where $\bar{\kappa}$ is the average condition number. In the worst-case (for example, when all data points are the same and $\bar{\kappa} = \kappa = 1$), Lee et al. (2022) require $b = n$ in order to attain an accelerated rate.

In the interpolation setting described in Section 3, the noise at the optimum vanishes and $\sigma = 0$ implying that $\chi = 0$. In this setting, we prove the following Corollary 1 in App. D. Hence, under interpolation, SHB with a sufficiently large batch-size results in accelerated convergence to the minimizer, matching the corresponding result for SGD with Nesterov acceleration (Vaswani et al., 2022, Theorem 6) and ASGD (Jain et al., 2018).

**Corollary 1.** For $L$-smooth, $\mu$ strongly-convex quadratics, under interpolation, SHB (Eq. (1)) with the same parameters as in Theorem 2 and batch-size $b$ s.t. $b \geq b^* := n\frac{1}{1 + \frac{n-1}{C\,\kappa^2}}$ (where $C$ is defined in Theorem 2) converges as:

$$\mathbb{E}\|w_T - w^*\| \leq \frac{6\sqrt{2}}{\sqrt{a}}\sqrt{\kappa}\exp\left(-\frac{T}{\sqrt{\kappa}}\frac{\sqrt{a}}{2}\max\left\{\frac{3}{4}, 1 - 2\sqrt{\kappa}\sqrt{\zeta}\right\}\right)\|w_0 - w^*\|$$

When the noise $\chi \neq 0$ but is assumed to be known, Corollary 4 (proved in App. D) shows that the step-size and momentum parameter of SHB can be adjusted to achieve an $\epsilon$ sub-optimality (for some desired $\epsilon > 0$) at an accelerated linear rate. In the above results, the batch-size threshold depends on $\kappa$. In the following section, we prove a lower-bound showing that a dependence on $\kappa$ is necessary.

## 4.2 Lower Bound for SHB

For SHB with the same step-size and momentum as Corollary 1, we show that there exists quadratics for which SHB with a batch-size lower than a certain threshold diverges.

**Theorem 3.** For a $\bar{L}$-smooth, $\bar{\mu}$ strongly-convex quadratic problem $f(w) := \frac{1}{n}\sum_{i=1}^{n}\frac{1}{2}w^T A_i w$ with $n$ samples and dimension $d = n = 100$ such that $w^* = 0$ and each $A_i$ is an $n$-by-$n$ matrix of all zeros except at the $(i,i)$ position, we run SHB (1) with $\alpha_k = \alpha = \frac{1}{L}$, $\beta_k = \beta = \left(1 - \frac{1}{2}\sqrt{\alpha\bar{\mu}}\right)^2$. If $b < \frac{1}{1+\frac{n-1}{e^{3.3\kappa^{0.6}}}}n$ and $\Delta_k := \begin{pmatrix} w_k \\ w_{k-1} \end{pmatrix}$, for a $c > 1$, after $6T$ iterations, we have that:

$$\mathbb{E}\left[\|\Delta_{6T}\|^2\right] > c^T \|\Delta_0\|^2 .$$

The above lower-bound demonstrates that the dependence on $\kappa$ is necessary in the threshold $b^*$ for the batch-size. We note that the designed problem with $n = d$ corresponds to a feasible linear system and therefore satisfies interpolation. Intuitively, Theorem 3 shows that in order to attain an accelerated rate for SHB, it is necessary to have a large batch-size to effectively control the error between the empirical Hessian $\frac{1}{b}\sum_{i\in B_k} A_i$ at iteration $k$ and the true Hessian. When the batch-size is not large enough, the aggressive updates for accelerated SHB increase this error resulting in divergence. Importantly, the above lower-bound also holds for the step-size and momentum parameters used in Bollapragada et al. (2023). We note that our lower-bound result still leaves open the possibility that there are other (less aggressive) choices of the step-size and momentum that can result in an (accelerated) convergence rate with a smaller batch-size. The proof of the above theorem in App. E takes advantage of symbolic mathematics programming (Meurer et al., 2017), and maybe of independent interest. In contrast to the above result, the lower bound in Kidambi et al. (2018) shows that there exist strongly-convex quadratics where SHB with a batch-size of 1 and any choice of step-size and momentum cannot result in an accelerated rate.

We have shown that for strongly-convex quadratics (not necessarily satisfying interpolation), SHB (with large batch-size) can result in accelerated convergence to the neighbourhood of the solution. Next, we design a multi-stage algorithm that ensures accelerated convergence to the minimizer.

---

**Algorithm 1:** Multi-stage SHB

**Input**: $T$ (iteration budget), $b$ (batch-size)
**Initialization**: $w_0$, $w_{-1} = w_0$, $k = 0$
$I = \left\lfloor \frac{1}{\ln(\sqrt{2})} \mathcal{W}\left(\frac{T\ln(\sqrt{2})}{384\sqrt{\kappa}}\right)\right\rfloor$ ($\mathcal{W}(.)$ is the Lambert W function[2])
$T_0 = \frac{T}{2}$
$\forall i \in [1, I]$, $T_i = \left\lceil \frac{4 \, 2^{i/2}\sqrt{\kappa}}{(2-\sqrt{2})}\left((i/2 + 5)\ln(2) + \ln(\sqrt{\kappa})\right)\right\rceil$
**for** $i = 0$; $i < I + 1$; $i = i + 1$ **do**
    Set $a_i = 2^{-i}$, $\alpha_i = \frac{a_i}{L}$, $\beta_i = \left(1 - \frac{1}{2}\sqrt{\alpha_i\mu}\right)^2$
    $x_0 = w_i$
    **for** $k = 0$; $k < T_i$; $k = k + 1$ **do**
        Sample batch $B_k$ and calculate $\nabla f_{ik}(x_k)$
        $x_{k+1} = x_k - \alpha_i\nabla f_{ik}(x_k) + \beta_i(x_k - x_{k-1})$
    **end**
    $w_{i+1} = x_{T_i}$
**end**
**return** $w_{I+1}$

**Algorithm 2:** Two-phase SHB

**Input**: $T$ (iteration budget), $b$ (batch-size), $c \in (0,1)$ (relative phase lengths)
**Initialization**: $w_0$, $w_{-1} = w_0$, $k = 0$
Set $T_0 = cT$
**for** $k = 0$; $k \le T_0$; $k = k + 1$ **do**
    Choose $a = 1$, set $\alpha, \beta$ according to Theorem 2
    Use Update 1
**end**
**for** $k = T_0 + 1$; $k \le T$; $k = k + 1$ **do**
    Set $\eta_k, \lambda_k$ according to Theorem 1
    Use Update 2
**end**
**return** $w_T$

---

### 4.3 Multi-stage SHB

In this section, we propose to use a multi-stage SHB algorithm (Algorithm 1) and analyze its convergence rate. The structure of our multi-stage algorithm is similar to Aybat et al. (2019) who studied Nesterov acceleration in the stochastic setting. For a fixed iteration budget $T$, Algorithm 1 allocates $T/2$ iterations to stage zero and divides the remaining $T/2$ iterations into $I$ stages. The length for each of these $I$ stages

---

[2]The Lambert W function is defined as: for $x, y \in \mathbb{R}$, $y = \mathcal{W}(x) \implies y\exp(y) = x$.

increases exponentially, while the step-size used in each stage decreases exponentially. This decrease in the step-size helps counter the variance and ensures convergence to the minimizer. Theorem 4 (proved in in App. F) shows that Algorithm 1 converges to the minimizer at an accelerated linear rate.

> **Theorem 4.** For $L$-smooth, $\mu$ strongly-convex quadratics with $\kappa > 1$, for $T \geq \bar{T} := \frac{3 \cdot 2^8 \sqrt{\kappa}}{\ln(2)} \max \left\{ 4\kappa, e^2 \right\}$,
>
> Algorithm 1 with $b \geq b^* := n \max \left\{ \frac{1}{1 + \frac{n-1}{C \kappa^2}}, \frac{1}{1 + \frac{(n-1) a_I}{3}} \right\}$ converges as:
>
> $$\mathbb{E} \left\| w_T - w^* \right\| \leq C_7 \exp \left( -\frac{T}{8\sqrt{\kappa}} \right) \left\| w_0 - w^* \right\| + C_8 \frac{\chi}{\sqrt{T}}$$
>
> where $C := 3^5 2^6$ and $C_7, C_8$ are polynomial in $\kappa$ and poly-logarithmic in $T$.

From Theorem 4, we see that Algorithm 1 achieves a convergence rate of $O \left( \exp \left( -\frac{T}{\sqrt{\kappa}} \right) + \frac{\chi}{\sqrt{T}} \right)$ to the minimizer. It is important to note that in comparison to Theorem 1, the sub-optimality above is in terms of $\mathbb{E} \left\| w_T - w^* \right\|$ (instead of $\mathbb{E} \left\| w_T - w^* \right\|^2$). Hence, the above rate is optimal for strongly-convex quadratics since the bias term decreases at an accelerated linear rate while the variance term goes down as $1/\sqrt{T}$. Unlike in Corollary 4, Algorithm 1 does not require the knowledge of $\chi$ and is hence noise-adaptive. When $\chi = 0$, Algorithm 1 matches the rate of SHB in Corollary 1.

In concurrent work, Pan et al. (2024) design a similar multi-stage SHB algorithm. However, the algorithm's analysis requires a bounded variance assumption which implies that for all $k \in [T]$, there exists a $\tilde{\sigma} < \infty$ such that $\mathbb{E} \left\| \nabla f(w_k) - \nabla f_{ik}(w_k) \right\|^2 \leq \tilde{\sigma}^2$. For strongly-convex quadratics, this assumption implies that the algorithm iterates lie in a compact set (Jain et al., 2018). Note that this assumption is much stronger than that in Theorem 4 which only requires that the variance at the optimum be bounded. With this bounded variance assumption, Pan et al. (2024) prove that their multi-stage SHB algorithm converges to the minimizer at an accelerated rate *without any condition on the mini-batch size*. This is in contrast with our result in Theorem 4 which requires the mini-batch size to be large enough. This discrepancy is because of the different assumptions on the noise. In Fig. 2a, we use the same feasible linear system as in Theorem 3 and demonstrate that with a batch-size 1, the algorithm in Pan et al. (2024) can diverge. This is because the iterates do not lie on a compact set and $\tilde{\sigma}$ can grow in an unbounded fashion for $O(T)$ iterations (see Fig. 2b), demonstrating that the bounded variance assumption is problematic even for simple examples.

With this assumption, Pan et al. (2024) prove that their multi-stage algorithm converges at a rate of $\tilde{O} \left( T\kappa \exp(-T/\sqrt{\kappa}) + \frac{d\tilde{\sigma}}{\sqrt{T}} \right)$ (for a similar definition of suboptimality as in Theorem 4). The above upper-bound implies that their algorithm can only achieve a sublinear rate even when solving feasible linear systems with a large batch-size (Jain et al., 2018). In comparison, Algorithm 1 with a large batch-size can achieve an accelerated linear rate when solving feasible linear systems. From a theoretical perspective, the $\tilde{O} \left( \kappa^{1/4} \exp(-T/\sqrt{\kappa}) + \frac{\chi}{\sqrt{T}} \right)$ bound in Theorem 4 is better in the bias term (by a factor of $T$) and hence requires fewer "warmup" iterations. It is also better in the variance term in that it does not incur a dimension dependence. While the bound established by Pan et al. (2024) holds for $T \geq \Omega(\sqrt{\kappa})$, our analysis requires $T \geq \Omega(\kappa\sqrt{\kappa})$ to achieve the convergence guarantee. This additional dependence on $\kappa$ is an artifact of our simplified proof that analyzes each stage independently. Specifically, we use the result from Wang et al. (2021) that introduces an additional $\sqrt{\kappa}$ "warm-up" iterations. These additional $\sqrt{\kappa}$ iterations in each stage introduce the additional $\kappa$ dependence. To conclude, compared to Pan et al. (2024), we achieve better convergence guarantees with a simpler analysis under more realistic assumptions for larger iterations.

Finally, we note that if the variance is guaranteed to be bounded for some problems, the proposed algorithms can exploit this additional assumption and achieve rates comparable to Pan et al. (2024) *without* a large batch-size requirement. Please refer to App. G for a detailed analysis.

In Theorem 4, we observe that the batch-size threshold $b^*$ depends on $a_I = 2^{-I} = O(1/T)$. In order to understand the implications of this requirement, consider the case when $T = \psi n$ (for some $\psi > 0$). In this case, $b^* = n \max \left\{ \frac{1}{1 + \frac{n-1}{C \kappa^2}}, \frac{1}{1 + \frac{1}{4\psi}} \right\}$. For practical problems, $n$ is of the order of millions compared to $T$ which

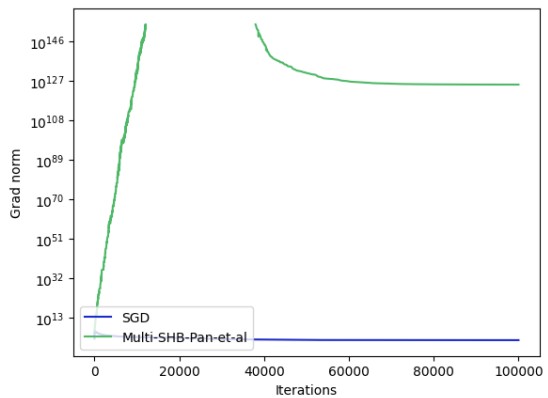 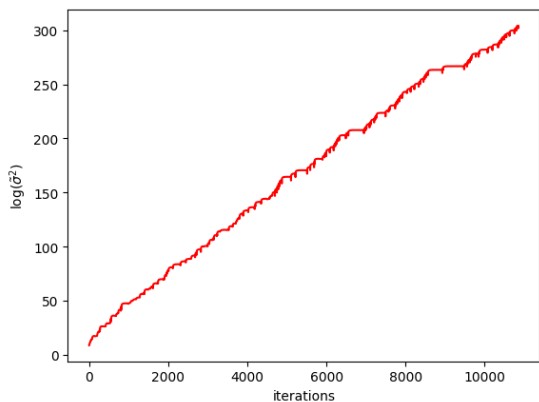

(a) Multi-stage SHB (Pan et al., 2024) with $b = 1$ diverges exponentially fast in the first few thousands iterations.

(b) The variance $\tilde{\sigma}^2$ is increasing.

Figure 2: Divergence of Multi-stage SHB (Pan et al., 2024) with $b = 1$ on the synthetic example in Theorem 3 with $\kappa = 5000$. We set $w_0 = \vec{100}$ and run the algorithm in (Pan et al., 2024) with $C = 2$. We consider 5 independent runs, and plot the average gradient norm $\|\nabla f(w_k)\|$ against the number of iterations. In Fig. 2b, we plot the (log) variance $\log\left(\mathbb{E}\|\nabla f(w_k) - \nabla f_{ik}(w_k)\|^2\right)$ against the number of iterations. We observe that multi-stage SHB diverges and the variance $\tilde{\sigma}^2$ increases, showing that the bounded variance assumption in Pan et al. (2024) is problematic.

is in the thousands and hence $\psi << 1$. Furthermore, when $n >> O(\kappa^2)$, $b^*$ is predominantly determined by the condition number.

An alternative way to reason about the above result is to consider a fixed batch-size $b$ as input. In this case, the following corollary presents the accelerated convergence of multi-stage SHB but only for a range of feasible $T$.

**Corollary 2.** For $L$-smooth, $\mu$ strongly-convex quadratics with $\kappa > 1$, Algorithm 1 with batch-size $b$ such that $b \geq b^* := n\frac{1}{1+\frac{n-1}{C\kappa^2}}$ attains the same rate as in Theorem 4 for $T \in \left[\frac{3 \cdot 2^8\sqrt{\kappa}}{\ln(2)}\max\left\{4\kappa, e^2\right\}, C_1\sqrt{\frac{(n-1)b}{3(n-b)}}\right]$, where $C := 3^5 2^6$ and $C_1$ is defined in the proof of Theorem 4 in App. F.

We have seen that a complicated algorithm can result in the optimal accelerated rate for a range of $T$. Next, we design a simple-to-implement algorithm that attains partially accelerated rates for all $T$.

### 4.4 Two-phase SHB

We design a two-phase SHB algorithm (Algorithm 2) that has a convergence guarantee for all $T$, but can only obtain a *partially accelerated rate* with a dependence on $\kappa^q$ for $q \in [\frac{1}{2}, 1]$. Here $q = \frac{1}{2}$ corresponds to the accelerated rate of Section 4.1, while $q = 1$ corresponds to the non-accelerated rate of Section 3. Algorithm 2 consists of two phases – in phase 1 consisting of $T_0$ iterations, it uses Eq. (1) with a constant step-size and momentum parameter (according to Theorem 2); in phase 2 consisting of $T_1 := T - T_0$ iterations, it uses Eq. (2) with an exponentially decreasing $\eta_k$ sequence and corresponding $\lambda_k$ (according to Theorem 1). The relative length of the two phases is governed by $c := T_0/T$. In App. H, we analyze the convergence of Algorithm 2 with general $c$ and prove Theorem 11. For a specific setting when $c = \frac{1}{2}$, we prove the following corollary.

**Corollary 3.** For $L$-smooth, $\mu$ strongly-convex quadratics with $\kappa > 4$, Algorithm 2 with batch-size $b$ such that $b \geq b^* = n\frac{1}{1+\frac{n-1}{C\kappa^2}}$ and $c = \frac{1}{2}$ results in a rate of $O\left(\exp\left(-\frac{T}{\kappa^{0.7}}\right) + \frac{\sigma}{\sqrt{T}}\right)$ for all $T$.

We observe that Algorithm 2, with a sub-optimal convergence rate of $O\left(\exp\left(-T/\kappa^{0.7}\right) + \sigma/\sqrt{T}\right)$, is faster than SGD and the non-accelerated SHB algorithm in Section 3. Compared to the accelerated SHB in Section 4.1,

the two-phase algorithm converges to the minimizer (instead of the neighbourhood). However, even in the interpolation setting when $\sigma = 0$, the two-phase algorithm (without any knowledge of $\sigma$) can only attain a partially accelerated rate.

## 5 Experimental Evaluation

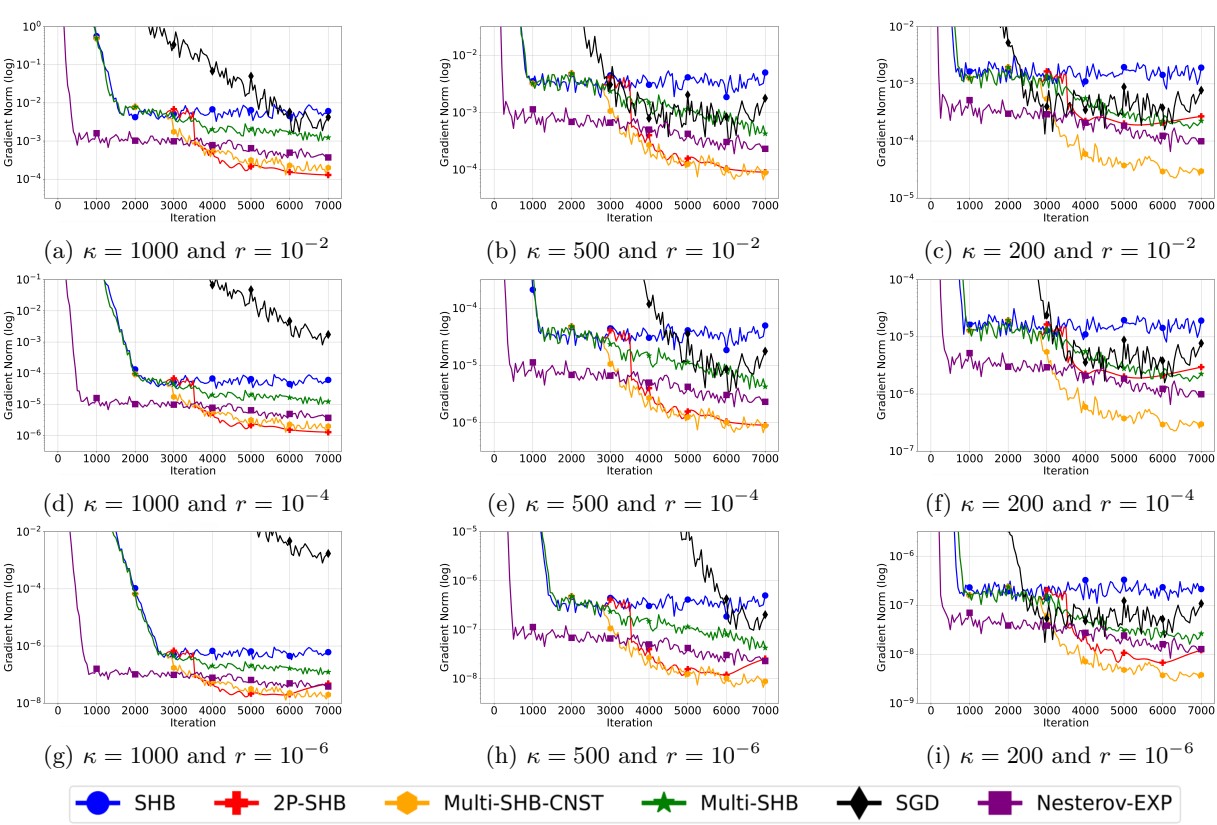

(a) $\kappa = 1000$ and $r = 10^{-2}$

(b) $\kappa = 500$ and $r = 10^{-2}$

(c) $\kappa = 200$ and $r = 10^{-2}$

(d) $\kappa = 1000$ and $r = 10^{-4}$

(e) $\kappa = 500$ and $r = 10^{-4}$

(f) $\kappa = 200$ and $r = 10^{-4}$

(g) $\kappa = 1000$ and $r = 10^{-6}$

(h) $\kappa = 500$ and $r = 10^{-6}$

(i) $\kappa = 200$ and $r = 10^{-6}$

Figure 3: Comparing `SHB`, `Multi-SHB`, `Multi-SHB-CNST`, `2P-SHB`, `SGD`, `Nesterov-EXP`, for the squared loss on synthetic datasets with different $\kappa$ and noise $r$. Both `SGD` and `SHB` converge to the neighborhood, but `SHB` attains an accelerated rate. `Multi-SHB`, `Multi-SHB-CNST` and `2P-SHB` result in smaller gradient norms and have similar convergence as `Nesterov-EXP`.

For our experimental evaluation[3], we consider minimizing strongly-convex quadratics. In particular, we generate random synthetic regression datasets with $n = 10000$ and $d = 20$. For this, we generate a random $w^*$ vector and a random feature matrix $X \in \mathbb{R}^{n \times d}$. We control the maximum and minimum eigenvalues of the resulting $X^T X$ matrix, thus controlling the $L$-smoothness and $\mu$-strong-convexity of the resulting quadratic problem. The measurements $y \in \mathbb{R}^n$ are generated according to the model: $y = Xw^* + s$ where $s \sim \mathcal{N}(0, rI_n)$ corresponds to Gaussian noise. We vary $\kappa \in \{1000, 500, 200\}$ and the magnitude of the noise $r \in \{10^{-2}, 10^{-4}, 10^{-6}\}$. These choices are motivated by Aybat et al. (2019). By controlling $r$, we can control the variance in the stochastic gradients. Using these synthetic datasets, we consider minimizing the unregularized linear regression loss: $f(w) = \frac{1}{2} \|Xw - y\|^2$. In this case, $A = X^T X$, $d = 2y^T X$ and $A_i = X_i^T X_i$, $d_i = 2y_i^T X_i$.

We compare the following methods: SHB with a constant step-size and momentum (set according to Theorem 2) with $a = 1$ (`SHB`), Multi-stage SHB (Algorithm 1) (`Multi-SHB`), Two-phase SHB (Algorithm 2) with $c = 0.5$ (`2P-SHB`), and use the following baselines – SGD (`SGD`), SGD with Nesterov acceleration and exponentially decreasing step-sizes (Vaswani et al., 2022) (`Nesterov-EXP`). Additionally, we consider a heuristic we refer to as Multi-stage SHB with constant momentum parameter (`Multi-SHB-CNST`). The heuristic has the same structure as Algorithm 1, but the momentum parameter in each stage is fixed i.e. $\beta_i = \left(1 - 1/2\sqrt{\kappa}\right)^2$. We will

---

[3]The code is available at https://github.com/anh-dang/accelerated_noise_adaptive_shb

see that this heuristic can result in better convergence than `Multi-SHB`. However, analyzing it theoretically is nontrivial. For each compared method, we use a mini-batch size $b = 0.9n$ to ensure that it is sufficiently large for SHB to achieve an accelerated rate for our choices of $\kappa$. We note that using $b = 0.9n$ on a noisy regression problem has enough stochasticity to meaningfully compare optimization methods. We fix the total number of iterations $T = 7000$ and initialization $w_0 = \mathbf{0}$. For each experiment, we consider 3 independent runs, and plot the average result. We will use the full gradient norm as the sub-optimality measure and plot it against the number of iterations.

From Fig. 3, we observe that: (i) both `SGD` and `SHB` converge to the neighborhood of the minimizer which depends on the noise $r$. However, `SHB` attains an accelerated rate, thus converging to the neighborhood faster. (ii) `Multi-SHB`, `Multi-SHB-CNST` and `2P-SHB` can better counteract the noise, and result in smaller gradient norm after reaching the neighborhood at an accelerated rate. (iii) The `Multi-SHB-CNST` heuristic results in slightly better empirical performance than `Multi-SHB` when $\kappa$ is relatively small. (iv) `2P-SHB` results in consistently better performance compared to `Multi-SHB`. (v) Across problems, the SHB variants have similar convergence as `Nesterov-EXP`.

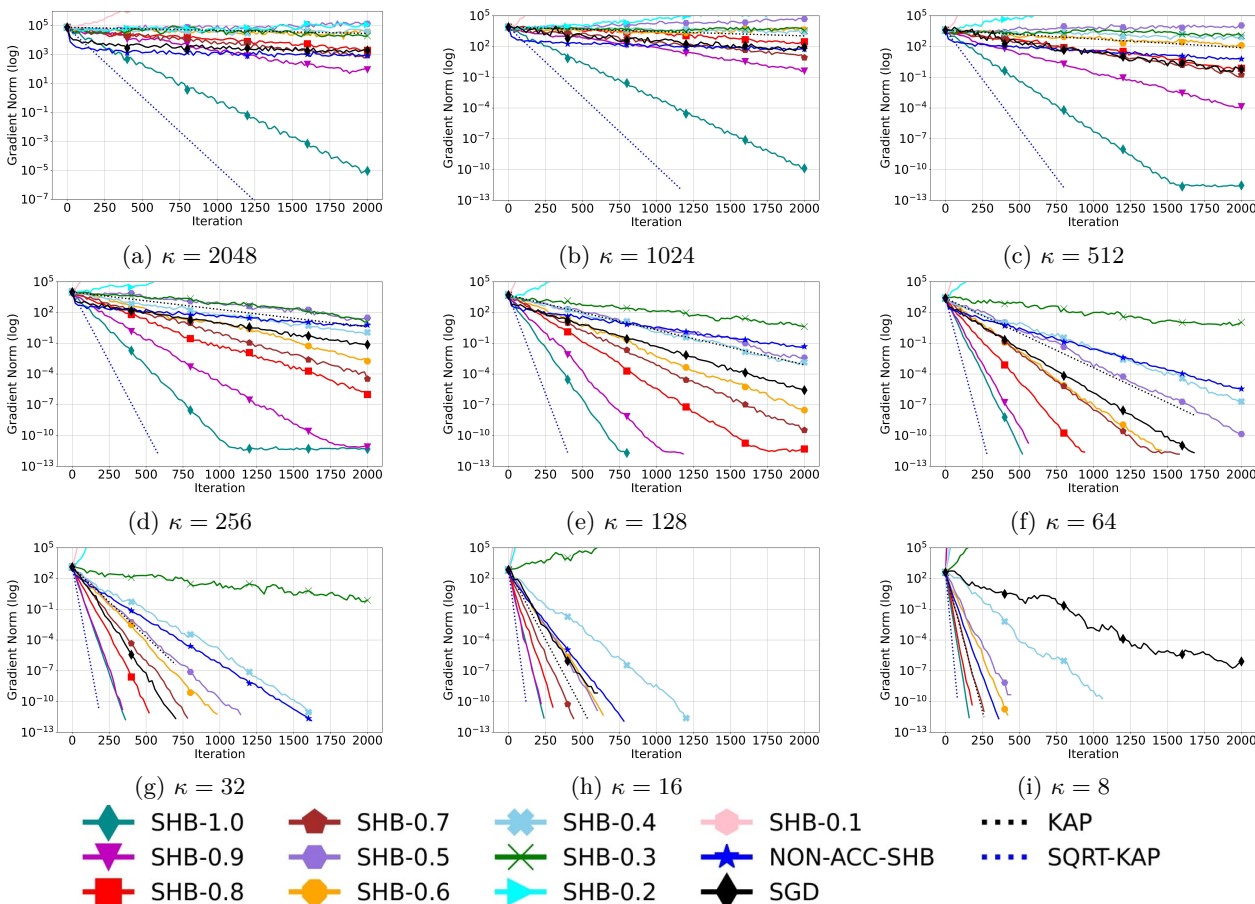

Figure 4: Comparison of `SHB-`$\xi$, `NON-ACC-SHB`, `SGD` and baselines `KAP`, `SQRT-KAP` for the squared loss on synthetic datasets with different $\kappa$. For large $\kappa$, SHB can converge in an accelerated rate if the batch-size is larger than the threshold $b^*$. The performances of `SGD` and `NON-ACC-SHB` are similar and significantly slower than SHB when $\kappa$ is large.

Next, we consider solving synthetic feasible linear systems with different values of $\kappa$, and examine the convergence of SHB with different batch-sizes. The data generation procedure is similar as above, however, there is no Gaussian noise ($s = 0$) and hence interpolation is satisfied. In particular, the measurements $y \in \mathbb{R}^n$ are now generated according to the model: $y = Xw^*$. We vary $\kappa \in \{8, 16, 32, 64, 128, 256, 512, 1024, 2048\}$ and batch-size $b = \xi n$ for $\xi \in \{0.1, 0.2, 0.3, 0.4, 0.5, 0.6, 0.7, 0.8, 0.9, 1.0\}$. We fix the total number of iterations $T = 2000$. For each experiment, we consider 5 independent runs, and plot the average result. We will use the

full gradient norm as the performance measure and plot it against the number of iterations. We compare the following methods: accelerated SHB with a constant step-size and momentum (set according to Theorem 2) with $a = 1$ and varying batch-size $\xi$ (SHB-$\xi$), non-accelerated SHB with a constant step-size and momentum (set according to Theorem 1) and a fixed batch-size $b = 0.3n$ (NON-ACC-SHB), SGD with a constant step-size and a fixed batch-size $b = 0.3n$ (SGD). We also add the following baselines to understand the dependence of $\kappa$: line proportional to $\exp\left(\frac{-T}{\kappa}\right)$ (KAP) and line proportional to $\exp\left(\frac{-T}{\sqrt{\kappa}}\right)$ (SQRT-KAP). The baselines are calculated by multiplying the initial gradient norm with the corresponding exponential term calculated at each iteration.

From Fig. 4, we observe that (i) when $\kappa$ is large, using SHB with smaller batch-sizes can result in divergence, (ii) SHB can only attain acceleration when the batch-size is larger than some $\kappa$-dependent threshold, and the extent of acceleration depends on the batch-size, (iii) across problems, the performance of SGD and NON-ACC-SHB is similar and slower than SHB when $\kappa$ is large, (iv) the larger batch-size, SHB converges at a rate similar to the SQRT-KAP baseline, (v) across problems, SGD converges at a rate similar to the KAP baseline. This verifies our theoretical results in Sections 4.1 and 4.2.

Finally, in App. I.2, we consider the algorithm proposed in Pan et al. (2024). We observe that with a sufficiently large batch-size, the method converges and has similar performance to the proposed SHB variants.

## 6    Conclusion

For the general smooth, strongly-convex setting, we developed a novel variant of SHB that uses exponentially decreasing step-sizes and achieves noise-adaptive non-accelerated linear convergence for *any mini-batch size*. This rate matches that of SGD and is the best achievable rate for SHB in this setting (given the negative results in Goujaud et al. (2023)). For strongly-convex quadratics, we demonstrated that SHB can achieve accelerated linear convergence if its mini-batch size is above a certain problem-dependent threshold. Our results imply that for strongly-convex quadratics where $n >> O(\kappa^2)$, SHB (and its multi-stage and two-phase variants) with a nearly constant (independent of $n$) mini-batch size can be provably better than SGD, thus quantifying the theoretical benefit of SHB. In the future, we aim to close the gap between the upper and lower-bounds on the mini-batch size required for SHB to attain an accelerated rate. Furthermore, we aim to improve our lower-bound and characterize the behaviour of SHB with any step-size and momentum parameter. On the more practical side, we hope to develop SHB variants that can attain an accelerated rate when the batch-size is large, and automatically default to non-accelerated rates for smaller batch-sizes.

## Acknowledgements

We would like to thank Saurabh Mishra and Duy Anh Nguyen for helpful discussions and feedback on the paper. This research was partially supported by the Natural Sciences and Engineering Research Council of Canada (NSERC) Discovery Grant RGPIN-2022-04816.

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

# Supplementary Material

## Organization of the Appendix

## A  Definitions

Our main assumptions are that each individual function $f_i$ is differentiable, has a finite minimum $f_i^*$, and is $L$-smooth, meaning that for all $v$ and $w$,

$$f_i(v) \leq f_i(w) + \langle \nabla f_i(w), v - w \rangle + \frac{L}{2} \|v - w\|^2, \qquad \text{(Individual Smoothness)}$$

which also implies that $f$ is $L$-smooth. A consequence of smoothness is the following bound on the norm of the stochastic gradients,

$$\|\nabla f_i(w)\|^2 \leq 2L(f_i(w) - f_i^*).$$

We also assume that each $f_i$ is convex, meaning that for all $v$ and $w$,

$$f_i(v) \geq f_i(w) - \langle \nabla f_i(w), w - v \rangle. \qquad \text{(Convexity)}$$

We will also assume that $f$ is $\mu$ strongly-convex, meaning that for all $v$ and $w$,

$$f(v) \geq f(w) + \langle \nabla f(w), v - w \rangle + \frac{\mu}{2} \|v - w\|^2. \qquad \text{(Strong Convexity)}$$

# B  Proofs for non-accelerated rates

We will require (Sebbouh et al., 2020, Theorem H.1). We include its proof for completeness.

**Theorem 5.** For $L$-smooth, $\mu$ strongly-convex functions, suppose $(\eta_k)_k$ is a decreasing sequence such that $\eta_0 = \eta$ and $0 < \eta_k < \frac{1}{2L}$. Define $\lambda_k := \frac{1-2\eta L}{\eta_k \mu}\left(1 - (1-\eta_k\mu)^k\right)$, $A_k := \|w_k - w^* + \lambda_k(w_k - w_{k-1})\|^2$, $\mathcal{E}_k := A_k + 2\eta_k\lambda_k(f(w_{k-1}) - f(w^*))$, $\alpha_k := \frac{\eta_k}{1+\lambda_{k+1}}$, $\beta_k := \lambda_k\frac{1-\eta_k\mu}{1+\lambda_{k+1}}$, $\sigma^2 := \mathbb{E}_i[f_i(w^*) - f_i^*] \geq 0$. Then SHB Eq. (1) converges as

$$\mathbb{E}[\mathcal{E}_{k+1}] \leq (1 - \eta_k\mu)\mathbb{E}[\mathcal{E}_k] + 2L\kappa\zeta^2\eta_k^2\sigma^2 \tag{3}$$

where $\zeta = \sqrt{\frac{n-b}{(n-1)b}}$.

*Proof.* We will first expand and bound the term $A_{k+1}$,

$$
\begin{aligned}
A_{k+1} &= \|w_{k+1} - w^* + \lambda_{k+1}(w_{k+1} - w_k)\|^2 \\
&= \|w_k - w^* - \alpha_k\nabla f_{ik}(w_k) + \beta_k(w_k - w_{k-1}) + \lambda_{k+1}\left[-\alpha_k\nabla f_{ik}(w_k) + \beta_k(w_k - w_{k-1})\right]\|^2 \\
&\hspace{10cm}\text{(SHB step)} \\
&= \|w_k - w^* - \alpha_k(1+\lambda_{k+1})\nabla f_{ik}(w_k) + \beta_k(1+\lambda_{k+1})(w_k - w_{k-1})\|^2 \\
&= \|w_k - w^* - \eta_k\nabla f_{ik}(w_k) + \lambda_k(1-\eta_k\mu)(w_k - w_{k-1})\|^2 \quad\text{(definition of $\alpha_k$ and $\beta_k$)} \\
&= \|w_k - w^* + \lambda_k(w_k - w_{k-1}) - \eta_k\left[\mu\lambda_k(w_k - w_{k-1}) + \nabla f_{ik}(w_k)\right]\|^2 \\
&= A_k + \eta_k^2\|\mu\lambda_k(w_k - w_{k-1}) + \nabla f_{ik}(w_k)\|^2 \\
&\quad - 2\eta_k\langle w_k - w^* + \lambda_k(w_k - w_{k-1}), \mu\lambda_k(w_k - w_{k-1}) + \nabla f_{ik}(w_k)\rangle \\
&= A_k + \eta_k^2\|\nabla f_{ik}(w_k)\|^2 + \underbrace{\eta_k^2\mu^2}_{\leq \eta_k\mu}\lambda_k^2\|w_k - w_{k-1}\|^2 \\
&\quad + 2\eta_k^2\mu\lambda_k\langle w_k - w_{k-1}, \nabla f_{ik}(w_k)\rangle - 2\eta_k\mu\lambda_k\langle w_k - w^*, w_k - w_{k-1}\rangle \\
&\quad - 2\eta_k\langle w_k - w^*, \nabla f_{ik}(w_k)\rangle - 2\eta_k\lambda_k\langle w_k - w_{k-1}, \nabla f_{ik}(w_k)\rangle - 2\eta_k\mu\lambda_k^2\|w_k - w_{k-1}\|^2 \\
&\leq A_k - \eta_k\mu\left(\lambda_k^2\|w_k - w_{k-1}\|^2 + 2\lambda_k\langle w_k - w^*, w_k - w_{k-1}\rangle\right) - 2\eta_k\langle w_k - w^*, \nabla f_{ik}(w_k)\rangle \\
&\quad + \underbrace{\eta_k^2\|\nabla f_{ik}(w_k)\|^2}_{\leq 2L\eta_k^2[f_{ik}(w_k) - f_{ik}^*]} + 2\eta_k^2\mu\lambda_k\langle w_k - w_{k-1}, \nabla f_{ik}(w_k)\rangle - 2\eta_k\lambda_k\langle w_k - w_{k-1}, \nabla f_{ik}(w_k)\rangle. \\
&\hspace{8cm}\text{(by $L$-smoothness of $f_{ik}$)}
\end{aligned}
$$

Add $B_{k+1} = 2\eta_{k+1}\lambda_{k+1}(f(w_k) - f^*)$ on both sides,

$$
\begin{aligned}
A_{k+1} + B_{k+1} &\leq A_k - \eta_k\mu\left(\lambda_k^2\|w_k - w_{k-1}\|^2 + 2\lambda_k\langle w_k - w^*, w_k - w_{k-1}\rangle\right) - 2\eta_k\langle w_k - w^*, \nabla f_{ik}(w_k)\rangle \\
&\quad + 2L\eta_k^2[f_{ik}(w_k) - f_{ik}^*] + 2\eta_k^2\mu\lambda_k\langle w_k - w_{k-1}, \nabla f_{ik}(w_k)\rangle \\
&\quad - 2\eta_k\lambda_k\langle w_k - w_{k-1}, \nabla f_{ik}(w_k)\rangle + 2\eta_{k+1}\lambda_{k+1}(f(w_k) - f^*) \\
&\leq A_k - \eta_k\mu\left(\lambda_k^2\|w_k - w_{k-1}\|^2 + 2\lambda_k\langle w_k - w^*, w_k - w_{k-1}\rangle\right) - 2\eta_k\langle w_k - w^*, \nabla f_{ik}(w_k)\rangle \\
&\quad + 2L\eta_k^2[f_{ik}(w_k) - f_{ik}^*] - 2\eta_k\lambda_k(1-\eta_k\mu)\langle w_k - w_{k-1}, \nabla f_{ik}(w_k)\rangle \\
&\quad + 2\eta_{k+1}\lambda_{k+1}[f(w_k) - f^*].
\end{aligned}
$$

Taking expectation w.r.t $i_k$, $f_{ik}(w_k) - f_{ik}^* = [f_{ik}(w_k) - f_{ik}(w^*)] + [f_{ik}(w^*) - f_{ik}^*]$ then

$$
\begin{aligned}
\mathbb{E}[A_{k+1} + B_{k+1}] &\leq \mathbb{E}[A_k] - \mathbb{E}\left[\eta_k\mu\left(\lambda_k^2\|w_k - w_{k-1}\|^2 + 2\lambda_k\langle w_k - w^*, w_k - w_{k-1}\rangle\right)\right] \\
&\quad - 2\eta_k\mathbb{E}[\langle w_k - w^*, \nabla f(w_k)\rangle] + 2L\kappa\zeta^2\eta_k^2\sigma^2 + 2L\eta_k^2\mathbb{E}[f(w_k) - f^*] \\
&\quad - 2\eta_k\lambda_k(1-\eta_k\mu)\mathbb{E}[\langle w_k - w_{k-1}, \nabla f(w_k)\rangle] + 2\eta_{k+1}\lambda_{k+1}\mathbb{E}[f(w_k) - f^*]. \\
&\hspace{10cm}\text{(Using Lemma 2)}
\end{aligned}
$$

Since $f$ is strongly-convex, $-2\eta_k\langle w_k - w^*, \nabla f(w_k)\rangle \leq -\eta_k\mu\|w_k - w^*\|^2 - 2\eta_k[f(w_k) - f^*]$, then

$$\mathbb{E}[\mathcal{E}_{k+1}] \leq \mathbb{E}[A_k] - \eta_k\mu\mathbb{E}[\underbrace{\|w_k - w^*\|^2 + \lambda_k^2\|w_k - w_{k-1}\|^2 + 2\lambda_k\langle w_k - w^*, w_k - w_{k-1}\rangle}_{A_k}]$$
$$+ 2L\kappa\zeta^2\eta_k^2\sigma^2 + 2L\eta_k^2\mathbb{E}[f(w_k) - f^*] - 2\eta_k\lambda_k(1 - \eta_k\mu)\mathbb{E}\left[\langle w_k - w_{k-1}, \nabla f(w_k)\rangle\right]$$
$$- 2\eta_k\mathbb{E}[f(w_k) - f^*] + 2\eta_{k+1}\lambda_{k+1}\mathbb{E}[f(w_k) - f^*]$$
$$\leq (1 - \eta_k\mu)\mathbb{E}[A_k] + 2L\kappa\zeta^2\eta_k^2\sigma^2 + 2L\eta_k^2\mathbb{E}[f(w_k) - f^*]$$
$$- 2\eta_k\lambda_k(1 - \eta_k\mu)\mathbb{E}\left[\langle w_k - w_{k-1}, \nabla f(w_k)\rangle\right]$$
$$- 2\eta_k\mathbb{E}[f(w_k) - f^*] + 2\eta_{k+1}\lambda_{k+1}\mathbb{E}[f(w_k) - f^*].$$

By convexity, $-\langle\nabla f(w_k), w_k - w_{k-1}\rangle \leq f(w_{k-1}) - f(w_k) = [f(w_{k-1}) - f^*] - [f(w_k) - f^*]$, then

$$\mathbb{E}[\mathcal{E}_{k+1}] \leq (1 - \eta_k\mu)\mathbb{E}[A_k] + 2L\kappa\zeta^2\eta_k^2\sigma^2 + \underbrace{2L\eta_k^2\mathbb{E}[f(w_k) - f^*]}_{\leq 4L\eta_k^2\mathbb{E}[f(w_k) - f^*]} + 2\eta_k\lambda_k(1 - \eta_k\mu)\mathbb{E}[f(w_{k-1}) - f^*]$$
$$- 2\eta_k\lambda_k(1 - \eta_k\mu)\mathbb{E}[f(w_k) - f^*] - 2\eta_k\mathbb{E}[f(w_k) - f^*] + 2\eta_{k+1}\lambda_{k+1}\mathbb{E}[f(w_k) - f^*]$$
$$\leq (1 - \eta_k\mu)\mathbb{E}[A_k + \underbrace{2\eta_k\lambda_k[f(w_{k-1}) - f^*]}_{B_k}] + 2L\kappa\zeta^2\eta_k^2\sigma^2 + 4L\eta_k^2\mathbb{E}[f(w_k) - f^*]$$
$$- 2\eta_k\lambda_k(1 - \eta_k\mu)\mathbb{E}[f(w_k) - f^*] - 2\eta_k\mathbb{E}[f(w_k) - f^*] + 2\eta_{k+1}\lambda_{k+1}\mathbb{E}[f(w_k) - f^*]$$
$$\leq (1 - \eta_k\mu)\mathbb{E}[\mathcal{E}_k] + 2L\kappa\zeta^2\eta_k^2\sigma^2$$
$$+ 2\mathbb{E}[f(w_k) - f^*]\left(2L\eta_k^2 - \eta_k\lambda_k(1 - \eta_k\mu) - \eta_k + \eta_{k+1}\lambda_{k+1}\right). \qquad \text{(Theorem 5 first part)}$$

We want to show that $2L\eta_k^2 - \eta_k\lambda_k(1 - \eta_k\mu) - \eta_k + \eta_{k+1}\lambda_{k+1} \leq 0$ which is equivalent to
$\eta_{k+1}\lambda_{k+1} \leq \eta_k\left(1 - 2L\eta_k + \lambda_k(1 - \eta_k\mu)\right)$.

$$\text{RHS} = \eta_k\left(1 - 2L\eta_k + \lambda_k(1 - \eta_k\mu)\right)$$
$$= \eta_k(1 - 2L\eta_k) + \eta_k\lambda_k(1 - \eta_k\mu)$$
$$= \eta_k(1 - 2L\eta_k) + \frac{1 - 2\eta L}{\mu}\left(1 - (1 - \eta_k\mu)^k\right)(1 - \eta_k\mu) \qquad \text{(definition of } \lambda_k)$$
$$= \eta_k(1 - 2L\eta_k) - \frac{1 - 2\eta L}{\mu}\eta_k\mu + \frac{1 - 2\eta L}{\mu}\left(1 - (1 - \eta_k\mu)^{k+1}\right)$$
$$= \frac{1 - 2\eta L}{\mu}\left(1 - (1 - \eta_k\mu)^{k+1}\right) + 2L\eta_k\underbrace{(\eta - \eta_k)}_{\geq 0} \qquad \text{(since } \eta \geq \eta_k)$$
$$\geq \frac{1 - 2\eta L}{\mu}\left(1 - (1 - \eta_k\mu)^{k+1}\right)$$
$$\geq \frac{1 - 2\eta L}{\mu}\left(1 - (1 - \eta_{k+1}\mu)^{k+1}\right) \qquad \text{(since } \eta_k \geq \eta_{k+1})$$
$$= \eta_{k+1}\lambda_{k+1} = \text{LHS}.$$

Hence,
$$\mathbb{E}[\mathcal{E}_{k+1}] \leq (1 - \eta_k\mu)\mathbb{E}[\mathcal{E}_k] + 2L\kappa\zeta^2\eta_k^2\sigma^2$$

$\square$

**Theorem 1.** For $L$-smooth, $\mu$ strongly-convex functions, SHB (Eq. (2)) with $\tau \geq 1$, $\upsilon = \frac{1}{4L}$, $\gamma = \left(\frac{\tau}{T}\right)^{1/T}$, $\gamma_k = \gamma^{k+1}$, $\eta_k = \upsilon\,\gamma_k$, $\eta = \eta_0$ and $\lambda_k := \frac{1-2\eta L}{\eta_k \mu}\left(1 - (1-\eta_k\mu)^k\right)$ converges as:

$$\mathbb{E}\left\|w_{T-1} - w^*\right\|^2 \leq C_4 \left\|w_0 - w^*\right\|^2 \exp\left(-\frac{T}{\kappa}\frac{\gamma}{4\ln(T/\tau)}\right) + C_4\,C_5\,\frac{\sigma^2}{T}$$

where $\zeta = \sqrt{\frac{n-b}{(n-1)\,b}}$ and $C_4, C_5$ are polynomial in $\kappa$ and poly-logarithmic in $T$.

*Proof.* From the result of Theorem 5 we have

$$\mathbb{E}[\mathcal{E}_k] \leq (1 - \eta_k\mu)\mathbb{E}[\mathcal{E}_{k-1}] + 2L\kappa\zeta^2\eta_k^2\sigma^2$$

Unrolling the recursion starting from $w_0$ and using the exponential step-sizes $\gamma_k$

$$\mathbb{E}[\mathcal{E}_T] \leq \mathbb{E}[\mathcal{E}_0] \prod_{k=1}^{T}\left(1 - \frac{\mu\gamma^k}{4L}\right) + 2L\kappa\zeta^2\sigma^2 \sum_{k=1}^{T}\left[\prod_{i=k+1}^{T} \gamma^{2k}\left(1 - \frac{\mu\gamma^i}{4L}\right)\right]$$

$$\leq \left\|w_0 - w^*\right\|^2 \exp\left(\underbrace{\frac{-\mu}{4L}\sum_{k=1}^{T}\gamma^k}_{:=C}\right) + 2L\kappa\zeta^2\sigma^2 \underbrace{\sum_{k=1}^{T}\gamma^{2k}\exp\left(-\frac{\mu}{4L}\sum_{i=k+1}^{T}\gamma^i\right)}_{:=D}$$

$$(\lambda_0 = 0 \text{ and } 1 - x < \exp(-x))$$

Using Lemma 3 to lower-bound $C$ then the first term can be bounded as

$$\left\|w_0 - w^*\right\|^2 \exp\left(\frac{-\mu}{4L}C\right) \leq \left\|w_0 - w^*\right\|^2 c_2 \exp\left(-\frac{T}{4\kappa}\frac{\gamma}{\ln(T/\tau)}\right)$$

where $\kappa = \frac{L}{\mu}$ and $c_2 = \exp\left(\frac{1}{2\kappa}\frac{2\tau}{\ln(T/\tau)}\right)$. Using Lemma 4 to upper-bound $D$, we have $D \leq \frac{32\kappa^2 c_2(\ln(T/\tau))^2}{e^2\gamma^2 T}$ then the second term can be bounded as

$$2L\kappa\zeta^2\sigma^2 D \leq \frac{64L\sigma^2 c_2\zeta^2\kappa^3}{e^2}\frac{(\ln(T/\tau))^2}{\gamma^2 T}$$

Hence

$$\mathbb{E}[\mathcal{E}_T] \leq \left\|w_0 - w^*\right\|^2 c_2 \exp\left(-\frac{T}{4\kappa}\frac{\gamma}{\ln(T/\tau)}\right) + \frac{64L\sigma^2 c_2\zeta^2\kappa^3}{e^2}\frac{(\ln(T/\tau))^2}{\gamma^2 T}$$

By Lemma 1, then

$$\mathbb{E}\left\|w_{T-1} - w^*\right\|^2 \leq \frac{c_2}{c_L}\left\|w_0 - w^*\right\|^2 \exp\left(-\frac{T}{\kappa}\frac{\gamma}{4\ln(T/\tau)}\right) + \frac{\sigma^2}{T}\frac{64L\zeta^2\kappa^3(\ln(T/\tau))^2}{e^2\gamma^2}\frac{c_2}{c_L}$$

Let $C_4(\kappa, T) := \frac{\exp\left(\frac{1}{2\kappa}\frac{2\tau}{\ln(T/\tau)}\right)}{\frac{4(1-\gamma)}{\mu^2}\left[1-\exp\left(-\frac{\gamma}{2\kappa}\right)\right]}$ and $C_5(\kappa, T) := \frac{64L\zeta^2\kappa^3(\ln(T/\tau))^2}{e^2\gamma^2}$, then

$$\mathbb{E}\left\|w_{T-1} - w^*\right\|^2 \leq C_4 \left\|w_0 - w^*\right\|^2 \exp\left(-\frac{T}{\kappa}\frac{\gamma}{4\ln(T/\tau)}\right) + \frac{\sigma^2}{T}C_4\,C_5$$

$$\square$$

## B.1 Helper Lemmas

**Lemma 1.** *For $\mathcal{E}_T := \|w_T - w^* + \lambda_T(w_T - w_{T-1})\|^2 + 2\eta_T \lambda_T(f(w_{t-1}) - f(w^*))$, $\mathcal{E}_T \geq c_L \|w_{T-1} - w^*\|^2$ where $c_L = \frac{4(1-\gamma)}{\mu^2}\left[1 - \exp\left(-\frac{\mu\gamma}{2L}\right)\right]$*

*Proof.*

$$\mathbb{E}[\mathcal{E}_T] = \mathbb{E}[A_T] + \mathbb{E}[B_T] \geq \mathbb{E}[B_T] = 2\lambda_T \eta_T \mathbb{E}[f(w_{T-1}) - f^*]$$

Hence, we want to lower-bound $\lambda_T \eta_T$ and we do this next

$$
\begin{aligned}
\lambda_T \eta_T &= \frac{1-\gamma}{\mu}\left[1 - \left(1 - \frac{\mu\gamma^{T+1}}{2L}\right)^T\right] && \text{(Using the definition of } \eta_k \text{ and } \lambda_k\text{)} \\
&\geq \frac{1-\gamma}{\mu}\left[1 - \exp\left(-T\gamma^T\frac{\mu\gamma}{2L}\right)\right] && \text{(Since } 1 - x \leq \exp(-x)\text{)} \\
&= \frac{1-\gamma}{\mu}\left[1 - \exp\left(-\frac{\mu\gamma}{2L}\right)\right] && \text{(Since } \gamma = \left(\frac{1}{T}\right)^{1/T}\text{)}
\end{aligned}
$$

Putting everything together, and using strong-convexity of $f$

$$\mathbb{E}[\mathcal{E}_T] \geq \underbrace{\frac{4(1-\gamma)}{\mu^2}\left[1 - \exp\left(-\frac{\mu\gamma}{2L}\right)\right]}_{:=c_L} \mathbb{E}\|w_{T-1} - w^*\|^2$$

$\square$

We restate (Vaswani et al., 2022, Lemma 2, Lemma 5, and Lemma 6) that we used in our proof.

**Lemma 2.** *If*

$$\sigma^2 := \mathbb{E}[f_i(w^*) - f_i^*],$$

*and each function $f_i$ is $\mu$ strongly-convex and $L$-smooth, then*

$$\sigma_\mathcal{B}^2 := \mathbb{E}_\mathcal{B}[f_\mathcal{B}(w^*) - f_\mathcal{B}^*] \leq \kappa \underbrace{\frac{n-b}{(n-1)b}}_{:=\zeta^2}\sigma^2.$$

**Lemma 3.** *For $\gamma = \left(\frac{\tau}{T}\right)^{1/T}$,*

$$A := \sum_{t=1}^{T}\gamma^t \geq \frac{\gamma T}{\ln(T/\tau)} - \frac{2\tau}{\ln(T/\tau)}$$

**Lemma 4.** *For $\gamma = \left(\frac{\tau}{T}\right)^{1/T}$ and any $\kappa > 0$, with $c_2 = \exp\left(\frac{1}{\kappa}\frac{2\tau}{\ln(T/\tau)}\right)$,*

$$\sum_{k=1}^{T}\gamma^{2k}\exp\left(-\frac{1}{\kappa}\sum_{i=k+1}^{T}\gamma^i\right) \leq \frac{4\kappa^2 c_2(\ln(T/\tau))^2}{e^2\gamma^2 T}$$

# C    Proofs for non-accelerated rates with misestimation

A practical advantage of using Eq. (2) with exponential step-sizes is its robustness to misspecification of $L$ and $\mu$. Specifically, in App. C.1, we analyze the convergence of SHB (Eq. (2)) when using an estimate $\hat{L}$ (rather than the true smoothness constant). In App. C.2, we analyze the convergence of SHB when using an estimate $\hat{\mu}$ for the strong-convexity parameter.

## C.1    $L$ misestimation

Without loss of generality, we assume that the estimate $\hat{L}$ is off by a multiplicative factor $\nu$ i.e. $\hat{L} = \frac{L}{\nu_L}$ for some $\nu_L > 0$. We note that $\hat{L}$ is a deterministic estimate of $L$. Here $\nu_L$ quantifies the estimation error with $\nu_L = 1$ corresponding to an exact estimation of $L$. In practice, it is typically possible to obtain lower-bounds on the smoothness constant. Hence, the $\nu_L > 1$ regime is of practical interest.

Similar to the dependence of SGD on smoothness mis-estimation obtained by Vaswani et al. (2022), Theorem 6 shows that with any mis-estimation on $L$ we can still recover the convergence rate of $O\left(\exp\left(\frac{-T}{\kappa}\right) + \frac{\sigma^2}{T}\right)$ to the minimizer $w^*$. Specifically, Theorem 6 demonstrates a convergence rate of $O\left(\exp\left(-\frac{\min\{\nu_L,1\}T}{\kappa}\right) + \frac{\max\{\nu_l^2,1\}(\sigma^2 + \Delta_f \max\{\ln(\nu_L),0\})}{T}\right)$. The first two terms in Theorem 6 are similar to those in Theorem 1. For $\nu_L \leq 1$, the third term is zero and the rate matches that in Theorem 1 upto a constant that depends on $\nu_L$. For $\nu_L > 1$, SHB initially diverges for $k_0$ iterations, but the exponential step-size decay ensures that the algorithm eventually converges to the minimizer. The initial divergence and the resulting slowdown in the rate is proportional to $\nu_L$. Finally, we note that Vaswani et al. (2022) demonstrate similar robustness for SGD with exponential step-sizes, while also proving the necessity of the slowdown in the convergence.

**Theorem 6.** Under the same settings as Theorem 1, SHB (Eq. (2)) with the estimated $\hat{L} = \frac{L}{\nu_L}$ results in the following convergence,

$$\mathbb{E}\|w_{T-1} - w^*\|^2$$
$$\leq \|w_0 - w^*\|^2 \frac{c_2}{c_L} \exp\left(-\frac{\min\{\nu_L, 1\}T}{2\kappa} \frac{\gamma}{\ln(T/\tau)}\right)$$
$$+ \frac{c_2}{c_L} \frac{32 L \kappa^3 \zeta^2 \ln(T/\tau)}{e^2 \gamma^2 T} \times \left[\left(\max\left\{1, \frac{\nu_L^2}{4L}\right\} \ln(T/\tau)\sigma^2\right)\right.$$
$$\left. + \left(\max\{0, \ln(\nu_L)\}\left(\sigma^2 + 2\Delta_f \frac{\nu_L - 1}{\nu_L \kappa}\right)\right)\right]$$

where $c_2 = \exp\left(\frac{1}{2\kappa} \frac{2\tau}{ln(T/\tau)}\right)$, $k_0 = T\frac{\ln(\nu_L)}{\ln(T/\tau)}$, and $\Delta_f = \max_{i \in [k_0]} \mathbb{E}[f(w_i) - f^*]$ and $c_L = \frac{4(1-\gamma)}{\mu^2}\left[1 - \exp\left(-\frac{\mu\gamma}{2L}\right)\right]$

*Proof.* Suppose we estimate $L$ to be $\hat{L}$. Now redefine

$$\eta_k = \frac{1}{2\hat{L}}\gamma_k$$
$$\hat{\lambda}_k = \frac{1 - 2\eta\hat{L}}{\eta_k\mu}\left(1 - (1 - \eta_k\mu)^k\right)$$
$$\hat{A}_k = \left\|w_k - w^* + \hat{\lambda}_k(w_k - w_{k-1})\right\|^2$$
$$\hat{B}_k = 2\eta_k\hat{\lambda}_k(f(w_{k-1}) - f(w^*))$$
$$\hat{\mathcal{E}}_k = \hat{A}_k + \hat{B}_k$$

Follow the proof of Theorem 5 until Theorem 5 first part step with the new definition,

$$\mathbb{E}[\hat{\mathcal{E}}_{k+1}] \leq (1 - \eta_k\mu)\mathbb{E}[\hat{\mathcal{E}}_k] + 2L\kappa\zeta^2\eta_k^2\sigma^2 + 2\mathbb{E}[f(w_k) - f^*]\underbrace{\left(2L\eta_k^2 - \eta_k\hat{\lambda}_k(1 - \eta_k\mu) - \eta_k + \eta_{k+1}\hat{\lambda}_{k+1}\right)}_{G} \quad (4)$$

$G$ can be bound as

$$
\begin{aligned}
G &= 2L\eta_k^2 - \eta_k\hat{\lambda}_k(1 - \eta_k\mu) - \eta_k + \eta_{k+1}\hat{\lambda}_{k+1} \\
&= \eta_k(2L\eta_k - 1) - \eta_k\hat{\lambda}_k(1 - \eta_k\mu) + \eta_{k+1}\hat{\lambda}_{k+1} \\
&= \eta_k(2L\eta_k - 1) + \eta_k(1 - 2\hat{L}\eta) - \frac{1 - 2\eta\hat{L}}{\mu}\left(1 - (1 - \eta_k\mu)^{k+1}\right) + \eta_{k+1}\hat{\lambda}_{k+1} && \text{(definition of } \hat{\lambda}_k) \\
&\leq 2\eta_k(L\eta_k - \hat{L}\eta) - \frac{1 - 2\eta\hat{L}}{\mu}\left(1 - (1 - \eta_{k+1}\mu)^{k+1}\right) + \eta_{k+1}\hat{\lambda}_{k+1} && (\eta_{k+1} \leq \eta_k) \\
&= 2\eta_k(L\eta_k - \hat{L}\eta) - \eta_{k+1}\hat{\lambda}_{k+1} + \eta_{k+1}\hat{\lambda}_{k+1} \\
&= 2\eta_k(L\eta_k - \hat{L}\eta)
\end{aligned}
$$

Hence Eq. (4) can be written as

$$\mathbb{E}[\hat{\mathcal{E}}_{k+1}] \leq (1 - \eta_k\mu)\mathbb{E}[\hat{\mathcal{E}}_k] + 2L\kappa\zeta^2\eta_k^2\sigma^2 + 4\mathbb{E}[f(w_k) - f^*]\eta_k(L\eta_k - \hat{L}\eta)$$

First case if $\nu_L \leq 1$ then $L\eta_k - \hat{L}\eta \leq 0$ and we will recover the proof of Theorem 1 with a slight difference including $\nu_L$.

$$\mathbb{E}[\hat{\mathcal{E}}_k] \leq \|w_0 - w^*\|^2 c_2 \exp\left(-\frac{\nu_L T}{2\kappa}\frac{\gamma}{\ln(T/\tau)}\right) + \frac{32L\kappa\zeta^2\sigma^2 c_2\kappa^2}{e^2}\frac{(\ln(T/\tau))^2}{\gamma^2 T}$$

Second case if $\nu_L > 1$

Let $k_0 = T\frac{\ln(\nu_L)}{\ln(T/\tau)}$ then for $k < k_0$ regime, $L\eta_k - \hat{L}\eta > 0$

$$\mathbb{E}[\hat{\mathcal{E}}_{k+1}] \leq (1 - \eta_k\mu)\mathbb{E}[\hat{\mathcal{E}}_k] + 2L\kappa\zeta^2\eta_k^2\sigma^2 + 4\mathbb{E}[f(w_k) - f^*]\eta_k(L\eta_k - \hat{L}\eta)$$

Let $\Delta_f = \max_{i\in[k_0]}\mathbb{E}[f(w_i) - f^*]$ and observe that $L\eta_k - \hat{L}\eta \leq L\eta_k\frac{\nu_L - 1}{\nu_L}$ then

$$
\begin{aligned}
\mathbb{E}[\hat{\mathcal{E}}_{k+1}] &\leq (1 - \eta_k\mu)\mathbb{E}[\hat{\mathcal{E}}_k] + 2L\kappa\zeta^2\eta_k^2\sigma^2 + 4L\eta_k^2\Delta_f\frac{\nu_L - 1}{\nu_L} \\
&= (1 - \frac{\mu\nu_L}{2L}\gamma^k)\mathbb{E}[\hat{\mathcal{E}}_k] + \underbrace{2L(\kappa\zeta^2\sigma^2 + 2\Delta_f\frac{\nu_L - 1}{\nu_L})}_{c_5}\eta_k^2
\end{aligned}
$$

Since $\nu_L > 1$

$$\mathbb{E}[\hat{\mathcal{E}}_{k+1}] \leq (1 - \frac{\mu}{2L}\gamma^k)\mathbb{E}[\hat{\mathcal{E}}_{k+1}] + c_5\eta_k^2$$

Unrolling the recursion for the first $k_0$ iterations we get

$$\mathbb{E}[\hat{\mathcal{E}}_{k_0}] \leq \mathbb{E}[\hat{\mathcal{E}}_0]\prod_{k=1}^{k_0-1}\left(1 - \frac{\mu}{2L}\gamma^k\right) + c_5\sum_{k=1}^{k_0-1}\gamma_k^2\prod_{i=k+1}^{k_0-1}\left(1 - \frac{\mu}{2L}\gamma_i\right)$$

Bounding the first term using Lemma 3,

$$\prod_{k=1}^{k_0-1}\left(1 - \frac{\mu}{2L}\gamma^k\right) \leq \exp\left(-\frac{\mu}{2L}\frac{\gamma - \gamma^{k_0}}{1 - \gamma}\right)$$

Bounding the second term using Lemma 4 similar to (Vaswani et al., 2022, Section C3)

$$\sum_{k=1}^{k_0-1} \gamma_k^2 \prod_{i=k+1}^{k_0-1} \left(1 - \frac{\mu}{2L}\gamma_i\right) \leq \exp\left(\frac{\gamma^{k_0}}{2\kappa(1-\gamma)}\right) \frac{16\kappa^2}{e^2\gamma^2} \frac{k_0 \ln(T/\tau)^2}{T^2}$$

Put everything together,

$$\mathbb{E}[\hat{\mathcal{E}}_{k_0}] \leq \|w_0 - w^*\|^2 \exp\left(-\frac{\mu}{2L}\frac{\gamma - \gamma^{k_0}}{1-\gamma}\right) + c_5 \exp\left(\frac{\gamma^{k_0}}{2\kappa(1-\gamma)}\right) \frac{16\kappa^2}{e^2\gamma^2} \frac{k_0 \ln(T/\tau)^2}{T^2}$$

Now consider the regime $k \geq k_0$ where $L\eta_k - \hat{L}\eta \leq 0$

$$\mathbb{E}[\hat{\mathcal{E}}_{k+1}] \leq (1 - \frac{\mu}{2L}\gamma^k)\mathbb{E}[\hat{\mathcal{E}}_k] + 2L\kappa\zeta^2\sigma^2\frac{\nu_L^2}{4L}\gamma_k^2$$

$$\leq (1 - \frac{\mu}{2L}\gamma^k)\mathbb{E}[\hat{\mathcal{E}}_k] + \frac{\nu_L^2\sigma^2}{2L}\gamma_k^2$$

Unrolling the recursion from $k = k_0$ to $T$

$$\mathbb{E}[\hat{\mathcal{E}}_T] \leq \mathbb{E}[\hat{\mathcal{E}}_{k_0}] \prod_{k=k_0}^{T}(1 - \frac{\mu}{2L}\gamma_k) + \frac{\nu_L^2\kappa\zeta^2\sigma^2}{2L} \sum_{k=k_0}^{T} \gamma_k^2 \prod_{i=k+1}^{T}(1 - \frac{\mu}{L}\gamma_i)$$

Bounding the first term using Lemma 3,

$$\prod_{k=k_0}^{T}\left(1 - \frac{\mu}{2L}\gamma^k\right) \leq \exp\left(-\frac{\mu}{2L}\frac{\gamma^{k_0} - \gamma^{T+1}}{1-\gamma}\right)$$

Bounding the second term using Lemma 4 similar to (Vaswani et al., 2022, Section C3)

$$\sum_{k=k_0}^{T} \gamma_k^2 \prod_{i=k+1}^{T}\left(1 - \frac{\mu}{2L}\gamma_i\right) \leq \exp\left(\frac{\gamma^{T+1}}{2\kappa(1-\gamma)}\right) \frac{16\kappa^2}{e^2\gamma^2} \frac{(T - k_0 + 1)\ln(T/\tau)^2}{T^2}$$

Hence, put everything together

$$\mathbb{E}[\hat{\mathcal{E}}_T] \leq \mathbb{E}[\hat{\mathcal{E}}_{k_0}] \exp\left(-\frac{\mu}{2L}\frac{\gamma^{k_0} - \gamma^{T+1}}{1-\gamma}\right) + \frac{\nu_L^2\kappa\zeta^2\sigma^2}{2L} \exp\left(\frac{\gamma^{T+1}}{2\kappa(1-\gamma)}\right) \frac{16\kappa^2}{e^2\gamma^2} \frac{(T - k_0 + 1)\ln(T/\tau)^2}{T^2}$$

Combining the bounds for two regimes

$$\mathbb{E}[\hat{\mathcal{E}}_T] \leq \exp\left(-\frac{\mu}{2L}\frac{\gamma^{k_0} - \gamma^{T+1}}{1-\gamma}\right) \left(\|w_0 - w^*\|^2 \exp\left(-\frac{\mu}{2L}\frac{\gamma - \gamma^{k_0}}{1-\gamma}\right) + c_5 \exp\left(\frac{\gamma^{k_0}}{2\kappa(1-\gamma)}\right) \frac{16\kappa^2}{e^2\gamma^2} \frac{k_0 \ln(T/\tau)^2}{T^2}\right)$$

$$+ \frac{\nu_L^2\kappa\zeta^2\sigma^2}{2L} \exp\left(\frac{\gamma^{T+1}}{2\kappa(1-\gamma)}\right) \frac{16\kappa^2}{e^2\gamma^2} \frac{(T - k_0 + 1)\ln(T/\tau)^2}{T^2}$$

$$= \|w_0 - w^*\|^2 \exp\left(-\frac{\mu}{2L}\frac{\gamma - \gamma^{T+1}}{1-\gamma}\right) + c_5 \exp\left(\frac{\gamma^{T+1}}{2\kappa(1-\gamma)}\right) \frac{16\kappa^2}{e^2\gamma^2} \frac{k_0 \ln(T/\tau)^2}{T^2}$$

$$+ \frac{\nu_L^2\kappa\zeta^2\sigma^2}{2L} \exp\left(\frac{\gamma^{T+1}}{2\kappa(1-\gamma)}\right) \frac{16\kappa^2}{e^2\gamma^2} \frac{(T - k_0 + 1)\ln(T/\tau)^2}{T^2}$$

Using Lemma 3 to bound the first term and noting that $\frac{\gamma^{T+1}}{1-\gamma} \leq \frac{2\tau}{\ln(T/\tau)}$, let $c_2 = \exp\left(\frac{1}{2\kappa}\frac{2\tau}{\ln(T/\tau)}\right)$

$$\mathbb{E}[\hat{\mathcal{E}}_T] \leq \|w_0 - w^*\|^2 \exp\left(-\frac{T}{2\kappa}\frac{\gamma}{\ln(T/\tau)}\right) + c_5 \frac{16c_2\kappa^2}{e^2\gamma^2} \frac{k_0 \ln(T/\tau)^2}{T^2} + \frac{\nu_L^2\kappa\zeta^2\sigma^2}{2L} \frac{16c_2\kappa^2}{e^2\gamma^2} \frac{(T - k_0 + 1)\ln(T/\tau)^2}{T^2}$$

Substitute the value of $c_5$ and $k_0$ we have

$$\mathbb{E}[\hat{\mathcal{E}}_T] \leq \|w_0 - w^*\|^2 \exp\left(-\frac{T}{2\kappa}\frac{\gamma}{\ln(T/\tau)}\right) + \frac{\nu_L^2 \kappa \zeta^2 \sigma^2}{LT}\frac{8c_2\kappa^2 \ln(T/\tau)^2}{e^2\gamma^2}$$
$$+ 32\left(\kappa\zeta^2\sigma^2 + 2\Delta_f\frac{\nu_L - 1}{\nu_L}\right)\frac{L}{T}\frac{c_2\kappa^2 \ln(\nu_L)\ln(T/\tau)}{e^2\gamma^2}$$

Combining the statements from $\nu_L \leq 1$ and $\nu_L > 1$ gives us

$$\mathbb{E}[\hat{\mathcal{E}}_T] \leq \|w_0 - w^*\|^2 c_2 \exp\left(-\frac{\min\{\nu_L, 1\}T}{2\kappa}\frac{\gamma}{\ln(T/\tau)}\right)$$
$$+ \frac{32Lc_2\kappa^2 \ln(T/\tau)}{e^2\gamma^2 T}\left(\max\left\{1, \frac{\nu_L^2}{4L}\right\}\ln(T/\tau)\kappa\zeta^2\sigma^2 + \max\{0, \ln(\nu_L)\}\left(\kappa\zeta^2\sigma^2 + 2\Delta_f\frac{\nu_L - 1}{\nu_L}\right)\right)$$

The next step is to remove the $\hat{L}$ from the LHS, and obtain a better measure of sub-optimality. By Lemma 1,

$$\mathbb{E}[\hat{\mathcal{E}}_T] \geq \underbrace{\frac{4(1-\gamma)}{\mu^2}\left[1 - \exp\left(-\frac{\mu\gamma}{2L}\right)\right]}_{:=c_L}\|w_{T-1} - w^*\|^2$$

Note that $c_L > 0$ is constant w.r.t $T$. Hence,

$$\mathbb{E}\|w_{T-1} - w^*\|^2 \leq \|w_0 - w^*\|^2 \frac{c_2}{c_L}\exp\left(-\frac{\min\{\nu_L, 1\}T}{2\kappa}\frac{\gamma}{\ln(T/\tau)}\right)$$
$$+ \frac{c_2}{c_L}\frac{32L\kappa^2\ln(T/\tau)}{e^2\gamma^2 T}\left(\max\left\{1, \frac{\nu_L^2}{4L}\right\}\ln(T/\tau)\kappa\zeta^2\sigma^2 + \max\{0, \ln(\nu_L)\}\left(\kappa\zeta^2\sigma^2 + 2\Delta_f\frac{\nu_L - 1}{\nu_L}\right)\right)$$

$\square$

## C.2 $\mu$ misestimation

Next, we analyze the effect of misspecifying $\mu$, the strong-convexity parameter. We assume we have access to an estimate $\hat{\mu} = \mu\nu_\mu$ where $\nu_\mu$ is the degree of misspecification. We note that $\hat{\mu}$ is a deterministic estimate of $\mu$. We only consider the case where we underestimate $\mu$, and hence $\nu_\mu \leq 1$. This is the typical case in practice – for example, while optimizing regularized convex loss functions in supervised learning, $\hat{\mu}$ is set to the regularization strength, and thus underestimates the true strong-convexity parameter.

Theorem 7 below demonstrates an $O\left(\exp\left(-\frac{\nu_\mu T}{\kappa}\right) + \frac{1}{\nu_\mu^2 T}\right)$ convergence to the minimizer. Hence, SHB with an underestimate of the strong-convexity results in slower convergence to the minimizer, with the slowdown again depending on the amount of misspecification.

**Theorem 7.** Under the same settings as Theorem 1, SHB (Eq. (2)) with the estimated $\hat{\mu} = \nu_\mu\mu$ for $\nu_\mu \leq 1$, results in the following convergence,

$$\mathbb{E}\|w_{T-1} - w^*\|^2 \leq \|w_0 - w^*\|^2 \frac{c_2}{c_\mu}\exp\left(-\frac{\nu_\mu T}{2\kappa}\frac{\gamma}{\ln(T/\tau)}\right) + \frac{32L\zeta^2 c_2\kappa^3}{\nu_\mu^2 e^2\gamma^2 c_\mu}\frac{(\ln(T/\tau))^2}{T}\sigma^2$$

where $c_2 = \exp\left(\frac{1}{2\kappa}\frac{2\tau}{\ln(T/\tau)}\right)$ and $c_\mu = \frac{4(1-\gamma)}{\nu_\mu^2\mu^2}\left[1 - \exp\left(-\frac{\nu_\mu\mu\gamma}{2L}\right)\right]$

*Proof.* Suppose we estimate $\mu$ to be $\hat{\mu}$. Now redefine

$$\hat{\lambda}_k = \frac{1 - 2\eta L}{\eta_k\hat{\mu}}\left(1 - (1 - \eta_k\hat{\mu})^k\right); \quad \hat{A}_k = \left\|w_k - w^* + \hat{\lambda}_k(w_k - w_{k-1})\right\|^2$$

$$\hat{B}_k = 2\eta_k\hat{\lambda}_k(f(w_{k-1}) - f(w^*))\,;\; \hat{\mathcal{E}}_k = \hat{A}_k + \hat{B}_k$$

Follow Theorem 5 first part steps with the new definition, the only difference was at the step where we use strongly-convex on $f$ for $-2\eta_k\langle w_k - w^*, \nabla f(w_k)\rangle \leq -\eta_k\mu\|w_k - w^*\|^2 - 2\eta_k[f(w_k) - f^*]$.

$$\mathbb{E}[\hat{\mathcal{E}}_{k+1}] \leq (1 - \eta_k\hat{\mu})\mathbb{E}[\hat{\mathcal{E}}_k] + 2L\kappa\zeta^2\eta_k^2\sigma^2 + \eta_k(\hat{\mu} - \mu)\|w_k - w^*\|^2$$

$$= (1 - \eta_k\nu_\mu\mu)\mathbb{E}[\hat{\mathcal{E}}_k] + 2L\kappa\zeta^2\eta_k^2\sigma^2 + \eta_k(\hat{\mu} - \mu)\|w_k - w^*\|^2$$

$$\leq (1 - \eta_k\nu_\mu\mu)\mathbb{E}[\hat{\mathcal{E}}_k] + 2L\kappa\zeta^2\eta_k^2\sigma^2 + \eta_k\mu(\nu_\mu - 1)\frac{2}{\mu}[f(w_k) - f^*] \qquad \text{(since } f \text{ is strongly-convex)}$$

$$= (1 - \eta_k\nu_\mu\mu)\mathbb{E}[\hat{\mathcal{E}}_k] + 2L\kappa\zeta^2\eta_k^2\sigma^2 + 2\eta_k(\nu_\mu - 1)[f(w_k) - f^*]$$

Since $\nu_\mu \leq 1$ then $2\eta_k(\nu_\mu - 1)[f(w_k) - f^*] \leq 0$ so

$$\mathbb{E}[\hat{\mathcal{E}}_{k+1}] \leq (1 - \eta_k\nu_\mu\mu)\mathbb{E}[\hat{\mathcal{E}}_k] + 2L\kappa\zeta^2\eta_k^2\sigma^2$$

Hence, following the same proof as Theorem 1

$$\mathbb{E}[\hat{\mathcal{E}}_T] \leq \|w_0 - w^*\|^2 c_2 \exp\left(-\frac{\nu_\mu T}{2\kappa}\frac{\gamma}{\ln(T/\tau)}\right) + \frac{32L\zeta^2\sigma^2 c_2\kappa^3}{\nu_\mu^2 e^2}\frac{(\ln(T/\tau))^2}{\gamma^2 T}$$

By Lemma 1,

$$\mathbb{E}[\hat{\mathcal{E}}_T] \geq \underbrace{\frac{4(1 - \gamma)}{\nu_\mu^2\mu^2}\left[1 - \exp\left(-\frac{\nu_\mu\mu\gamma}{2L}\right)\right]}_{:=c_\mu}\|w_{T-1} - w^*\|^2$$

Note that $c_\mu > 0$ is constant w.r.t $T$. Hence,

$$\|w_{T-1} - w^*\|^2 \leq \|w_0 - w^*\|^2\frac{c_2}{c_\mu}\exp\left(-\frac{\nu_\mu T}{2\kappa}\frac{\gamma}{\ln(T/\tau)}\right) + \frac{32L\zeta^2 c_2\kappa^3}{\nu_\mu^2 e^2\gamma^2 c_\mu}\frac{(\ln(T/\tau))^2}{T}\sigma^2$$

$$\square$$

# D   Proofs for upper bound SHB

**Lemma 5.** *For $L$-smooth and $\mu$ strongly-convex quadratics, SHB (Eq. (1)) with $\alpha_k = \alpha = \frac{a}{L}$ and $a \leq 1$, $\beta_k = \beta = \left(1 - \frac{1}{2}\sqrt{\alpha\mu}\right)^2$, batch-size $b$ satisfies the following recurrence relation,*

$$\mathbb{E}[\|\Delta_T\|] \leq C_0\,\rho^T\,\|\Delta_0\| + 2aC_0\,\zeta(b)\left[\sum_{k=0}^{T-1}\rho^{T-1-k}\,\mathbb{E}\,\|\Delta_k\|\right] + \frac{aC_0\,\chi\,\zeta(b)}{L}\left[\sum_{k=0}^{T-1}\rho^{T-1-k}\right],$$

*where $\Delta_k := \begin{bmatrix} w_k - w^* \\ w_{k-1} - w^* \end{bmatrix}$, $C_0 \leq 3\sqrt{\frac{\kappa}{a}}$, $\zeta(b) = \sqrt{3\frac{n-b}{(n-1)b}}$ and $\rho = 1 - \frac{\sqrt{a}}{2\sqrt{\kappa}}$*

*Proof.* With the definition of SHB (1), if $\nabla f_{ik}(w)$ is the mini-batch gradient at iteration $k$, then, for quadratics,

$$\underbrace{\begin{bmatrix} w_{k+1} - w^* \\ w_k - w^* \end{bmatrix}}_{\Delta_{k+1}} = \underbrace{\begin{bmatrix} (1+\beta)I_d - \alpha A & -\beta I_d \\ I_d & 0 \end{bmatrix}}_{\mathcal{H}}\underbrace{\begin{bmatrix} w_k - w^* \\ w_{k-1} - w^* \end{bmatrix}}_{\Delta_k} + \alpha\underbrace{\begin{bmatrix} \nabla f(w_k) - \nabla f_{ik}(w_k) \\ 0 \end{bmatrix}}_{\delta_k}$$

$$\Delta_{k+1} = \mathcal{H}\Delta_k + \alpha\delta_k$$

Recursing from $k = 0$ to $T - 1$, taking norm and expectation w.r.t to the randomness in all iterations.

$$\mathbb{E}[\|\Delta_T\|] \leq \|\mathcal{H}^T \Delta_0\| + \alpha \mathbb{E}\left[\left\|\sum_{k=0}^{T-1} \mathcal{H}^{T-1-k} \delta_k\right\|\right]$$

Using Theorem 8 and Corollary 6, for any vector $v$, $\|\mathcal{H}^k v\| \leq C_0 \, \rho^k \, \|v\|$ where $\rho = \sqrt{\beta}$. Hence,

$$\mathbb{E}[\|\Delta_T\|] \leq C_0 \, \rho^T \, \|\Delta_0\| + \frac{C_0 \, a}{L} \left[\sum_{k=0}^{T-1} \rho^{T-1-k} \, \mathbb{E}\|\delta_k\|\right] \qquad (\alpha = \tfrac{a}{L})$$

In order to simplify $\delta_k$, we will use the result from Lemma 7 and Lohr (2021),

$$\mathbb{E}_k[\|\delta_k\|^2] = \mathbb{E}_k[\|\nabla f(w_k) - \nabla f_{i_k}(w_k)\|^2] = \frac{n-b}{(n-1)\,b} \, \mathbb{E}_i \|\nabla f(w_k) - \nabla f_i(w_k)\|^2$$

(Sampling with replacement where $b$ is the batch-size and $n$ is the total number of examples)

$$= \frac{n-b}{(n-1)\,b} \, \mathbb{E}_i \|\nabla f(w_k) - \nabla f(w^*) - \nabla f_i(w_k) + \nabla f_i(w^*) - \nabla f_i(w^*)\|^2 \qquad (\nabla f(w^*) = 0)$$

$$\leq 3 \frac{n-b}{(n-1)\,b} \left[\mathbb{E}_i \|\nabla f(w_k) - \nabla f(w^*)\|^2 + \mathbb{E}_i \|\nabla f_i(w_k) - \nabla f_i(w^*)\|^2 + \mathbb{E}_i \|\nabla f_i(w^*)\|^2\right]$$

$$( \, (a+b+c)^2 \leq 3[a^2 + b^2 + c^2])$$

$$\leq 3 \frac{n-b}{(n-1)\,b} \left[L^2 \, \mathbb{E}_i \|w_k - w^*\|^2 + L^2 \, \mathbb{E}_i \|w_k - w^*\|^2 + \mathbb{E}_i \|\nabla f_i(w^*)\|^2\right]$$

(Using the $L$ smoothness of $f$ and $f_i$)

$$\leq 3 \frac{n-b}{(n-1)\,b} \left[2L^2 \, \|w_k - w^*\|^2 + \chi^2\right]$$

($w_k$ is independent of the randomness and by definition $\chi^2 = \mathbb{E}_i \|\nabla f_i(w^*)\|^2$)

$$\leq 3 \frac{n-b}{(n-1)\,b} \left[2L^2[\|w_k - w^*\|^2 + \|w_{k-1} - w^*\|^2] + \chi^2\right] \qquad (\|w_{k-1} - w^*\|^2 \geq 0)$$

$$\implies \mathbb{E}_k[\|\Delta_k\|^2] \leq 3 \frac{n-b}{(n-1)\,b} \left[2L^2 \, \|\Delta_k\|^2 + \chi^2\right] \qquad (\text{Definition of } \Delta_k)$$

$$\implies \mathbb{E}_k[\|\Delta_k\|] \leq \underbrace{\sqrt{3 \frac{n-b}{(n-1)\,b}}}_{:=\zeta(b)} \left[\sqrt{2L^2} \, \|\Delta_k\| + \chi\right]$$

(Taking square-roots, using Jensen's inequality on the LHS and $\sqrt{a+b} \leq \sqrt{a} + \sqrt{b}$ on the RHS)

$$\implies \mathbb{E}_k[\|\delta_k\|] \leq \sqrt{2}L \, \zeta(b) \, \|\Delta_k\| + \zeta(b) \, \chi$$

Putting everything together,

$$\mathbb{E}[\|\Delta_T\|] \leq C_0 \, \rho^T \, \|\Delta_0\| + \sqrt{2}aC_0 \, \zeta(b) \, \mathbb{E}\left[\sum_{k=0}^{T-1} \rho^{T-1-k} \, \|\Delta_k\|\right] + \frac{aC_0 \, \chi \, \zeta(b)}{L} \left[\sum_{k=0}^{T-1} \rho^{T-1-k}\right]$$

$$\square$$

**Theorem 2.** For $L$-smooth, $\mu$ strongly-convex quadratics, SHB (Eq. (1)) with $\alpha_k = \alpha = \frac{a}{L}$ for $a \leq 1$, $\beta_k = \beta = \left(1 - \frac{1}{2}\sqrt{\alpha\mu}\right)^2$, batch-size $b$ s.t. $b \geq b^* := n \max\left\{ \frac{1}{1 + \frac{n-1}{C\kappa^2}}, \frac{1}{1 + \frac{(n-1)a}{3}} \right\}$ converges as:

$$\mathbb{E}\left\|w_T - w^*\right\| \leq \frac{6\sqrt{2}\sqrt{\kappa}}{\sqrt{a}} \exp\left(-\frac{\sqrt{a}\,T}{2\sqrt{\kappa}} \max\left\{\frac{3}{4}, 1 - 2\sqrt{\kappa}\sqrt{\zeta}\right\}\right) \left\|w_0 - w^*\right\| + \frac{12\sqrt{a}\chi}{\mu} \min\left\{1, \frac{\zeta}{\sqrt{a}}\right\}$$

where $\chi := \sqrt{\mathbb{E}\left\|\nabla f_i(w^*)\right\|^2}$, $\zeta = \sqrt{3\frac{n-b}{(n-1)\,b}}$ and $C := 3^5 2^6$.

*Proof.* Using Lemma 5, we have that,

$$\mathbb{E}\left\|\Delta_T\right\| \leq C_0\,\rho^T\left\|\Delta_0\right\| + \sqrt{2}aC_0\,\zeta\left[\sum_{k=0}^{T-1}\rho^{T-1-k}\mathbb{E}\left\|\Delta_k\right\|\right] + \frac{aC_0\,\zeta\,\chi}{L}\left[\sum_{k=0}^{T-1}\rho^{T-1-k}\right]$$

We use induction to prove that for all $T \geq 1$,

$$\mathbb{E}\left\|\Delta_T\right\| \leq 2C_0\left[\rho + \sqrt{\zeta}\sqrt{a}\right]^T\left\|\Delta_0\right\| + \frac{2C_0\,\zeta\,a\,\chi}{L(1-\rho)}$$

where $\rho + \sqrt{\zeta}\sqrt{a} < 1$.

**Base case**: By Theorem 8, $C_0 \geq 1$ hence $\left\|\Delta_0\right\| \leq 2C_0\left\|\Delta_0\right\| + \frac{2C_0 a\zeta\,\chi}{L(1-\rho)}$

**Inductive hypothesis**: For all $k \in \{0, 1, \ldots, T-1\}$, $\left\|\Delta_k\right\| \leq 2C_0\left[\rho + \sqrt{\zeta}\sqrt{a}\right]^k\left\|\Delta_0\right\| + \frac{2C_0\,a\zeta\,\chi}{L(1-\rho)}$.

**Inductive step**: Using the above inequality,

$$\mathbb{E}\left\|\Delta_T\right\| \leq C_0\,\rho^T\left\|\Delta_0\right\| + \sqrt{2}aC_0\,\zeta\left[\sum_{k=0}^{T-1}\rho^{T-1-k}\mathbb{E}\left\|\Delta_k\right\|\right] + \frac{aC_0\,\zeta\,\chi}{L}\left[\sum_{k=0}^{T-1}\rho^{T-1-k}\right]$$

$$\leq C_0\left[\rho + \sqrt{\zeta}\sqrt{a}\right]^T\left\|\Delta_0\right\| + \sqrt{2}aC_0\,\zeta\left[\sum_{k=0}^{T-1}\rho^{T-1-k}\mathbb{E}\left\|\Delta_k\right\|\right] + \frac{aC_0\,\zeta\,\chi}{L}\left[\sum_{k=0}^{T-1}\rho^k\right] \quad \text{(Since } \zeta, a > 0\text{)}$$

$$\leq C_0\left[\rho + \sqrt{\zeta}\sqrt{a}\right]^T\left\|\Delta_0\right\| + \frac{\sqrt{2}aC_0\,\zeta}{\rho}\rho^T\left[\sum_{k=0}^{T-1}\rho^{-k}\left(2C_0\left[\rho + \sqrt{\zeta}\sqrt{a}\right]^k\left\|\Delta_0\right\| + \frac{2C_0\,a\zeta\,\chi}{L(1-\rho)}\right)\right]$$

$$+ \frac{aC_0\,\zeta\,\chi}{L}\frac{1 - \rho^T}{1 - \rho} \qquad \text{(Sum of geometric series and using the inductive hypothesis)}$$

$$= C_0\left[\rho + \sqrt{\zeta}\sqrt{a}\right]^T\left\|\Delta_0\right\| + \frac{2\sqrt{2}\,aC_0^2\,\zeta}{\rho}\rho^T\left[\sum_{k=0}^{T-1}\left(\frac{\rho + \sqrt{\zeta}\sqrt{a}}{\rho}\right)^k\right]\left\|\Delta_0\right\|$$

$$+ \frac{2\sqrt{2}\,a^2C_0^2\,\zeta^2\chi}{\rho L(1-\rho)}\rho^T\left[\sum_{k=0}^{T-1}\left(\frac{1}{\rho}\right)^k\right] + \frac{aC_0\,\zeta\,\chi}{L}\frac{1 - \rho^T}{1 - \rho}$$

First, we need to prove that $\frac{2\sqrt{2}\,aC_0^2\,\zeta}{\rho}\rho^T\left[\sum_{k=0}^{T-1}\left(\frac{\rho + \sqrt{\zeta}\sqrt{a}}{\rho}\right)^k\right]\left\|\Delta_0\right\| \leq C_0\left[\rho + \sqrt{\zeta}\sqrt{a}\right]^T\left\|\Delta_0\right\|$.

$$\frac{2\sqrt{2}\,aC_0^2\,\zeta}{\rho}\rho^T\left[\sum_{k=0}^{T-1}\left(\frac{\rho + \sqrt{\zeta}\sqrt{a}}{\rho}\right)^k\right]\left\|\Delta_0\right\| = \frac{2\sqrt{2}\,aC_0^2\,\zeta}{\rho}\rho^T\frac{\left(\frac{\rho + \sqrt{\zeta}\sqrt{a}}{\rho}\right)^T - 1}{\left(\frac{\rho + \sqrt{\zeta}\sqrt{a}}{\rho}\right) - 1}\left\|\Delta_0\right\|$$

$$\text{(Sum of geometric series)}$$

$$\leq 2\sqrt{2}\,\sqrt{a}C_0^2\,\sqrt{\zeta}\,\left(\rho+\sqrt{\zeta}\sqrt{a}\right)^T\|\Delta_0\|$$

Hence, we require that,

$$2\sqrt{2}\,\sqrt{a}C_0^2\,\sqrt{\zeta}\leq C_0 \implies \zeta\leq\frac{1}{8\,C_0^2}\frac{1}{a}$$

Hence it suffices to choose $\zeta$ s.t.

$$\implies \zeta\leq\frac{a}{3^2 2^3\kappa}\frac{1}{a} \qquad\qquad\text{(Since } C_0\leq 3\sqrt{\tfrac{\kappa}{a}}\text{)}$$

$$\implies \zeta\leq\frac{1}{3^2 2^3\kappa}$$

$$\implies \frac{n-b}{(n-1)\,b}\leq\frac{1}{3^5 2^6\,\kappa^2} \implies \frac{b}{n}\geq\frac{1}{1+\frac{n-1}{3^5 2^6\,\kappa^2}} \qquad\text{(Using the definition of } \zeta\text{)}$$

Since the batch-size $b$ satisfies the condition that: $\frac{b}{n}\geq\frac{1}{1+\frac{n-1}{C\,\kappa^2}}$ for $C:=15552=3^5 2^6$, the above requirement is satisfied, and $\zeta\leq\frac{1}{3^2 2^3\kappa}$.

Next, we need to show $D:=\frac{2\sqrt{2}a^2 C_0^2\,\zeta^2\chi}{\rho L(1-\rho)}\rho^T\left[\sum_{k=0}^{T-1}\left(\frac{1}{\rho}\right)^k\right]+\frac{aC_0\,\zeta\,\chi}{L}\frac{1-\rho^T}{1-\rho}\leq\frac{2C_0\,a\zeta\,\chi}{L(1-\rho)}$

$$D=\frac{2\sqrt{2}\,a^2 C_0^2\,\zeta^2\chi}{\rho L(1-\rho)}\rho^T\left[\sum_{k=0}^{T-1}\left(\frac{1}{\rho}\right)^k\right]+\frac{aC_0\,\zeta\,\chi}{L}\frac{1-\rho^T}{1-\rho}$$

$$=\frac{2\sqrt{2}\,a^2 C_0^2\,\zeta^2\chi}{\rho L(1-\rho)}\rho^T\frac{\left(\frac{1}{\rho}\right)^T-1}{\left(\frac{1}{\rho}\right)-1}+\frac{aC_0\,\zeta\,\chi}{L}\frac{1-\rho^T}{1-\rho} \qquad\text{(Sum of geometric series)}$$

$$<\frac{2\sqrt{2}\,a^2 C_0^2\,\zeta^2\chi}{\rho L(1-\rho)}\rho^T\frac{1-\rho^T}{1-\rho}\frac{\rho}{\rho^T}+\frac{aC_0\,\zeta\,\chi}{L(1-\rho)}$$

$$<\frac{2\sqrt{2}\,a^2 C_0^2\,\zeta^2\chi}{L(1-\rho)^2}+\frac{aC_0\,\zeta\,\chi}{L(1-\rho)}$$

Since we want $D\leq\frac{2aC_0\,\zeta\,\chi}{L(1-\rho)}$, we require that

$$\frac{2\sqrt{2}\,a^2 C_0^2\,\zeta^2\chi}{L(1-\rho)^2}\leq\frac{aC_0\,\zeta\,\chi}{L(1-\rho)}$$

$$\implies \frac{2\sqrt{2}\,C_0 a\zeta}{1-\rho}\leq 1$$

Ensuring this imposes an additional constraint on $\zeta$. We require $\zeta$ such that,

$$\zeta\leq\frac{1-\rho}{2\sqrt{2}\,C_0\,a} \implies \zeta\leq\frac{1}{4\sqrt{2}\,\sqrt{a}\,\sqrt{\kappa}}\frac{1}{C_0} \qquad\text{(Since } \rho=1-\frac{\sqrt{a}}{2\sqrt{\kappa}}\text{)}$$

Hence it suffices to choose $\zeta$ such that,

$$\zeta\leq\frac{1}{12\sqrt{2}\kappa} \qquad\qquad\text{(Since } C_0\leq 3\sqrt{\tfrac{\kappa}{a}}\text{)}$$

Since the condition on the batch-size ensures that $\zeta\leq\frac{1}{3^2 2^3\kappa}$, this condition is satisfied. Hence,

$$\mathbb{E}\|\Delta_T\|\leq 2C_0\left[\rho+\sqrt{\zeta}\sqrt{a}\right]^T\|\Delta_0\|+\frac{2C_0\,a\,\zeta\,\chi}{L(1-\rho)}$$

This completes the induction.

In order to bound the noise term as $\frac{12\sqrt{a}\chi}{\mu} \min\left\{1, \frac{\zeta}{\sqrt{a}}\right\}$, we will require an additional constraint on the batch-size that ensures $\zeta \leq \sqrt{a}$. Using the definition of $\zeta$, we require that,

$$\sqrt{3\frac{n-b}{(n-1)b}} \leq \sqrt{a}$$

$$\implies \frac{b}{n} \geq \frac{1}{1 + \frac{(n-1)a}{3}},$$

which is satisfied by the condition on the batch-size. From the result of the induction,

$$
\begin{aligned}
\mathbb{E}\left\|\Delta_T\right\| &\leq 2C_0 \left[\rho + \sqrt{\zeta}\sqrt{a}\right]^T \|\Delta_0\| + \frac{2C_0\, a\, \zeta\, \chi}{L(1-\rho)} \\
&= 2C_0 \left[1 - \frac{\sqrt{a}}{2\sqrt{\kappa}} + \sqrt{\zeta}\sqrt{a}\right]^T \|\Delta_0\| + \frac{2C_0\, a\, \zeta\, \chi}{L}\frac{2\sqrt{\kappa}}{\sqrt{a}} && \left(\rho = 1 - \frac{\sqrt{a}}{2\sqrt{\kappa}}\right) \\
&= 2C_0 \left[1 - \frac{\sqrt{a}}{2\sqrt{\kappa}}\left(1 - 2\sqrt{\kappa}\sqrt{\zeta}\right)\right]^T \|\Delta_0\| + \frac{2C_0\, a\, \zeta\, \chi}{L}\frac{2\sqrt{\kappa}}{\sqrt{a}} \\
&\qquad \left(1 - \tfrac{1}{3\sqrt{2}} \leq \left(1 - 2\sqrt{\kappa}\sqrt{\zeta}\right) \leq 1 \text{ because of the constraint on batch-size}\right) \\
&\leq 6\sqrt{\frac{\kappa}{a}}\left[1 - \frac{\sqrt{a}}{2\sqrt{\kappa}}\max\left\{\frac{3}{4}, 1 - 2\sqrt{\kappa}\sqrt{\zeta}\right\}\right]^T \|\Delta_0\| + \frac{2a\,\zeta\,\chi}{L}3\sqrt{\frac{\kappa}{a}}\frac{2\sqrt{\kappa}}{\sqrt{a}} && \left(C_0 \leq 3\sqrt{\tfrac{\kappa}{a}}\right) \\
&\leq \frac{6}{\sqrt{a}}\sqrt{\kappa}\left[1 - \frac{\sqrt{a}}{2\sqrt{\kappa}}\max\left\{\frac{3}{4}, 1 - 2\sqrt{\kappa}\sqrt{\zeta}\right\}\right]^T \|\Delta_0\| + \frac{12\sqrt{a}\chi}{\mu}\min\left\{1, \frac{\zeta}{\sqrt{a}}\right\} && \left(\zeta \leq \sqrt{a}\right) \\
\implies \mathbb{E}\left\|w_T - w^*\right\| &\leq \frac{6\sqrt{2}}{\sqrt{a}}\sqrt{\kappa}\exp\left(-\frac{T}{\sqrt{\kappa}}\frac{\sqrt{a}}{2}\max\left\{\frac{3}{4}, 1 - 2\sqrt{\kappa}\sqrt{\zeta}\right\}\right)\|w_0 - w^*\| + \frac{12\sqrt{a}\chi}{\mu}\min\left\{1, \frac{\zeta}{\sqrt{a}}\right\} \\
&\qquad \text{(for all } x,\, 1 - x \leq \exp(-x))
\end{aligned}
$$

$\square$

**Corollary 1.** For $L$-smooth, $\mu$ strongly-convex quadratics, under interpolation, SHB (Eq. (1)) with the same parameters as in Theorem 2 and batch-size $b$ s.t. $b \geq b^* := n\frac{1}{1 + \frac{n-1}{C\kappa^2}}$ (where $C$ is defined in Theorem 2) converges as:

$$\mathbb{E}\left\|w_T - w^*\right\| \leq \frac{6\sqrt{2}}{\sqrt{a}}\sqrt{\kappa}\exp\left(-\frac{T}{\sqrt{\kappa}}\frac{\sqrt{a}}{2}\max\left\{\frac{3}{4}, 1 - 2\sqrt{\kappa}\sqrt{\zeta}\right\}\right)\|w_0 - w^*\|$$

*Proof.* Under interpolation $\chi = 0$. This removes the additional constraint on $b^*$ that depends on the constant $a$, finishing the proof. $\square$

**Corollary 4.** Under the same conditions of Theorem 2, for a target error $\epsilon > 0$, setting $a := \min\left\{1, \left(\frac{\mu}{24\chi}\right)^2\epsilon\right\}$ and $T \geq \frac{2\sqrt{\kappa}}{\sqrt{a}\left(1 - 2\sqrt{\kappa}\sqrt{\zeta}\right)}\log\left(\frac{12\sqrt{2}\sqrt{\kappa}\|w_0 - w^*\|}{\sqrt{a\epsilon}}\right)$ ensures that $\|w_T - w^*\| \leq \sqrt{\epsilon}$.

*Proof.* Using Theorem 2, we have that,

$$E\left\|w_T - w^*\right\| \leq \frac{6\sqrt{2}}{\sqrt{a}}\sqrt{\kappa}\exp\left(-\frac{\sqrt{a}\left(1 - 2\sqrt{\kappa}\sqrt{\zeta}\right)}{2}\frac{T}{\sqrt{\kappa}}\right)\|w_0 - w^*\| + \frac{12\sqrt{a}\chi}{\mu}\min\left\{1, \frac{\zeta}{\sqrt{a}}\right\}$$

Using the step-size similar to that for SGD in (Gower et al., 2019, Theorem 3.1), we see that to get $\sqrt{\epsilon}$ accuracy first we consider $\frac{12\sqrt{a}\chi}{\mu} \leq \frac{\sqrt{\epsilon}}{2}$ that implies $a \leq (\frac{\mu}{24\chi})^2\epsilon$.

We also need $\frac{6\sqrt{2}}{\sqrt{a}}\sqrt{\kappa} \exp\left(-\frac{\sqrt{a}\left(1-2\sqrt{\kappa}\sqrt{\zeta}\right)}{2}\frac{T}{\sqrt{\kappa}}\right)\|w_0 - w^*\| \leq \frac{\sqrt{\epsilon}}{2}$. Taking log on both sides,

$$\left(-\frac{\sqrt{a}\left(1 - 2\sqrt{\kappa}\sqrt{\zeta}\right)}{2}\frac{T}{\sqrt{\kappa}}\right) \leq \log\left(\frac{\sqrt{\epsilon}}{2}\frac{\sqrt{a}}{6\sqrt{2}\sqrt{\kappa}}\frac{1}{\|w_0 - w^*\|}\right)$$

$$\implies T \geq \frac{2\sqrt{\kappa}}{\sqrt{a}\left(1 - 2\sqrt{\kappa}\sqrt{\zeta}\right)}\log\left(\frac{12\sqrt{2}\sqrt{\kappa}\|w_0 - w^*\|}{\sqrt{a\epsilon}}\right)$$

$\square$

### D.1 Helper Lemmas

We restate (Wang et al., 2021, Theorem 5) that we used in our proof.

**Theorem 8.** Let $H := \begin{bmatrix} (1+\beta)I_d - \alpha A & \beta I_d \\ I_d & 0 \end{bmatrix} \in \mathbb{R}^{2d \times 2d}$ where $A \in \mathbb{R}^{dn \times d}$ is a positive definite matrix. Fix a vector $v_0 \in \mathbb{R}^d$. If $\beta$ is chosen to satisfy

$1 \geq \beta \geq \max\left\{\left(1 - \sqrt{\alpha\lambda_{\min}(A)}\right)^2, \left(1 - \sqrt{\alpha\lambda_{\max}(A)}\right)^2\right\}$ then

$$\left\|H^k v_0\right\| \leq \left(\sqrt{\beta}\right)^k C_0 \left\|v_0\right\|$$

where the constant

$$C_0 := \frac{\sqrt{2}(\beta+1)}{\sqrt{\min\left\{h\left(\beta, \alpha\lambda_{\min}(A)\right), h\left(\beta, \alpha\lambda_{\max}(A)\right)\right\}}} \geq 1$$

and $h\left(\beta, z\right) := -\left(\beta - (1 - \sqrt{z})^2\right)\left(\beta - (1 + \sqrt{z})^2\right)$.

**Lemma 6.** *For a positive definite matrix $A$, denote $\kappa := \frac{\lambda_{\max}(A)}{\lambda_{\min}(A)} = \frac{L}{\mu}$. Set $\alpha = \frac{a}{\lambda_{\max}(A)} = \frac{a}{L}$ for $a \leq 1$ and $\beta = \left(1 - \frac{1}{2}\sqrt{\alpha\lambda_{\min}(A)}\right)^2 = \left(1 - \frac{\sqrt{a}}{2\sqrt{\kappa}}\right)^2$. Then, $C_0 := \frac{\sqrt{2}(\beta+1)}{\sqrt{\min\{h(\beta,\alpha\lambda_{\min}(A)),h(\beta,\alpha\lambda_{\max}(A))\}}} \leq 3\sqrt{\frac{\kappa}{a}}$ and $h\left(\beta, z\right) := -\left(\beta - (1 - \sqrt{z})^2\right)\left(\beta - (1 + \sqrt{z})^2\right)$.*

*Proof.* Using the definition of $h\left(\beta, z\right)$ with the above setting for $\beta$ and simplifying,

$$h(\beta, \alpha\mu) = 3\alpha\mu\left(1 - \frac{1}{2}\sqrt{\alpha\mu} - \frac{3}{16}\alpha\mu\right)$$

$$= 3\frac{a}{\kappa}\left(1 - \frac{\sqrt{a}}{2\sqrt{\kappa}} - \frac{3a}{16\kappa}\right) \qquad (\alpha = \tfrac{a}{L})$$

$$\geq 3\frac{a}{\kappa}\left(1 - \frac{1}{2\sqrt{\kappa}} - \frac{3}{16\kappa}\right) \qquad (a \leq 1)$$

$$\geq 3\frac{a}{\kappa}\left(1 - \frac{1}{2} - \frac{3}{16}\right) = \frac{15}{16}\frac{a}{\kappa} \qquad (\kappa \geq 1)$$

$$\implies \frac{\sqrt{2}(1+\beta)}{\sqrt{h(\beta, \alpha\mu)}} \leq \frac{2\sqrt{2}}{\sqrt{\frac{15}{16}\frac{a}{\kappa}}} = \frac{8\sqrt{2}\sqrt{\kappa}}{\sqrt{15a}} \leq 3\sqrt{\frac{\kappa}{a}} \qquad (\beta \leq 1)$$

Now we need to bound $\frac{\sqrt{2}(1+\beta)}{\sqrt{h(\beta,\alpha L)}}$. Using the definition of $h\left(\beta, z\right)$ and simplifying,

$$h(\beta, \alpha L) = (2\sqrt{\alpha L} - \sqrt{\alpha\mu} - \alpha L + \frac{1}{4}\alpha\mu)(\sqrt{\alpha\mu} + 2\sqrt{\alpha L} + \alpha L - \frac{1}{4}\alpha\mu)$$

$$= 4a - \frac{a}{\kappa} - 2\frac{a^{3/2}}{\sqrt{\kappa}} + \frac{1}{2}\frac{a^{3/2}}{\kappa^{3/2}} - a^2\left[1 - \frac{1}{2\kappa} + \frac{1}{16\kappa^2}\right] \qquad \text{(setting } \alpha = a/L \text{ and expanding above)}$$

$$= a\left[4 - \frac{1}{\kappa} - 2\frac{a^{1/2}}{\sqrt{\kappa}} + \frac{1}{2}\frac{a^{1/2}}{\kappa^{3/2}} - a\left(1 - \frac{1}{2\kappa} + \frac{1}{16\kappa^2}\right)\right]$$

$$= a\left[4 - \frac{1}{\kappa} - \sqrt{a}\left(\frac{2}{\sqrt{\kappa}} - \frac{1}{2\kappa^{3/2}}\right) - a\left(1 - \frac{1}{2\kappa} + \frac{1}{16\kappa^2}\right)\right]$$

Since $\kappa \geq 1$, $\frac{2}{\sqrt{\kappa}} - \frac{1}{2\kappa^{3/2}} > 0$ and $1 - \frac{1}{2\kappa} + \frac{1}{16\kappa^2} > 0$, hence

$$h(\beta, \alpha L) \geq a\left[4 - \frac{1}{\kappa} - \left(\frac{2}{\sqrt{\kappa}} - \frac{1}{2\kappa^{3/2}}\right) - \left(1 - \frac{1}{2\kappa} + \frac{1}{16\kappa^2}\right)\right] \qquad (a \leq \sqrt{a} \leq 1)$$

$$= a\left[4 - \left(\frac{2}{\sqrt{\kappa}} - \frac{1}{2\kappa^{3/2}}\right) - \left(1 + \frac{1}{2\kappa} + \frac{1}{16\kappa^2}\right)\right]$$

Both $\frac{2}{\sqrt{\kappa}} - \frac{1}{2\kappa^{3/2}}$ and $1 + \frac{1}{2\kappa} + \frac{1}{16\kappa^2}$ are decreasing functions of $\kappa$ for $\kappa \geq 1$.

Hence, $\text{RHS}(\kappa) := \left[4 - \left(\frac{2}{\sqrt{\kappa}} - \frac{1}{2\kappa^{3/2}}\right) - \left(1 + \frac{1}{2\kappa} + \frac{1}{16\kappa^2}\right)\right]$ is an increasing function of $\kappa$. Since,
$h(\beta, \alpha L) \geq \text{RHS}(\kappa) \geq \text{RHS}(1)$ for all $\kappa \geq 1$,

$$h(\beta, \alpha L) \geq a\left[4 - 2 + \frac{1}{2} - 1 - \frac{1}{2} - \frac{1}{16}\right] = \frac{15a}{16} \qquad (\beta \leq 1)$$

Using the above lower-bound for $\frac{\sqrt{2}(1+\beta)}{\sqrt{h(\beta, \alpha L)}}$ we have

$$\frac{\sqrt{2}(1+\beta)}{\sqrt{h(\beta, \alpha L)}} \leq \frac{8\sqrt{2}}{\sqrt{15a}} \leq \frac{3}{\sqrt{a}}$$

Putting everything together we get,

$$C_0 \leq \max\left\{3\sqrt{\frac{\kappa}{a}}, \frac{3}{\sqrt{a}}\right\} \implies C_0 \leq 3\sqrt{\frac{\kappa}{a}}$$

**Lemma 7.** *For batch sampling method where each batch is sampling without replacement from the dataset.*

$$\mathbb{E}\left[\|\nabla f_b(w_k) - \nabla f(w_k)\|^2\right] = \frac{n-b}{(n-1)b}\mathbb{E}\left[\|\nabla f_i(w_k) - \nabla f(w_k)\|^2\right]$$

*where $\nabla f_b(w_k) = \frac{1}{b}\sum_{i \in \mathcal{B}} \nabla f_i(w_k)$*

*Proof.* First, $\mathbb{E}[\nabla f_b(w_k)] = \mathbb{E}\left[\frac{1}{b}\sum_{i \in \mathcal{B}} \nabla f_i(w_k)\right] = \frac{1}{b}\sum_{i \in \mathcal{B}} \mathbb{E}[\nabla f_i(w_k)] = \frac{1}{b}\sum_{i \in \mathcal{B}} \nabla f(w_k) = \nabla f(w_k)$. Then we will calculate the variance of $\nabla f_b(w_k)$,

$$\text{Var}\left(\nabla f_b(w_k)\right) = \text{Var}\left(\frac{1}{b}\sum_{i \in \mathcal{B}} \nabla f_i(w_k)\right)$$

$$= \frac{1}{b^2}\text{Var}\left(\sum_{i \in \mathcal{B}} \nabla f_i(w_k)\right)$$

$$= \frac{1}{b^2}\text{Var}\left(\sum_{i=1}^{n} \nabla f_i(w_k)X_i\right)$$

where $X_i$ is an indicator if sample $i$ is in the batch $\mathcal{B}$

$$\implies \text{Var}\left(\nabla f_b(w_k)\right) = \frac{1}{b^2}\left(\sum_{i=1}^{n} \text{Var}\left[\nabla f_i(w_k)X_i\right] + 2\sum_{\substack{j \neq k}}^{j,k \in \mathcal{N}} \text{Cov}\left[\nabla f_j(w_k)X_j, \nabla f_k(w_k)X_k\right]\right)$$

Denote $\nabla f_i(w_k) = \nabla_i$ and $\nabla f_b(w_k) = \nabla_b$ for simplification, hence

$$\text{Var}\,[\nabla_i X_i] = \nabla_i^2 \text{Var}\,[X_i] = \nabla_i^2 \frac{b}{n}\frac{n-b}{n} \qquad \text{(a sample is in the batch with probability } \tfrac{b}{n})$$

$$\text{Cov}\,[\nabla_j X_j, \nabla_k X_k] = \mathbb{E}[\nabla_j X_j \nabla_k X_k] - \mathbb{E}[\nabla_j X_j]\mathbb{E}[\nabla_k X_k]$$
$$= \nabla_j \nabla_k \left(\mathbb{E}[X_j X_k] - \mathbb{E}[X_j]\mathbb{E}[X_k]\right)$$

Since $\mathbb{E}[X_j X_k] = \Pr[\text{both samples } i,j \text{ are in the batch}] = \binom{n-2}{b-2}/\binom{n}{b}$ and $\mathbb{E}[X_j] = \mathbb{E}[X_k] = \tfrac{b}{n}$,

$$\implies \text{Cov}\,[\nabla_j X_j, \nabla_k X_k] = \nabla_j \nabla_k \left(\frac{b(b-1)}{n(n-1)} - \frac{b^2}{n^2}\right)$$
$$= \nabla_j \nabla_k \frac{b(b-n)}{n^2(n-1)}$$

Plug back to $\text{Var}\,(\nabla_b)$ then,

$$\text{Var}\,(\nabla_b) = \frac{1}{b^2}\left(\frac{b(n-b)}{n^2}\left[\sum_{i=1}^{n}\nabla_i^2\right] + 2\frac{b(b-n)}{n^2(n-1)}\left[\sum_{\substack{j\neq k}}^{j,k\in\mathcal{N}}\nabla_j\nabla_k\right]\right)$$
$$= \frac{n-b}{(n-1)b}\left(\frac{n-1}{n^2}\left[\sum_{i=1}^{n}\nabla_i^2\right] - \frac{2}{n^2}\left[\sum_{j\neq k}\nabla_j\nabla_k\right]\right)$$
$$= \frac{n-b}{(n-1)b}\left(\left[\frac{1}{n}\sum_{i=1}^{n}\nabla_i^2\right] - \frac{1}{n^2}\left[\sum_{i=1}^{n}\nabla_i^2 + 2\sum_{j\neq k}\nabla_j\nabla_k\right]\right)$$
$$= \frac{n-b}{(n-1)b}\left(\mathbb{E}[\nabla_i^2] - \left(\mathbb{E}[\nabla_i^2]\right)^2\right)$$
$$= \frac{n-b}{(n-1)b}\text{Var}\,(\nabla_i)$$
$$\implies \mathbb{E}\left[\|\nabla f_b(w_k) - \nabla f(w_k)\|^2\right] = \frac{n-b}{(n-1)b}\mathbb{E}\left[\|\nabla f_i(w_k) - \nabla f(w_k)\|^2\right]$$

$\square$

## E   Proofs for lower bound SHB

Before looking at the general lower-bound for $n$ samples, it is instructive to consider the lower-bound arguments for a 2-sample example. The arguments for the general $n$-sample example are similar.

**Theorem 9.** For a $\bar{L}$-smooth, $\bar{\mu}$ strong-convex quadratics problem $f(w) := \frac{1}{2}\sum_{i=1}^{2}\frac{1}{2}w^T A_i w$ with 2 samples and dimension $d = n = 2$ such that $w^* = 0$ and each $A_i$ is a 2-by-2 matrix of all zeros except at the $(i,i)$ position, we run SHB (1) with $\alpha_k = \alpha = \frac{1}{L}$, $\beta_k = \beta = \left(1 - \frac{1}{2}\sqrt{\alpha\bar{\mu}}\right)^2$. With a batch-size 1, when $\kappa > 6$, after $3T$ iterations, we have the following: if $\Delta_k := \begin{pmatrix} w_k \\ w_{k-1}\end{pmatrix}$, for a $c = 1.1 > 1$,

$$\mathbb{E}[\|\Delta_{3T}\|^2] > c^T \|\Delta_0\|^2$$

*Proof.* By definition, the 2 samples are: $A_1 = \begin{pmatrix} \mu & 0 \\ 0 & 0 \end{pmatrix}$ $A_2 = \begin{pmatrix} 0 & 0 \\ 0 & L \end{pmatrix}$, and hence $A = \frac{1}{2}\begin{pmatrix} \mu & 0 \\ 0 & L \end{pmatrix}$.

Calculating the smoothness and strong-convexity of the resulting problem, $\bar{L} = \frac{L}{2}, \bar{\mu} = \frac{\mu}{2}, \kappa = \frac{L}{\mu}$. By the

definition of SHB (1), we have that,

$$\begin{bmatrix} w_{k+1} - w^* \\ w_k - w^* \end{bmatrix} = \begin{bmatrix} (1+\beta)I_d - \alpha A_k & -\beta I_d \\ I_d & 0 \end{bmatrix} \begin{bmatrix} w_k - w^* \\ w_{k-1} - w^* \end{bmatrix}$$

Let $w_k^{(1)}, w_k^{(2)}$ be the first and second coordinate of $w_k$ respectively, $A_k^{(i,j)}$ is the element in $(i, j)$-position of $A$. Since $w^* = \mathbf{0}$, the above update can be written as:

$$\begin{bmatrix} w_{k+1}^{(1)} \\ w_{k+1}^{(2)} \\ w_k^{(1)} \\ w_k^{(2)} \end{bmatrix} = \begin{bmatrix} 1+\beta - \alpha A_k^{(1,1)} & 0 & -\beta & 0 \\ 0 & 1+\beta - \alpha A_k^{(2,2)} & 0 & -\beta \\ 1 & 0 & 0 & 0 \\ 0 & 1 & 0 & 0 \end{bmatrix} \begin{bmatrix} w_k^{(1)} \\ w_k^{(2)} \\ w_{k-1}^{(1)} \\ w_{k-1}^{(2)} \end{bmatrix}$$

Hence, we can separate the two coordinates and interpret the update as SHB in 1 dimension for each coordinate.

Subsequently, we only focus on the second coordinate which corresponds to $L$ ($A_k^{(2,2)}$) in matrix $A$.

$$w_{k+1} = w_k - \alpha A_k^{22} w_k + \beta(w_k - w_{k-1})$$

$$\implies \begin{pmatrix} w_{k+1} \\ w_k \end{pmatrix} = \begin{pmatrix} 1+\beta - \frac{2A_k^{22}}{L} & -\beta \\ 1 & 0 \end{pmatrix} \begin{pmatrix} w_k \\ w_{k-1} \end{pmatrix} \qquad (\alpha = 1/\bar{L} = 2/L)$$

Denoting $\Delta_k := \begin{pmatrix} w_k \\ w_{k-1} \end{pmatrix}$, the above update is

$$\Delta_{k+1} = H_k \Delta_k$$

where $H_k$ is either $H_1 := \begin{pmatrix} 1+\beta & -\beta \\ 1 & 0 \end{pmatrix}$ (corresponding to $A_1^{22} = 0$) or $H_2 := \begin{pmatrix} -1+\beta & -\beta \\ 1 & 0 \end{pmatrix}$ (corresponding to $A_2^{22} = L$) with probability 0.5.

In order to prove divergence, we will analyze three iterations of the update in expectation. We enumerate across 8 possible sequences (depending on which sample is chosen): $(1, 1, 1), (1, 1, 2) \ldots (2, 2, 2)$. For example, if the sequence is $(1, 1, 2)$, the corresponding update (across 3 iterations) is:

$$\Delta_{k+3} = H_{(1,1,2)} \Delta_k \text{ where } H_{(1,1,2)} := H_2 H_1 H_1$$

We denote $\mathcal{H}_i$ to be the matrix corresponding to the $i$-th permutation. For example, $\mathcal{H}_1 := H_{(1,1,1)}$. Next, we analyze the suboptimality $\|\Delta_k\|^2$ in expectation.

$$\mathbb{E}[\|\Delta_{k+3}\|^2] = \frac{1}{8} \sum_{i=1}^{8} \|\mathcal{H}_i \Delta_k\|^2 \qquad \text{(probability for each of the 8 sequences is } 1/8)$$

Representing $\Delta_k$ in polar coordinates, for a $\theta_k \in [0, 2\pi]$, $\Delta_k := r_k \phi_k$ where $r_k \in \mathbb{R}_+$ and $\phi_k = \begin{pmatrix} \sin(\theta_k) \\ \cos(\theta_k) \end{pmatrix}$,

$$\mathbb{E}[\|\Delta_{k+3}\|^2] = \frac{1}{8} \sum_{i=1}^{8} \|\mathcal{H}_i r_k \phi_k\|^2$$

$$= \frac{r_k^2}{8} \sum_{i=1}^{8} \|\mathcal{H}_i \phi_k\|^2$$

In order to analyze the divergence of SHB, we define the *norm square increase factor* $\Psi := \frac{\mathbb{E}[\|\Delta_{k+3}\|^2]}{\|\Delta_k\|^2}$,

$$
\begin{aligned}
\Psi &= \frac{\mathbb{E}[\|\Delta_{k+3}\|^2]}{\|\Delta_k\|^2} \\
&= \frac{\frac{r_k^2}{8} \sum_{i=1}^8 \|\mathcal{H}_i \phi_k\|^2}{r_k^2 \|\phi_k\|^2} && (\|\Delta_k\|^2 = \|r_k \phi_k\|^2 = r_k^2 \|\phi_k\|^2) \\
&= \frac{1}{8} \frac{\sum_{i=1}^8 \|\mathcal{H}_i \phi_k\|^2}{\|\phi_k\|^2} \\
&= \frac{1}{8} \sum_{i=1}^8 \|\mathcal{H}_i \phi_k\|^2 && (\|\phi_k\|^2 = 1)
\end{aligned}
$$

$\Psi$ depends on $\phi_k$ and hence it is a function of $\theta_k$. Using symbolic mathematics programming (Meurer et al., 2017), we can calculate $\Psi$ as an expression of $\beta, \theta$,

$$
\begin{aligned}
\Psi = &\frac{1}{8} \sum_{i=1}^8 \left\| \mathcal{H}_i \begin{pmatrix} \sin(\theta) \\ \cos(\theta) \end{pmatrix} \right\|^2 \\
= &- \beta^6 \sin(2\theta) + \beta^6 + 3\beta^5 \sin(2\theta) + \beta^5 \cos(2\theta) - 3\beta^5 \\
&- 5\beta^4 \sin(2\theta) - 2\beta^4 \cos(2\theta) + 6\beta^4 \\
&+ 2\beta^3 \sin(2\theta) + 3\beta^3 \cos(2\theta) - 3\beta^3 \\
&- 2\beta^2 \sin(2\theta) - 3\beta^2 \cos(2\theta) + 5\beta^2 - \cos(2\theta) + 1
\end{aligned}
$$

We first verify that $\Psi(\beta, \theta)$ is monotonically increasing w.r.t $\beta \in [0, 1]$ by taking derivative of $\Psi(\beta, \theta)$ w.r.t $\beta$. We plot the derivative for $\beta \in [0, 1]$ and $\theta \in [0, 2\pi]$. From Fig. 5a, we can see that the derivative of $\Psi(\beta, \theta)$ is positive for $\beta \in [0, 1]$ and $\theta \in [0, 2\pi]$.

Choosing $\beta = 0.63$ (corresponding to $\kappa = 6$), we plot $\Psi$ against $\theta$ in Fig. 5b and minimize $\Psi$ w.r.t $\theta$, finding the minimum to be 1.1. Since $\min(\Psi) = 1.1 > 1$, the sub-optimality is increasing in expectation for any $\Delta_k$ when $\beta = 0.63$. Hence, since $\Psi(\beta, \theta)$ is monotonically increasing with respect to $\beta$ (Fig. 5a), when $\kappa > 6$ (correspond to $\beta > 0.63$), for an arbitrary $\Delta_k$,

$$
\mathbb{E}[\|\Delta_{k+3}\|^2] > c\, \mathbb{E}[\|\Delta_k\|^2]
$$

where $c \geq 1.1$ for all $\kappa > 6$. Unrolling the recursion starting from 0 to $3T$,

$$
\mathbb{E}[\|\Delta_{3T}\|^2] > c^T \|\Delta_0\|^2
$$

Since $c \geq 1.1 > 1$, the second coordinate will diverge and SHB will diverge consequently (Fig. 5c). $\qquad\square$

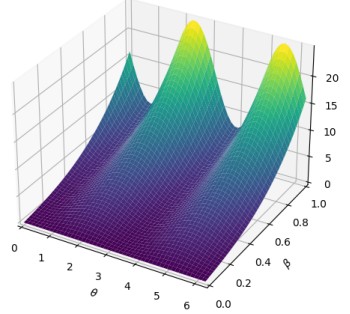
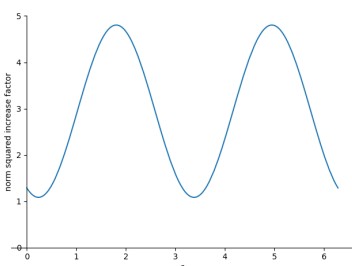
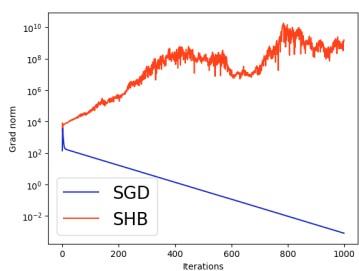

(a) 3D plot of derivative of $\Psi(\beta, \theta)$ with respect to $\beta$ for $\beta \in [0, 1]$ and $\theta \in [0, 2\pi]$. The whole plane is above 0 hence $\Psi(\beta, \theta)$ is monotonically increasing for $\beta \in [0, 1]$ for any $\theta$.

(b) Plot of $\Psi$ against $\theta$ for $\beta = 0.63$

(c) Plot of SHB vs SGD for the 2-sample case with $b = 1$, $\kappa = 6$. SHB diverges while SGD converges

Figure 5: Figures for 2-sample SHB lower bound proofs

**Theorem 3.** For a $\bar{L}$-smooth, $\bar{\mu}$ strongly-convex quadratic problem $f(w) := \frac{1}{n} \sum_{i=1}^{n} \frac{1}{2} w^T A_i w$ with $n$ samples and dimension $d = n = 100$ such that $w^* = 0$ and each $A_i$ is an $n$-by-$n$ matrix of all zeros except at the $(i, i)$ position, we run SHB (1) with $\alpha_k = \alpha = \frac{1}{L}$, $\beta_k = \beta = \left(1 - \frac{1}{2}\sqrt{\alpha\bar{\mu}}\right)^2$. If $b < \frac{1}{1 + \frac{n-1}{e^{3.3\kappa^{0.6}}}} n$ and $\Delta_k := \begin{pmatrix} w_k \\ w_{k-1} \end{pmatrix}$, for a $c > 1$, after $6T$ iterations, we have that:

$$\mathbb{E}\left[\|\Delta_{6T}\|^2\right] > c^T \|\Delta_0\|^2 .$$

*Proof.* Denote $L = \max_{i \in n} A_i^{(i,i)}$ and $\mu = \min_{i \in n} A_i^{(i,i)}$. For the strongly-convex quadratic objective function $f(w) := \frac{1}{n} \sum_{i=1}^{n} \frac{1}{2} w^T A_i w$, $w^* = \vec{0}$.

Since each $A_i$ is diagonal, similar to Theorem 9, we can separate the coordinates and consider SHB in 1 dimension for each of the coordinates. Subsequently, we only focus on coordinate $u$ that corresponds to the largest $A_i^{(i,i)}$ i.e. $u = \arg\max_{i \in n} A_i^{(i,i)}$. The update for this coordinate is given by:

$$w_{k+1} = w_k - \alpha \nabla f_{ik}(w_k) + \beta(w_k - w_{k-1}),$$

where $\nabla f_{ik}(w_k) = \frac{1}{b} \sum_{i \in B_k} \nabla f_i(w_k) = \frac{1}{b} \left(\sum_{i \in B_k} A_i^{(u,u)}\right) w_k$. Hence,

$$w_{k+1} = w_k - \alpha \frac{1}{b} \left(\sum_b A_{ik}^{(u,u)}\right) w_k + \beta(w_k - w_{k-1})$$

Similar to Theorem 9, we calculate the smoothness of $f(w)$ as $\bar{L} = \lambda_{\max}\left(\nabla^2 f(w)\right) = \lambda_{\max}\left(\frac{1}{n} \sum_{i=1}^{n} A_i\right) = \frac{L}{n}$. Hence $\alpha = \frac{1}{\bar{L}} = \frac{n}{L}$ and the full update can be written as:

$$w_{k+1} = w_k - \frac{n}{bL} \left(\sum_{i \in B_k} A_i^{(u,u)}\right) w_k + \beta(w_k - w_{k-1})$$

$$\implies \begin{pmatrix} w_{k+1} \\ w_k \end{pmatrix} = \begin{pmatrix} 1 + \beta - \frac{n}{b} \frac{\sum_{i \in B_k} A_i^{(u,u)}}{L} & -\beta \\ 1 & 0 \end{pmatrix} \begin{pmatrix} w_k \\ w_{k-1} \end{pmatrix}$$

In each iteration, we randomly sample (without replacement) $b$ examples. Hence, the probability that $A_u$ is in the batch is $\frac{b}{n}$. When $A_u$ is in the batch, $\sum_{i \in B_k} A_i^{(u,u)} = L$. On the other hand, when $A_u$ is not in the

batch, $\sum_{i \in B_k} A_i^{(u,u)} = 0$. Similar to Theorem 9, we define $\Delta_k := \begin{pmatrix} w_k \\ w_{k-1} \end{pmatrix}$. Hence, the update can be rewritten as:

$$\Delta_{k+1} = H_k \Delta_k$$

where $H_k$ is either $H_1 := \begin{pmatrix} 1+\beta & -\beta \\ 1 & 0 \end{pmatrix}$ w.p $\rho_1 := \frac{n-b}{n}$ (corresponding to when $A_u$ is not in the batch) or $H_2 := \begin{pmatrix} 1 - \frac{n}{b} + \beta & -\beta \\ 1 & 0 \end{pmatrix}$ w.p $\rho_2 := 1 - \rho_1 = \frac{b}{n}$ (corresponding to when $A_u$ is in the batch).

We will use the same technique as in Theorem 9 and analyze six iterations of the update in expectation using symbolic mathematics programming (Meurer et al., 2017). For this, we denote $\mathcal{H}_i$ to be the matrix corresponding to the $i$-th permutation of $2^6$ possible sequences and $\bar{\rho}_i$ to be the probability of that sequence. Therefore, $\bar{\rho}_i$ is a product of $\rho_1, \rho_2$ corresponding to matrices $H_1, H_2$ in the $i$-th sequence. For example, when $\mathcal{H}_1 = H_{(1,1,1,1,1,1)} = H_1 H_1 H_1 H_1 H_1 H_1$, $\bar{\rho}_1 = \rho_1^6$. Writing the suboptimality $\|\Delta_k\|^2$ in expectation,

$$\mathbb{E} \|\Delta_{k+6}\|^2 = \sum_{i=1}^{2^6} \bar{\rho}_i \|\mathcal{H}_i \Delta_k\|^2$$

Representing $\Delta_k$ in polar coordinates, for a $\theta_k \in [0, 2\pi]$, $\Delta_k := r_k \phi_k$ where $r_k \in \mathbb{R}_+$ and $\phi_k = \begin{pmatrix} \sin(\theta_k) \\ \cos(\theta_k) \end{pmatrix}$.

The norm square increase factor $\Psi := \frac{\mathbb{E}[\|\Delta_{k+6}\|^2]}{\mathbb{E}[\|\Delta_k\|^2]}$ is given by:

$$\begin{aligned} \Psi &= \frac{\mathbb{E} \|\Delta_{k+6}\|^2}{\|\Delta_k\|^2} \\ &= \frac{r_k^2 \sum_{i=1}^{2^6} \bar{\rho}_i \|\mathcal{H}_i \phi_k\|^2}{r_k^2 \|\phi_k\|^2} \\ &= \sum_{i=1}^{2^6} \bar{\rho}_i \|\mathcal{H}_i \phi_k\|^2 \end{aligned}$$

Using symbolic mathematics programming, we write $\Psi$ as a function of $b, \beta, \theta$ (see Fig. 6 for the complete expression) and analyze $\Psi(b, \beta, \theta)$. Similar to Theorem 9, we first show that $\Psi(b, \beta, \theta)$ is monotonically increasing w.r.t $\beta$. Using the expression of $\Psi'_\beta(b, \beta, \theta) = \frac{\partial \Psi(b, \beta, \theta)}{\partial \beta}$, for each $b \in [n-1]$, we plot $\Psi'_\beta(b, \beta, \theta)$ for $\beta \in [0.25, 1)$, $\theta \in [0, 2\pi]$ and observe that $\Psi'_\beta$ is positive. In Fig. 7a, we show an example plot of $\Psi'_\beta(b, \beta, \theta)$ when $b = 70$. Furthermore, we discretize $\beta$ and $\theta$ to numerically verify that for any $b \in [n-1]$, $\Psi'_\beta(b, \beta, \theta)$ is greater than 0. In Table 1, we show an example for values of $\Psi'_\beta(b, \beta, \theta)$ when $b = 70$. Hence for every $b \in [n-1]$, $\Psi(b, \beta, \theta)$ is a monotonically increasing function in $\beta$.

Next, for each batch-size $b \in [n-1]$, we minimize $\Psi(b, \beta, \theta)$ and find $\beta^*(b)$ as the smallest $\beta$ such that $\Psi(b, \beta, \theta) > 1$. In Fig. 7b, when $b = 70$, we plot minimum of $\Psi(b, \beta, \theta)$ w.r.t $\theta$ and show the corresponding $\beta^*(b)$. Since $\Psi(b, \beta, \theta)$ is monotonically increasing w.r.t $\beta$, we conclude that for a fixed batch-size $b \in [n-1]$, $\forall \theta \in [0, 2\pi]$, $\forall \beta \in (\beta^*(b), 1)$, $\Psi(b, \beta, \theta) > 1$.

From the definition of $\beta$, we can calculate the corresponding $\kappa$ for any $\beta \in [0.25, 1)$ as $\kappa = \left( \frac{1}{2(1-\sqrt{\beta})} \right)^2$.

Hence, for a fixed batch-size $b$, the coordinate $u$ (and hence SHB) will diverge if $\kappa > \kappa^*(b)$ (corresponding to $\beta^*(b)$).

From Theorem 2, we see that the *batch factor* equal to $\frac{n-b}{(n-1)b}$ must be sufficiently small to ensure convergence of SHB. In particular, SHB converges at an accelerated rate if $\frac{n-b}{(n-1)b} \leq \frac{1}{C\kappa^2}$. Hence, in order to derive the lower-bound, we plot $\log \left( \frac{n-b}{(n-1)b} \right)$ against $\log(\kappa^*(b))$ in Fig. 7c. We observe that for larger $\kappa^*(b)$, the batch factor is smaller. In other words, when $\kappa$ is large, SHB requires a larger batch-size to avoid divergence.

$$-\beta^{12}\sin(2\theta)+\beta^{12}+9\beta^{11}\sin(2\theta)+\beta^{11}\cos(2\theta)-9\beta^{11}-25\beta^{10}\sin(2\theta)-7\beta^{10}\cos(2\theta)+25\beta^{10}+19\beta^{9}\sin(2\theta)+12\beta^{9}\cos(2\theta)-18\beta^{9}+$$
$$7\beta^{8}\sin(2\theta)+2\beta^{8}\cos(2\theta)-12\beta^{8}-\beta^{7}\sin(2\theta)-7\beta^{7}\cos(2\theta)+5\beta^{7}-4\beta^{6}\sin(2\theta)-6\beta^{6}\cos(2\theta)+8\beta^{6}-2\beta^{5}\sin(2\theta)+2\beta^{5}+\beta^{4}\cos(2\theta)-$$
$$\beta^{4}+\beta^{3}\cos(2\theta)-\beta^{3}-\frac{500\beta^{10}\sin(2\theta)}{b}-\frac{50\beta^{10}\cos(2\theta)}{b}+\frac{550\beta^{10}}{b}+\frac{2800\beta^{9}\sin(2\theta)}{b}+\frac{800\beta^{9}\cos(2\theta)}{b}-\frac{3200\beta^{9}}{b}-\frac{3500\beta^{8}\sin(2\theta)}{b}-$$
$$\frac{2350\beta^{8}\cos(2\theta)}{b}+\frac{4350\beta^{8}}{b}-\frac{1300\beta^{7}\sin(2\theta)}{b}+\frac{400\beta^{7}\cos(2\theta)}{b}+\frac{1600\beta^{7}}{b}+\frac{1400\beta^{6}\sin(2\theta)}{b}+\frac{2350\beta^{6}\cos(2\theta)}{b}-\frac{2850\beta^{6}}{b}+$$
$$\frac{1300\beta^{5}\sin(2\theta)}{b}+\frac{800\beta^{5}\cos(2\theta)}{b}-\frac{2000\beta^{5}}{b}+\frac{100\beta^{4}\sin(2\theta)}{b}-\frac{550\beta^{4}\cos(2\theta)}{b}+\frac{350\beta^{4}}{b}-\frac{100\beta^{3}\sin(2\theta)}{b}-\frac{800\beta^{3}\cos(2\theta)}{b}+\frac{800\beta^{3}}{b}-$$
$$\frac{100\beta^{2}\sin(2\theta)}{b}-\frac{50\beta^{2}\cos(2\theta)}{b}+\frac{150\beta^{2}}{b}+\frac{50\cos(2\theta)}{b}-\frac{50}{b}-\frac{100000\beta^{8}\sin(2\theta)}{b^{2}}-\frac{25000\beta^{8}\cos(2\theta)}{b^{2}}+\frac{125000\beta^{8}}{b^{2}}+\frac{300000\beta^{7}\sin(2\theta)}{b^{2}}-$$
$$\frac{180000\beta^{7}\cos(2\theta)}{b^{2}}-\frac{420000\beta^{7}}{b^{2}}-\frac{30000\beta^{6}\sin(2\theta)}{b^{2}}-\frac{170000\beta^{6}\cos(2\theta)}{b^{2}}+\frac{110000\beta^{6}}{b^{2}}-\frac{230000\beta^{5}\sin(2\theta)}{b^{2}}+\frac{240000\beta^{5}\cos(2\theta)}{b^{2}}+$$
$$\frac{420000\beta^{5}}{b^{2}}-\frac{70000\beta^{4}\sin(2\theta)}{b^{2}}+\frac{45000\beta^{4}\cos(2\theta)}{b^{2}}+\frac{45000\beta^{4}}{b^{2}}+\frac{30000\beta^{3}\sin(2\theta)}{b^{2}}+\frac{180000\beta^{3}\cos(2\theta)}{b^{2}}-\frac{180000\beta^{3}}{b^{2}}+$$
$$\frac{40000\sqrt{2}\beta^{2}\sin\left(2\theta+\frac{\pi}{4}\right)}{b^{2}}-\frac{80000\beta^{2}}{b^{2}}-\frac{25000\cos(2\theta)}{b^{2}}+\frac{25000}{b^{2}}-\frac{10000000\beta^{6}\sin(2\theta)}{b^{3}}-\frac{5000000\beta^{6}\cos(2\theta)}{b^{3}}+\frac{15000000\beta^{6}}{b^{3}}+$$
$$\frac{12000000\beta^{5}\sin(2\theta)}{b^{3}}+\frac{16000000\beta^{5}\cos(2\theta)}{b^{3}}-\frac{24000000\beta^{5}}{b^{3}}+\frac{11000000\beta^{4}\sin(2\theta)}{b^{3}}+\frac{5000000\beta^{4}\cos(2\theta)}{b^{3}}-\frac{17000000\beta^{4}}{b^{3}}-$$
$$\frac{3000000\beta^{3}\sin(2\theta)}{b^{3}}-\frac{16000000\beta^{3}\cos(2\theta)}{b^{3}}+\frac{16000000\beta^{3}}{b^{3}}-\frac{6000000\beta^{2}\sin(2\theta)}{b^{3}}-\frac{9000000\beta^{2}\cos(2\theta)}{b^{3}}+\frac{15000000\beta^{2}}{b^{3}}+\frac{5000000\cos(2\theta)}{b^{3}}-$$
$$\frac{5000000}{b^{3}}-\frac{500000000\sqrt{2}\beta^{4}\sin\left(2\theta+\frac{\pi}{4}\right)}{b^{4}}+\frac{1000000000\beta^{4}}{b^{4}}+\frac{1000000000\beta^{3}\sin(2\theta)}{b^{4}}+\frac{500000000\beta^{3}\cos(2\theta)}{b^{4}}-\frac{500000000\beta^{3}}{b^{4}}+$$
$$\frac{400000000\beta^{2}\sin(2\theta)}{b^{4}}+\frac{800000000\beta^{2}\cos(2\theta)}{b^{4}}-\frac{1200000000\beta^{2}}{b^{4}}-\frac{500000000\cos(2\theta)}{b^{4}}+\frac{500000000}{b^{4}}-\frac{10000000000\beta^{2}\sin(2\theta)}{b^{5}}-$$
$$\frac{25000000000\beta^{2}\cos(2\theta)}{b^{5}}+\frac{35000000000\beta^{2}}{b^{5}}+\frac{25000000000\cos(2\theta)}{b^{5}}-\frac{25000000000}{b^{5}}-\frac{500000000000\cos(2\theta)}{b^{6}}+\frac{500000000000}{b^{6}}$$

Figure 6: $\Psi$ as a function of $b, \beta, \theta$

In order to quantify the relationship between $\kappa^*(b)$ and $b$, we calculate the best-fit line for our plot in Fig. 7c using linear regression. The slope corresponding to the best fit line is $-0.6$ and the y-intercept is $-3.8$. Hence, we can conclude that,

$$\log\left(\frac{n-b}{(n-1)b}\right) < -0.6\log(\kappa^*(b)) - 3.3$$

$$\implies \frac{n-b}{(n-1)b} < \frac{e^{-3.3}}{(\kappa^*(b))^{0.6}}$$

$$\implies (\kappa^*(b))^{0.6} > \frac{(n-1)b}{(n-b)e^{3.3}}$$

Previously, we have shown that $\Psi(b, \kappa, \theta) > 1$ for all $\kappa > \kappa^*(b)$. Hence, $\Psi(b, \kappa, \theta) > 1$ when

$$\kappa^{0.6} > (\kappa^*(b))^{0.6}$$

$$\implies \kappa^{0.6} > \frac{(n-1)b}{(n-b)e^{4}}$$

$$\implies \frac{n-b}{(n-1)b} > \frac{1}{e^{3.3}\kappa^{0.6}}$$

$$\implies \frac{b}{n} < \frac{1}{1+\frac{n-1}{e^{3.3}\kappa^{0.6}}}$$

$$\implies b < \frac{1}{1+\frac{n-1}{e^{3.3}\kappa^{0.6}}}n$$

Therefore, when the batch-size $b < \frac{1}{1+\frac{n-1}{e^{3.3}(\kappa)^{0.6}}}n = \Omega(\kappa^{0.6})$, the norm square increase factor $\Psi(b, \kappa, \theta)$ will be greater than 1 which leads to divergence. For an arbitrary $\Delta_k$,

$$\mathbb{E}\left\|\Delta_{k+6}\right\|^2 > c\,\mathbb{E}\left\|\Delta_k\right\|^2$$

$$\implies \mathbb{E}\left\|\Delta_{6T}\right\|^2 > c^T\left\|\Delta_0\right\|^2$$

Since $c > 1$, SHB will consequently diverge. In Fig. 7d, we plot the gradient norm of SHB for $\kappa = 10$ and observe that when $b = 10 < \frac{1}{1+\frac{n-1}{e^{3.3}(\kappa)^{0.6}}}n$, SHB diverges empirically verifying our lower-bound. $\qquad\square$

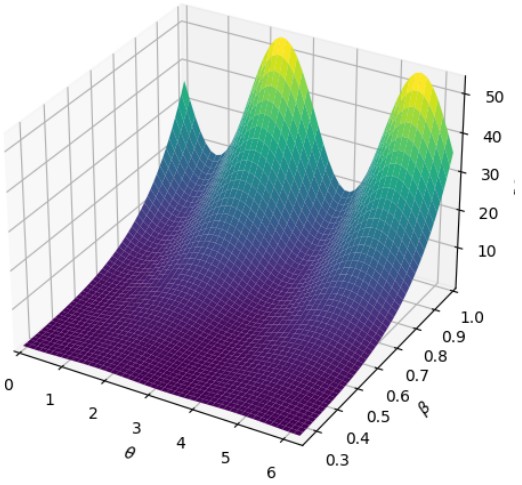

(a) 3D plot of derivative of $\Psi(b, \beta, \theta)$ with respect to $\beta$ for a sample $b = 70$, $\beta \in [0.25, 1)$ and $\theta \in [0, 2\pi]$. The whole plane is above 0 hence $\Psi(70, \beta, \theta)$ is monotonically increasing for $\beta \in [0, 1]$ for any $\theta$.

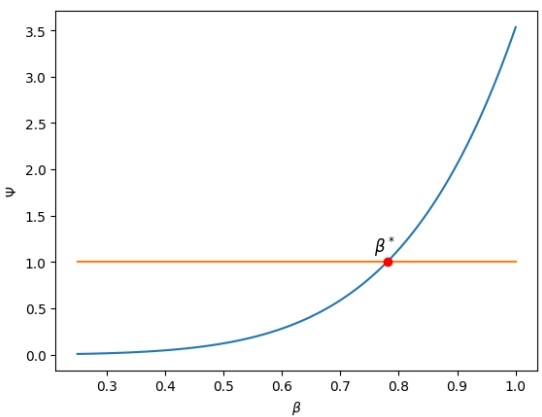

(b) Plot of minimum $\Psi(b, \beta, \theta)$ with respect to $\theta$ for a sample $b = 70$, $\beta \in [0.25, 1)$ and $\theta \in [0, 2\pi]$. $\beta^*(b)$ is smallest $\beta$ such that $\Psi > 1$.

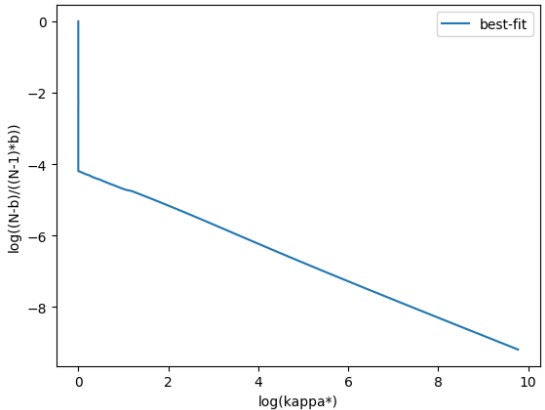

(c) Plot of log batch factor $\log\left(\frac{n-b}{(n-1)b}\right)$ against $\log(\kappa^*)$. Using linear regression, the slope of the best fit line is $-0.6$ and the y-intercept is $-3.8$

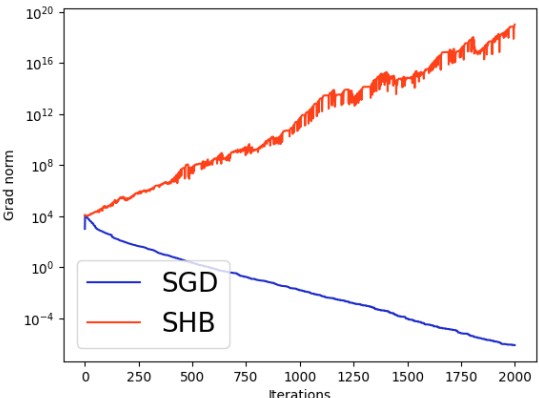

(d) Plot of SHB vs SGD for the $n$-sample case with $\kappa = 10$, $n = 100$, $b = 10 < \frac{1}{1 + \frac{n-1}{e^{3.3}(\kappa)^{0.6}}}n$. SHB diverges while SGD converges

Figure 7: Figures for $n$-sample SHB lower bound proofs

| $\theta$ | | | | | $\beta$ | | | | | |
|---|---|---|---|---|---|---|---|---|---|---|
| | 0.250 | 0.333 | 0.416 | 0.500 | 0.583 | 0.666 | 0.749 | 0.833 | 0.915 | 0.999 |
| 0 | 0.120 | 0.299 | 0.667 | 1.364 | 2.608 | 4.719 | 8.168 | 13.643 | 22.137 | 35.083 |
| $0.2\pi$ | 0.203 | 0.387 | 0.701 | 1.216 | 2.037 | 3.311 | 5.238 | 8.102 | 12.283 | 18.281 |
| $0.4\pi$ | 0.490 | 0.891 | 1.532 | 2.515 | 3.970 | 6.055 | 8.962 | 12.911 | 18.151 | 24.968 |
| $0.6\pi$ | 0.584 | 1.114 | 2.012 | 3.466 | 5.734 | 9.160 | 14.193 | 21.423 | 31.632 | 45.902 |
| $0.8\pi$ | 0.354 | 0.748 | 1.477 | 2.754 | 4.892 | 8.333 | 13.703 | 21.875 | 34.094 | 52.153 |
| $1.0\pi$ | 0.120 | 0.299 | 0.667 | 1.364 | 2.608 | 4.719 | 8.168 | 13.643 | 22.137 | 35.082 |
| $1.2\pi$ | 0.203 | 0.387 | 0.701 | 1.216 | 2.037 | 3.311 | 5.238 | 8.102 | 12.283 | 18.281 |
| $1.4\pi$ | 0.490 | 0.891 | 1.532 | 2.515 | 3.970 | 6.055 | 8.962 | 12.911 | 18.151 | 24.968 |
| $1.6\pi$ | 0.584 | 1.114 | 2.012 | 3.466 | 5.734 | 9.160 | 14.193 | 21.423 | 31.632 | 45.902 |
| $1.8\pi$ | 0.354 | 0.748 | 1.477 | 2.754 | 4.892 | 8.333 | 13.703 | 21.875 | 34.094 | 52.153 |
| $2\pi$ | 0.120 | 0.299 | 0.667 | 1.364 | 2.608 | 4.719 | 8.168 | 13.643 | 22.137 | 35.083 |

Table 1: Values of $\Psi_{\beta}^{'}(b, \beta, \theta)$ when $b = 70$ for different $\beta \in [0.25, 1)$ and $\theta \in [0, 2\pi]$

## F   Proofs for multi-stage SHB

**Theorem 4.** For $L$-smooth, $\mu$ strongly-convex quadratics with $\kappa > 1$, for $T \geq \bar{T} := \frac{3 \cdot 2^8 \sqrt{\kappa}}{\ln(2)} \max \left\{ 4\kappa, e^2 \right\}$,

Algorithm 1 with $b \geq b^* := n \max \left\{ \frac{1}{1 + \frac{n-1}{C\kappa^2}}, \frac{1}{1 + \frac{(n-1)a_I}{3}} \right\}$ converges as:

$$\mathbb{E} \left\| w_T - w^* \right\| \leq C_7 \exp \left( -\frac{T}{8\sqrt{\kappa}} \right) \left\| w_0 - w^* \right\| + C_8 \frac{\chi}{\sqrt{T}}$$

where $C := 3^5 2^6$ and $C_7, C_8$ are polynomial in $\kappa$ and poly-logarithmic in $T$.

*Proof.* Stage zero consists of $T_0 = \frac{T}{2}$ iterations with $\alpha = \frac{1}{L}$ and $\beta = \left( 1 - \frac{1}{2\sqrt{\kappa}} \right)^2$. Let $T_i$ be the last iteration in stage $i$, $T_I = T$. Using the result of Theorem 2 with $a = 1$ for $T_0$ iterations in stage zero and defining $\Delta_t := w_t - w^*$,

$$\mathbb{E} \left\| w_T - w^* \right\| \leq \frac{6\sqrt{2}}{\sqrt{a}} \sqrt{\kappa} \exp \left( -\frac{\sqrt{a}}{4} \frac{T}{\sqrt{\kappa}} \right) \left\| w_0 - w^* \right\| + \frac{12\sqrt{a}\chi}{\mu} \min \left\{ 1, \frac{\zeta}{\sqrt{a}} \right\}$$

$$\mathbb{E} \left\| \Delta_{T_0} \right\| \leq 6\sqrt{2}\sqrt{\kappa} \exp \left( -\frac{T}{8\sqrt{\kappa}} \right) \left\| w_0 - w^* \right\| + \frac{12\chi}{\mu} \min \left\{ 1, \frac{\zeta}{\sqrt{a}} \right\}$$

$$\leq 6\sqrt{2}\sqrt{\kappa} \exp \left( -\frac{T}{8\sqrt{\kappa}} \right) \left\| w_0 - w^* \right\| + \frac{12\chi}{\mu}$$

We split the remaining $\frac{T}{2}$ iterations into $I$ stages. For stage $i \in [1, I]$, we set $\alpha_i = \frac{a_i}{L}$ and choose $a_i = 2^{-i}$. Using Theorem 2 for stage $i$,

$$\mathbb{E} \left\| \Delta_{T_i} \right\| \leq 6\sqrt{2} \sqrt{\frac{\kappa}{a_i}} \exp \left( -\frac{\sqrt{a_i}}{4} \frac{T_i}{\sqrt{\kappa}} \right) \mathbb{E} \left\| \Delta_{T_i-1} \right\| + \frac{12\sqrt{a_i}\chi}{\mu} \min \left\{ 1, \frac{\zeta}{\sqrt{a_i}} \right\}$$

$$\leq 6\sqrt{2} \, 2^{i/2} \sqrt{\kappa} \exp \left( -\frac{1}{4 \, 2^{i/2}} \frac{T_i}{\sqrt{\kappa}} \right) \mathbb{E} \left\| \Delta_{T_i-1} \right\| + \frac{12\chi}{\mu \, 2^{i/2}}$$

$$\leq \exp \left( (i/2 + 5) \ln(2) + \ln(\sqrt{\kappa}) - \frac{T_i}{2^{i/2} 4 \sqrt{\kappa}} \right) \mathbb{E} \left\| \Delta_{T_i-1} \right\| + \frac{12\chi}{\mu \, 2^{i/2}}$$

Now we want to find $T_i$ such that $(i/2 + 5) \ln(2) + \ln(\sqrt{\kappa}) - \frac{T_i}{2^{i/2} 4 \sqrt{\kappa}} \leq -\frac{T_i}{2^{(i+1)/2} 4 \sqrt{\kappa}}$.

$$\implies T_i \geq \frac{4}{(2 - \sqrt{2})} 2^{i/2} \sqrt{\kappa} ((i/2 + 5) \ln(2) + \ln(\sqrt{\kappa}))$$

$$\implies T_i = \left\lceil \frac{4}{(2 - \sqrt{2})} 2^{i/2} \sqrt{\kappa} ((i/2 + 5) \ln(2) + \ln(\sqrt{\kappa})) \right\rceil$$

$$\implies \frac{T_i}{2^{(i+1)/2} 4 \sqrt{\kappa}} \geq \frac{1}{\sqrt{2} - 1} ((i/2 + 5) \ln(2) + \ln(\sqrt{\kappa})) \geq 2 \ln(2)(i/2 + 5) + 2 \ln(\sqrt{\kappa}) \geq (i/2 + 5) + \ln(\kappa)$$

$$\implies \exp \left( -\frac{T_i}{2^{(i+1)/2} 4 \sqrt{\kappa}} \right) \leq \frac{1}{\kappa} \exp(-(i/2 + 5))$$

Define $\rho_i := \frac{1}{\kappa} \exp(-(i/2 + 5))$. If we unroll the above for $I$ stages we have:

$$\mathbb{E}[\|\Delta_{T_I}\|] \leq \prod_{i=1}^{I} \rho_i \mathbb{E} \left\| \Delta_{T_0} \right\| + \frac{12\chi}{\mu} \sum_{i=1}^{I} 2^{-i/2} \prod_{j=i+1}^{I} \rho_j$$

$$= \exp \left( -\sum_{i=1}^{I} (i/2 + 5) - I \ln \kappa \right) \mathbb{E} \left\| \Delta_{T_0} \right\| + \frac{12\chi}{\mu} \sum_{i=1}^{I} 2^{-i/2} \exp \left( -\sum_{j=i+1}^{I} (j/2 + 5) - i \ln \kappa \right)$$

$$\leq \exp\left(-I^2/4 - I\ln\kappa\right)\mathbb{E}\left\|\Delta_{T_0}\right\| + \frac{12\chi}{\mu}\sum_{i=1}^{I}2^{-i/2}\exp\left(-\sum_{j=i+1}^{I}(j/2) - i\ln\kappa\right)$$

$$\leq \exp\left(-I^2/4 - I\ln\kappa\right)\mathbb{E}\left\|\Delta_{T_0}\right\| + \frac{12\chi}{\mu}\sum_{i=1}^{I}2^{-i/2}\exp\left(-\frac{(I-i)(I+i+1)}{4} - i\ln\kappa\right)$$

$$\leq \exp\left(-I^2/4 - I\ln\kappa\right)\mathbb{E}\left\|\Delta_{T_0}\right\| + \frac{12\chi}{\mu}\sum_{i=1}^{I}2^{-i/2}\,2^{\left(-\frac{(I^2-i^2+I-i)}{4}\right)}\exp\left(-i\ln\kappa\right) \qquad \text{(since } 2 \leq e)$$

$$= \exp\left(-I^2/4 - I\ln\kappa\right)\mathbb{E}\left\|\Delta_{T_0}\right\| + \frac{12\chi}{\mu}\sum_{i=1}^{I}2^{\left(-\frac{I}{4}\right)}\exp\left(-i\ln\kappa\right) \qquad \text{(since } I^2 \geq i^2)$$

$$\leq \exp\left(-I^2/4 - I\ln\kappa\right)\mathbb{E}\left\|\Delta_{T_0}\right\| + \frac{12\,\chi\kappa}{\mu(\kappa-1)}\frac{1}{2^{\left(\frac{I}{4}\right)}} \qquad \text{(Simplifying } \sum\frac{1}{\kappa^i})$$

$$\leq \exp\left(-I^2/4\right)\mathbb{E}\left\|\Delta_{T_0}\right\| + \frac{12\,\chi\kappa}{\mu(\kappa-1)}\frac{1}{2^{\left(\frac{I}{4}\right)}}$$

Putting together the convergence from stage 0 and stages $[1, I]$,

$$\mathbb{E}[\|\Delta_T\|] \leq \exp\left(-I^2/4\right)\left(6\sqrt{2}\sqrt{\kappa}\,\exp\left(-\frac{T}{8\sqrt{\kappa}}\right)\|w_0 - w^*\| + \frac{12\chi}{\mu}\right) + \frac{12\,\chi\kappa}{\mu(\kappa-1)}\frac{1}{2^{\left(\frac{I}{4}\right)}}$$

$$\leq \frac{1}{2^{\left(\frac{I}{4}\right)}}\left(6\sqrt{2}\sqrt{\kappa}\,\exp\left(-\frac{T}{8\sqrt{\kappa}}\right)\|w_0 - w^*\| + \frac{12\chi}{\mu}\right) + \frac{12\,\chi\kappa}{\mu(\kappa-1)}\frac{1}{2^{\left(\frac{I}{4}\right)}}$$

$$\leq \frac{1}{2^{\left(\frac{I}{4}\right)}}\left(6\sqrt{2}\sqrt{\kappa}\,\exp\left(-\frac{T}{8\sqrt{\kappa}}\right)\|w_0 - w^*\|\right) + \frac{24\,\chi\kappa}{\mu(\kappa-1)}\frac{1}{2^{\left(\frac{I}{4}\right)}} \qquad (5)$$

Now we need to bound the number of iterations in $\sum_{i=1}^{I}T_i$.

$$\sum_{i=1}^{I}T_i \leq \sum_{i=1}^{I}\left[\frac{4}{(2-\sqrt{2})}2^{i/2}\sqrt{\kappa}((i/2+5)\ln(2) + \ln(\sqrt{\kappa})) + 1\right]$$

$$\leq 8\sum_{i=1}^{I}2^{i/2}\sqrt{\kappa}((i/2+5)\ln(2) + \ln(\sqrt{\kappa})) + I$$

$$\leq 8\sqrt{\kappa}\sum_{i=1}^{I}2^{i/2}\left\{4i + \ln(\sqrt{\kappa})\right\} + I \qquad \text{(For } i \geq 1)$$

$$\leq 8\sqrt{\kappa}\left\{4I + \ln(\sqrt{\kappa})\right\}\sum_{i=1}^{I}2^{i/2} + I$$

$$\leq 8\sqrt{\kappa}\left\{4I + \ln(\sqrt{\kappa})\right\}\frac{2^{(I+1)/2}}{\sqrt{2}-1} + I$$

$$\leq 16\sqrt{\kappa}\left[5I + \ln(\sqrt{\kappa})\right]2^{(I+1)/2}$$

Assume that $I \geq \ln(\sqrt{\kappa})$. In this case,

$$\sum_{i=1}^{I}T_i \leq 192\sqrt{\kappa}\,I\,2^{(I/2)} \qquad (6)$$

We need to set $I$ s.t. the upper-bound on the total number of iterations in the $I$ stages is smaller than the available budget on the iterations which is equal to $T/2$. Hence,

$$\frac{T}{2} \geq \frac{192}{\ln(\sqrt{2})}\sqrt{\kappa}\,\exp\left(\ln(\sqrt{2})\,I\right)\left(I\ln(\sqrt{2})\right)$$

$$\implies \exp\left(\ln(\sqrt{2})\, I\right)\left(I\ln(\sqrt{2})\right) \le \frac{T\ln(\sqrt{2})}{384\sqrt{\kappa}} \implies I\ln(\sqrt{2}) \le \mathcal{W}\left(\frac{T\ln(\sqrt{2})}{384\sqrt{\kappa}}\right)$$

$$\text{(where } \mathcal{W} \text{ is the Lambert function)}$$

Hence, it suffices to set $I = \left\lfloor \frac{1}{\ln(\sqrt{2})}\mathcal{W}\left(\frac{T\ln(\sqrt{2})}{384\sqrt{\kappa}}\right)\right\rfloor$. We know that,

$$\frac{I}{2} \ge \frac{\frac{1}{\ln(\sqrt{2})}\mathcal{W}\left(\frac{T\ln(\sqrt{2})}{384\sqrt{\kappa}}\right) - 1}{2}$$

$$\implies \exp\left(\frac{I}{2}\right) \ge \sqrt{1/e}\left(\exp\left(\mathcal{W}\left(\frac{T\ln(\sqrt{2})}{384\sqrt{\kappa}}\right)\right)\right)^{1/(2\ln(\sqrt{2}))}$$

$$= \sqrt{1/e}\left(\frac{\frac{T\ln(\sqrt{2})}{384\sqrt{\kappa}}}{\mathcal{W}\left(\frac{T\ln(\sqrt{2})}{384\sqrt{\kappa}}\right)}\right)^{1/(2\ln(\sqrt{2}))} \qquad \text{(since } \exp(\mathcal{W}(x)) = \tfrac{x}{\mathcal{W}(x)})$$

For $x \ge e^2$, $W(x) \le \sqrt{1 + 2\log^2(x)}$. Assuming $T \ge \frac{384\sqrt{\kappa}}{\ln(\sqrt{2})}e^2$ so $\frac{T\ln(\sqrt{2})}{384\sqrt{\kappa}} \ge e^2$,

$$\exp\left(\frac{I}{2}\right) \ge \sqrt{1/e}\left(\frac{\frac{T\ln(\sqrt{2})}{384\sqrt{\kappa}}}{\sqrt{1+2\log^2\left(\frac{T\ln(\sqrt{2})}{384\sqrt{\kappa}}\right)}}\right)^{1/\ln(2)}$$

Since $2^x = (\exp(x))^{\ln(2)}$,

$$2^{\left(\frac{I}{2}\right)} = (\exp(I/2))^{\ln(2)} \ge (\sqrt{1/e})^{\ln(2)}\left(\frac{\frac{T\ln(\sqrt{2})}{384\sqrt{\kappa}}}{\sqrt{1+2\log^2\left(\frac{T\ln(\sqrt{2})}{384\sqrt{\kappa}}\right)}}\right)^{\ln(2)/\ln(2)}$$

$$= (\exp(I/2))^{\ln(2)} \ge (\sqrt{1/e})^{\ln(2)}\left(\frac{\frac{T\ln(\sqrt{2})}{384\sqrt{\kappa}}}{\sqrt{1+2\log^2\left(\frac{T\ln(\sqrt{2})}{384\sqrt{\kappa}}\right)}}\right)$$

$$\implies \frac{1}{2^{I/2}} \le 2\frac{\sqrt{1+2\log^2\left(\frac{T\ln(\sqrt{2})}{384\sqrt{\kappa}}\right)}\,384\sqrt{\kappa}}{T\ln(\sqrt{2})}$$

$$= \frac{2^9\,3\sqrt{\kappa\left(1+2\log^2\left(\frac{T\ln(\sqrt{2})}{384\sqrt{\kappa}}\right)\right)}}{\ln(2)}\frac{1}{T}$$

Define $C_1 := \frac{2^9\,3\sqrt{\kappa\left(1+2\log^2\left(\frac{T\ln(\sqrt{2})}{384\sqrt{\kappa}}\right)\right)}}{\ln(2)}$, meaning that $\frac{1}{2^{I/2}} \le \frac{C_1}{T}$. Using the overall convergence rate,

$$\mathbb{E}[\|w_T - w^*\|] \le \frac{1}{2^{\left(\frac{I}{4}\right)}}\left(6\sqrt{2}\sqrt{\kappa}\exp\left(-\frac{T}{8\sqrt{\kappa}}\right)\|w_0 - w^*\|\right) + \frac{24\chi\kappa}{\mu(\kappa-1)}\frac{1}{2^{\left(\frac{I}{4}\right)}}$$

$$\le \frac{\sqrt{C_1}}{\sqrt{T}}\left(6\sqrt{2}\sqrt{\kappa}\exp\left(-\frac{T}{8\sqrt{\kappa}}\right)\|w_0 - w^*\|\right) + \frac{24\chi\kappa}{\mu(\kappa-1)}\frac{\sqrt{C_1}}{\sqrt{T}}$$

$$\implies \mathbb{E}\|w_T - w^*\| \le 6\sqrt{2}\sqrt{C_1}\sqrt{\kappa}\frac{1}{\sqrt{T}}\exp\left(-\frac{T}{8\sqrt{\kappa}}\right)\|w_0 - w^*\| + \frac{24\chi\kappa}{\mu(\kappa-1)}\frac{\sqrt{C_1}}{\sqrt{T}}$$

We assumed that $I \geq \ln(\sqrt{\kappa})$ meaning that we want $T$ s.t.

$$\left\lfloor \frac{1}{\ln(\sqrt{2})} \mathcal{W}\left(\frac{T\ln(\sqrt{2})}{384\sqrt{\kappa}}\right)\right\rfloor \geq \ln(\sqrt{\kappa})$$

Since $\mathcal{W}(x) \geq \log\left(\frac{\sqrt{4x+1}+1}{2}\right)$ for $x > 0$ we need to have:

$$\frac{1}{\ln(\sqrt{2})} \mathcal{W}\left(\frac{T\ln(\sqrt{2})}{384\sqrt{\kappa}}\right) - 1 \geq \ln\sqrt{\kappa}$$

$$\implies \log\left(\frac{\sqrt{4\frac{T\ln(\sqrt{2})}{384\sqrt{\kappa}}+1}+1}{2}\right) \geq \ln(\sqrt{\kappa}) + 1$$

$$\implies T \geq \frac{384\sqrt{\kappa}}{4\ln(\sqrt{2})}\left((2^{\ln(\sqrt{\kappa})+2} - 2)^2 - 1\right)$$

Hence, it suffices to choose

$$\implies T > \frac{96\sqrt{\kappa}}{\ln(\sqrt{2})} 16 \cdot 2^{\ln(\kappa)}$$

Since $e > 2$

$$\implies T > \frac{96\sqrt{\kappa}}{\ln(\sqrt{2})} 16 \cdot e^{\ln(\kappa)} = \frac{3 \cdot 2^9 \kappa\sqrt{\kappa}}{\ln(\sqrt{2})}$$

Therefore to satisfy all the assumptions we need that

$$T \geq \max\left\{\frac{3 \cdot 2^9 \kappa\sqrt{\kappa}}{\ln(\sqrt{2})}, \frac{384\sqrt{\kappa}}{\ln(\sqrt{2})}e^2\right\}$$

$$= \max\left\{\frac{3 \cdot 2^{10} \kappa\sqrt{\kappa}}{\ln(2)}, \frac{3 \cdot 2^8 e^2\sqrt{\kappa}}{\ln(2)}\right\}$$

Let $C_3 = \frac{3 \cdot 2^8 \max\{4\kappa, e^2\}}{\ln(2)}$ then

$$T \geq \sqrt{\kappa}C_3$$

With this constraint then $\frac{1}{\sqrt{T}} \leq \frac{1}{\kappa^{1/4}\sqrt{C_3}}$. Hence the final convergence rate can be presented as

$$\mathbb{E}\|w_T - w^*\| \leq 6\sqrt{2}\sqrt{C_1}\sqrt{\kappa} \frac{1}{\kappa^{1/4}\sqrt{C_3}} \exp\left(-\frac{T}{8\sqrt{\kappa}}\right)\|w_0 - w^*\| + \frac{24\chi\kappa}{\mu(\kappa-1)}\frac{\sqrt{C_1}}{\sqrt{T}}$$

$$= 6\sqrt{2}\sqrt{\frac{C_1}{C_3}}\kappa^{1/4}\exp\left(-\frac{T}{8\sqrt{\kappa}}\right)\|w_0 - w^*\| + \frac{24\chi\kappa}{\mu(\kappa-1)}\frac{\sqrt{C_1}}{\sqrt{T}}$$

$$\leq 6\left(1 + 2\log^2\left(\frac{T\ln(\sqrt{2})}{384\sqrt{\kappa}}\right)\right)^{\frac{1}{4}}\exp\left(-\frac{T}{8\sqrt{\kappa}}\right)\|w_0 - w^*\| + \frac{\chi}{\sqrt{T}}\frac{24\kappa\sqrt{C_1}}{\mu(\kappa-1)}$$

Let $C_7(\kappa, T) := 6\left(1 + 2\log^2\left(\frac{T\ln(\sqrt{2})}{384\sqrt{\kappa}}\right)\right)^{\frac{1}{4}}$ and $C_8(\kappa, T) := \frac{24\kappa\sqrt{C_1}}{\mu(\kappa-1)}$ then

$$\mathbb{E}\|w_T - w^*\| \leq C_7 \exp\left(-\frac{T}{8\sqrt{\kappa}}\right)\|w_0 - w^*\| + C_8\frac{\chi}{\sqrt{T}}$$

$\square$

**Corollary 2.** For $L$-smooth, $\mu$ strongly-convex quadratics with $\kappa > 1$, Algorithm 1 with batch-size $b$ such that $b \geq b^* := n \frac{1}{1+\frac{n-1}{C\kappa^2}}$ attains the same rate as in Theorem 4 for $T \in \left[ \frac{3 \cdot 2^8 \sqrt{\kappa}}{\ln(2)} \max\left\{4\kappa, e^2\right\}, C_1 \sqrt{\frac{(n-1)b}{3(n-b)}} \right]$, where $C := 3^5 2^6$ and $C_1$ is defined in the proof of Theorem 4 in App. F.

*Proof.* By the batch-size constraint in Theorem 4, we need $\frac{b}{n} \geq \frac{1}{1+\frac{(n-1)a_I}{3}}$. From Theorem 4, $a_I = 2^{-I} \leq \frac{C_1^2}{T^2}$ hence

$$\frac{b}{n} \geq \frac{1}{1+\frac{n-1}{3(T/C_1)^2}}$$

$$\implies 1 + \frac{n-1}{3\left(T/C_1\right)^2} \leq \frac{n}{b}$$

$$\implies 3\left(T/C_1\right)^2 \left(1 - \frac{n}{b}\right) \leq 1 - n$$

$$\implies \left(T/C_1\right)^2 \leq \frac{(n-1)b}{3(n-b)}$$

$$\implies T \leq C_1 \sqrt{\frac{(n-1)b}{3(n-b)}}$$

$\square$

# G   Proofs for SHB with bounded noise assumption

**Theorem 10.** For $L$-smooth and $\mu$ strongly-convex quadratics, SHB (Eq. (1)) with $\alpha_k = \alpha = \frac{a}{L}$ and $a \leq 1$, $\beta_k = \beta = \left(1 - \frac{1}{2}\sqrt{\alpha\mu}\right)^2$, bounded gradient noise $\mathbb{E}_i \|\nabla f(w_k) - \nabla f_i(w_k)\|^2 \leq \tilde{\sigma}^2$, and batch-size $b$ has the following convergence rate,

$$\mathbb{E}\|w_T - w^*\| \leq 3\sqrt{2}\sqrt{\frac{\kappa}{a}} \exp\left(-\frac{\sqrt{a}}{2}\frac{T}{\sqrt{\kappa}}\right) \|w_0 - w^*\| + \frac{6\zeta\tilde{\sigma}}{\mu},$$

*Proof.* With the definition of SHB (1), if $\nabla f_{ik}(w)$ is the mini-batch gradient at iteration $k$, then, for quadratics,

$$\underbrace{\begin{bmatrix} w_{k+1} - w^* \\ w_k - w^* \end{bmatrix}}_{\Delta_{k+1}} = \underbrace{\begin{bmatrix} (1+\beta)I_d - \alpha A & -\beta I_d \\ I_d & 0 \end{bmatrix}}_{\mathcal{H}} \underbrace{\begin{bmatrix} w_k - w^* \\ w_{k-1} - w^* \end{bmatrix}}_{\Delta_k} + \alpha \underbrace{\begin{bmatrix} \nabla f(w_k) - \nabla f_{ik}(w_k) \\ 0 \end{bmatrix}}_{\delta_k}$$

$$\Delta_{k+1} = \mathcal{H}\Delta_k + \alpha\delta_k$$

Recurring from $k = 0$ to $T - 1$, taking norm and expectation w.r.t to the randomness in all iterations.

$$\mathbb{E}[\|\Delta_T\|] \leq \|\mathcal{H}^T\Delta_0\| + \alpha\mathbb{E}\left[\left\|\sum_{k=0}^{T-1} \mathcal{H}^{T-1-k}\delta_k\right\|\right]$$

In order to simplify $\delta_k$, we will use the result from Lemma 7 and Lohr (2021),

$$\mathbb{E}_k[\|\delta_k\|^2] = \mathbb{E}_k[\|\nabla f(w_k) - \nabla f_{ik}(w_k)\|^2]$$

$$= \frac{n-b}{n\,b}\mathbb{E}_i\|\nabla f(w_k) - \nabla f_i(w_k)\|^2$$

(Sampling with replacement where $b$ is the batch-size and $n$ is the total number of examples)

$$\leq \frac{n-b}{n\,b}\tilde{\sigma}^2$$

$$\implies \mathbb{E}_k[\|\delta_k\|] \le \sqrt{\frac{n-b}{n\,b}}\,\tilde{\sigma}$$
$$= \zeta\tilde{\sigma}$$

Using Theorem 8 and Corollary 6 (Wang et al., 2021), for any vector $v$, $\left\|\mathcal{H}^k v\right\| \le C_0\,\rho^k\,\|v\|$ where $\rho = \sqrt{\beta}$. Hence,

$$
\begin{aligned}
\mathbb{E}[\|\Delta_T\|] &\le C_0\,\rho^T\,\|\Delta_0\| + \frac{C_0\,a}{L}\left[\sum_{k=0}^{T-1}\rho^{T-1-k}\,\mathbb{E}\,\|\delta_k\|\right] && (\alpha = \tfrac{a}{L}) \\
&= C_0\,\rho^T\,\|\Delta_0\| + \frac{C_0\,a}{L}\left[\sum_{k=0}^{T-1}\rho^{T-1-k}\,\zeta\tilde{\sigma}\right] \\
&\le C_0\,\rho^T\,\|\Delta_0\| + \frac{C_0\,a\zeta\tilde{\sigma}}{L}\left[\sum_{k=0}^{T-1}\rho^k\right] \\
&= C_0\,\rho^T\,\|\Delta_0\| + \frac{C_0\,a\zeta\tilde{\sigma}}{L}\frac{1-\rho^T}{1-\rho} \\
&\le C_0\,\rho^T\,\|\Delta_0\| + \frac{C_0\,a\zeta\tilde{\sigma}}{(1-\rho)L}
\end{aligned}
$$

We have $C_0 \le 3\sqrt{\frac{\kappa}{a}}$, $\rho = 1 - \frac{\sqrt{a}}{2\sqrt{\kappa}}$ then,

$$
\begin{aligned}
E[\|\Delta_T\|] &\le 3\sqrt{\frac{\kappa}{a}}\left(1 - \frac{\sqrt{a}}{2\sqrt{\kappa}}\right)^T\|\Delta_0\| + 3\sqrt{\frac{\kappa}{a}}\frac{2\sqrt{\kappa}}{\sqrt{a}}\frac{a\zeta\tilde{\sigma}}{L} \\
&\le 3\sqrt{\frac{\kappa}{a}}\left(1 - \frac{\sqrt{a}}{2\sqrt{\kappa}}\right)^T\|\Delta_0\| + 6\frac{\zeta\tilde{\sigma}}{\mu} \\
\implies \mathbb{E}\,\|w_T - w^*\| &\le 3\sqrt{2}\sqrt{\frac{\kappa}{a}}\exp\left(-\frac{\sqrt{a}}{2}\frac{T}{\sqrt{\kappa}}\right)\|w_0 - w^*\| + 6\frac{\zeta\tilde{\sigma}}{\mu}
\end{aligned}
$$

$\square$

Since the proof of Theorem 4 analyzes each stage independently and successively uses Theorem 2, we can repeat the proof of Theorem 4 using Theorem 10, yielding a similar $O\left(\exp\left(-\frac{T}{\sqrt{\kappa}}\right) + \frac{\tilde{\sigma}}{T}\right)$ rate without any $\kappa$ dependence on the batch-size.

# H    Proofs for two-phase SHB

**Theorem 11.** For $L$-smooth, $\mu$ strongly-convex quadratics with $\kappa > 4$, Algorithm 2 with batch-size $b$ such that $b \geq b^* = n \frac{1}{1 + \frac{n-1}{C \kappa^2}}$ results in the following convergence,

$$
\mathbb{E} \|w_{T-1} - w^*\| \leq 6\sqrt{\frac{2c_2\kappa}{c_L}} \exp\left(-\frac{T}{\kappa^q} \frac{\gamma}{8\ln((1-c)\,T)}\right) \|w_0 - w^*\|
$$
$$
+ \frac{12\sqrt{6Lc_2}\,\zeta\sigma}{\sqrt{c_L}\mu} \exp\left(-\frac{(1-c)T\,\gamma}{8\kappa\ln(T)}\right) + \frac{8\sqrt{Lc_2}\,\ln(T)\zeta\kappa^{3/2}}{e\,\gamma\sqrt{c_L}} \frac{\sigma}{\sqrt{(1-c)T}}
$$

where $q = 1 - \frac{\ln(c\sqrt{\kappa}+1-c)}{\ln(\kappa)}$, $\zeta = \sqrt{\frac{n-b}{(n-1)b}}$ captures the dependence on the batch-size, $c_2 = \exp\left(\frac{1}{\kappa\ln((1-c)T)}\right)$, $c_L = \frac{4(1-\gamma)}{\mu^2}\left[1 - \exp\left(-\frac{\mu\,\gamma}{2L}\right)\right]$ and $C := 3^5 2^6$.

*Proof.* After $T_0$ iterations, by Theorem 2 with $a = 1$, and $\chi = \sqrt{\mathbb{E}\|\nabla f_i(w^*)\|^2}$,

$$
\mathbb{E}\|w_{T_0} - w^*\| \leq 6\sqrt{2}\sqrt{\kappa}\,\exp\left(-\frac{T_0}{4\sqrt{\kappa}}\right)\|w_0 - w^*\| + \frac{12\sqrt{3}\zeta\chi}{\mu} \qquad \text{(since in Theorem 2, } \zeta = \sqrt{3\frac{n-b}{(n-1)b}})
$$

After $T_1$ iterations, by Theorem 1 with $\gamma = (1/T_1)^{1/T_1}$ and $\tau = 1$,

$$
\mathbb{E}[\mathcal{E}_T] \leq \|w_{T_0} - w^*\|^2 c_2 \exp\left(-\frac{T_1\,\gamma}{4\kappa\ln(T_1)}\right) + \frac{64L\sigma^2 c_2\zeta^2\kappa^3}{e^2\,\gamma^2}\frac{(\ln(T_1))^2}{T_1}
$$

Taking square-root on both sides and using that $\sqrt{\mathbb{E}[\mathcal{E}_T]} \geq \mathbb{E}[\sqrt{\mathcal{E}_T}]$

$$
\mathbb{E}[\sqrt{\mathcal{E}_T}] \leq \|w_{T_0} - w^*\|\sqrt{c_2}\exp\left(-\frac{T_1\,\gamma}{8\kappa\ln(T_1)}\right) + \frac{8\sqrt{Lc_2}\,\sigma\zeta\kappa^{3/2}}{e\,\gamma}\frac{(\ln(T_1))}{\sqrt{T_1}}
$$

Taking expectation over the randomness in iterations $t = 0$ to $T_0 - 1$,

$$
\mathbb{E}[\sqrt{\mathcal{E}_T}] \leq \sqrt{c_2}\exp\left(-\frac{T_1\,\gamma}{8\kappa\ln(T_1)}\right)\mathbb{E}\|w_{T_0} - w^*\| + \frac{8\sqrt{Lc_2}\,\sigma\zeta\kappa^{3/2}}{e\,\gamma}\frac{(\ln(T_1))}{\sqrt{T_1}}
$$

Using the above inequality and using that $\chi^2 \leq 2L\,\sigma^2$,

$$
\implies \mathbb{E}[\sqrt{\mathcal{E}_T}] \leq \sqrt{c_2}\exp\left(-\frac{T_1\,\gamma}{8\kappa\ln(T_1)}\right)\left[6\sqrt{2}\sqrt{\kappa}\,\exp\left(-\frac{T_0}{4\sqrt{\kappa}}\right)\|w_0 - w^*\| + \frac{12\sqrt{3}\sqrt{2L}\zeta\sigma}{\mu}\right]
$$
$$
+ \frac{8\sqrt{Lc_2}\,\sigma\zeta\kappa^{3/2}}{e\,\gamma}\frac{(\ln(T_1))}{\sqrt{T_1}}
$$
$$
= 6\sqrt{2}\sqrt{c_2}\sqrt{\kappa}\,\exp\left(-\frac{T_0}{4\sqrt{\kappa}} - \frac{T_1\,\gamma}{8\kappa\ln(T_1)}\right)\|w_0 - w^*\|
$$
$$
+ \frac{12\sqrt{6L}\,\sqrt{c_2}\,\zeta\sigma}{\mu}\exp\left(-\frac{T_1\,\gamma}{8\kappa\ln(T_1)}\right) + \frac{8\sqrt{Lc_2}\,\sigma\zeta\kappa^{3/2}}{e\,\gamma}\frac{(\ln(T_1))}{\sqrt{T_1}}
$$

For $T_1 \geq e$ and since $\gamma \leq 1$

$$
\mathbb{E}[\sqrt{\mathcal{E}_T}] \leq 6\sqrt{2}\sqrt{c_2}\sqrt{\kappa}\,\exp\left(-\frac{\gamma}{8\ln(T_1)}\left(\frac{T_0}{\sqrt{\kappa}} + \frac{T_1}{\kappa}\right)\right)\|w_0 - w^*\|
$$

$$+ \frac{12\sqrt{6L}\sqrt{c_2}\,\zeta\sigma}{\mu}\exp\left(-\frac{T_1\,\gamma}{8\kappa\ln(T_1)}\right) + \frac{8\sqrt{Lc_2}\,\sigma\zeta\kappa^{3/2}}{e\,\gamma}\frac{(\ln(T_1))}{\sqrt{T_1}}$$

Consider term $A := 6\sqrt{2}\sqrt{c_2}\sqrt{\kappa}\exp\left(-\frac{\gamma}{8\ln(T_1)}\left(\frac{T_0}{\sqrt{\kappa}}+\frac{T_1}{\kappa}\right)\right)\|w_0-w^*\|$. We have,

$$\begin{aligned}
\frac{T_0}{\sqrt{\kappa}}+\frac{T_1}{\kappa} &= \frac{cT}{\sqrt{\kappa}}+\frac{(1-c)T}{\kappa}\\
&= T\frac{c\sqrt{\kappa}+1-c}{\kappa}
\end{aligned}$$

Suppose $\frac{T}{\kappa^q} = T\frac{c\sqrt{\kappa}+1-c}{\kappa}$ then

$$\begin{aligned}
\kappa^q &= \frac{\kappa}{c\sqrt{\kappa}+1-c}\\
\implies q &= 1 - \frac{\ln(c\sqrt{\kappa}+1-c)}{\ln(\kappa)}
\end{aligned}$$

Since $c \in (0,1)$, $q \in (0.5,1)$, then

$$A \le 6\sqrt{2c_2\kappa}\exp\left(-\frac{T}{\kappa^q}\frac{\gamma}{8\ln((1-c)\,T)}\right)\|w_0-w^*\|$$

Consider the noise $B := \frac{12\sqrt{6Lc_2}\,\zeta\sigma}{\mu}\exp\left(-\frac{T_1\,\gamma}{8\kappa\ln(T_1)}\right) + \frac{8\sqrt{Lc_2}\,\sigma\zeta\kappa^{3/2}}{e\,\gamma}\frac{(\ln(T_1))}{\sqrt{T_1}}$.

$$\begin{aligned}
B &= \frac{12\sqrt{6Lc_2}\,\zeta\sigma}{\mu}\exp\left(-\frac{T_1\,\gamma}{8\kappa\ln(T_1)}\right) + \frac{8\sqrt{Lc_2}\,\sigma\zeta\kappa^{3/2}}{e\,\gamma}\frac{(\ln(T_1))}{\sqrt{T_1}}\\
&= \frac{12\sqrt{6Lc_2}\,\zeta\sigma}{\mu}\exp\left(-\frac{(1-c)T\,\gamma}{8\kappa\ln((1-c)T)}\right) + \frac{8\sqrt{Lc_2}\,\sigma\zeta\kappa^{3/2}}{e\,\gamma}\frac{(\ln((1-c)T))}{\sqrt{(1-c)T}}\\
&\le \frac{12\sqrt{6Lc_2}\,\zeta\sigma}{\mu}\exp\left(-\frac{(1-c)T\,\gamma}{8\kappa\ln(T)}\right) + \frac{8\sqrt{Lc_2}\,\sigma\zeta\kappa^{3/2}}{e\,\gamma}\frac{\ln(T)}{\sqrt{(1-c)T}} \qquad \text{(Since } c \in (0,1))
\end{aligned}$$

Putting everything together,

$$\begin{aligned}
\mathbb{E}[\sqrt{\mathcal{E}_T}] \le{}& 6\sqrt{2c_2\kappa}\exp\left(-\frac{T}{\kappa^q}\frac{\gamma}{8\ln((1-c)\,T)}\right)\|w_0-w^*\|\\
&+ \frac{12\sqrt{6Lc_2}\,\zeta\sigma}{\mu}\exp\left(-\frac{(1-c)T\,\gamma}{8\kappa\ln(T)}\right) + \frac{8\sqrt{Lc_2}\,\sigma\zeta\kappa^{3/2}}{e\,\gamma}\frac{\ln(T)}{\sqrt{(1-c)T}}
\end{aligned}$$

Using Lemma 1 with $c_L = \frac{4(1-\gamma)}{\mu^2}\left[1-\exp\left(-\frac{\mu\,\gamma}{2L}\right)\right]$ then

$$\begin{aligned}
\sqrt{c_L}\,\mathbb{E}\|w_{T-1}-w^*\| \le{}& 6\sqrt{2c_2\kappa}\exp\left(-\frac{T}{\kappa^q}\frac{\gamma}{8\ln((1-c)\,T)}\right)\|w_0-w^*\|\\
&+ \frac{12\sqrt{6Lc_2}\,\zeta\sigma}{\mu}\exp\left(-\frac{(1-c)T\,\gamma}{8\kappa\ln(T)}\right) + \frac{8\sqrt{Lc_2}\,\sigma\zeta\kappa^{3/2}}{e\,\gamma}\frac{\ln(T)}{\sqrt{(1-c)T}}\\
\implies \mathbb{E}\|w_{T-1}-w^*\| \le{}& 6\sqrt{\frac{2c_2\kappa}{c_L}}\exp\left(-\frac{T}{\kappa^q}\frac{\gamma}{8\ln((1-c)\,T)}\right)\|w_0-w^*\|\\
&+ \frac{12\sqrt{6Lc_2}\,\zeta\sigma}{\sqrt{c_L}\mu}\exp\left(-\frac{(1-c)T\,\gamma}{8\kappa\ln(T)}\right) + \frac{8\sqrt{Lc_2}\,\ln(T)\zeta\kappa^{3/2}}{e\,\gamma\sqrt{c_L}}\frac{\sigma}{\sqrt{(1-c)T}}
\end{aligned}$$

$\square$

**Corollary 3.** For $L$-smooth, $\mu$ strongly-convex quadratics with $\kappa > 4$, Algorithm 2 with batch-size $b$ such that $b \geq b^* = n \frac{1}{1 + \frac{n-1}{C \kappa^2}}$ and $c = \frac{1}{2}$ results in a rate of $O\left(\exp\left(-\frac{T}{\kappa^{0.7}}\right) + \frac{\sigma}{\sqrt{T}}\right)$ for all $T$.

*Proof.* From Theorem 11, $q = 1 - \frac{\ln(c\sqrt{\kappa}+1-c)}{\ln(\kappa)}$, plug in $c = \frac{1}{2}$, then

$$q = 1 - \frac{\ln\left(\frac{\sqrt{\kappa}+1}{2}\right)}{\ln(\kappa)}$$

Since $q$ is monotonically decreasing with respect to $\kappa$ and $\kappa > 4$ then

$$q \leq 1 - \frac{\ln\left(\frac{\sqrt{4}+1}{2}\right)}{\ln(4)} \approx 0.7$$

Hence the bias term in the rate of Theorem 11 converges as $O\left(\exp\left(-\frac{T}{\kappa^q}\right)\right) \leq O\left(\exp\left(-\frac{T}{\kappa^{0.7}}\right)\right)$.

The noise term converges at the rate of $O\left(\frac{\sigma}{\sqrt{T}}\right)$ hence the convergence rate of Algorithm 2 when $c = 0.5$ is $O\left(\exp\left(-\frac{T}{\kappa^{0.7}}\right) + \frac{\sigma}{\sqrt{T}}\right)$. $\square$

# I Additional experiments

## I.1 Quadratics experiments on LIBSVM datasets

To conduct experiments for smooth, strongly-convex functions, we adopt the settings from Vaswani et al. (2022). Our experiment involves the SHB variant and other commonly used optimization methods. The comparison will be based on two common supervised learning losses, squared loss for regression tasks and logistic loss for classification. We will utilize a linear model with $\ell_2$-regularization $\frac{\lambda}{2} \|w\|^2$ in which $\lambda = 0.01$. To assess the performance of the optimization methods, we use *ijcnn* and *rcv1* data sets from LIBSVM (Chang & Lin, 2011). For each dataset, the training iterations will be fixed at $T = 100n$, where $n$ is the number of samples in the training dataset, and we will use a batch-size of 100. To ensure statistical significance, each experiment will be run 5 times independently, and the average result and standard deviation will be plotted. We will use the full gradient norm as the performance measure and plot it against the number of gradient evaluations.

The methods for comparison are: SGD with constant step-sizes (`K-CNST`), SGD with exponentially decreasing step-sizes (Vaswani et al., 2022) (`K-EXP`), SGD with exponentially decreasing step-sizes and SLS (Vaswani et al., 2022; 2021) (`SLS-EXP`), SHB with constant step-sizes (set according to Theorem 5) (`SHB-CNST`), SHB with exponentially decreasing step-sizes (set according to Theorem 1) (`SHB-EXP`), SHB with exponentially decreasing step-sizes (set according to Theorem 1) and SLS (Vaswani et al., 2021) (`SHB-SLS-EXP`).

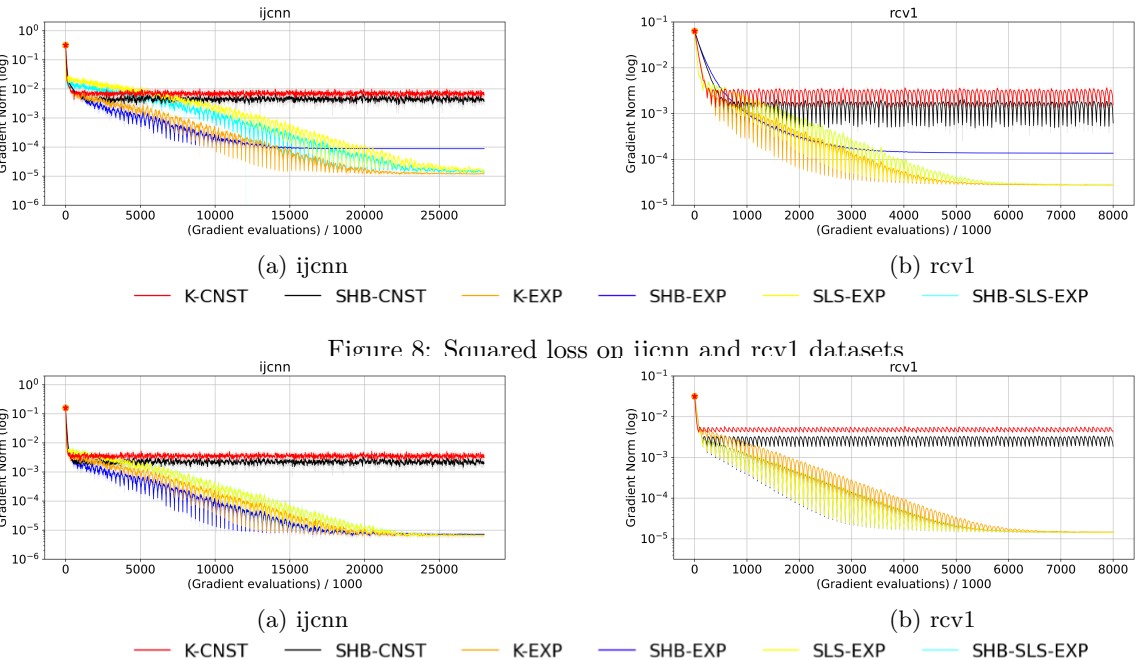

(a) ijcnn

(b) rcv1

Figure 8: Squared loss on ijcnn and rcv1 datasets

(a) ijcnn

(b) rcv1

Figure 9: Logistic loss on ijcnn and rcv1 datasets

From Fig. 8 and Fig. 9, we observe that exponentially decreasing step-sizes for both SHB and SGD have close performance and they both outperform their constant step-sizes variants. We also note that using stochastic line-search by Vaswani et al. (2021), SHB-SLS-EXP matches the performance of the variant with known smoothness.

## I.2 Comparison to Pan et al. (2024) multi-stage SHB

In this section, we consider minimizing smooth, strongly-convex quadratics. The data generation procedure is similar to Section 5. We vary $\kappa \in \{2000, 1000, 500, 200, 100\}$ and the magnitude of the noise $r \in \{10^{-2}, 10^{-4}, 10^{-6}, 10^{-8}\}$. For each dataset, we use a batch-size $b = 0.9n$ to ensure that it is sufficiently large for SHB to achieve an accelerated rate with all of our chosen $\kappa$. We fix the total number of iterations $T = 7000$ and initialization $w_0 = \vec{0}$. For each experiment, we consider 3 independent runs, and plot the

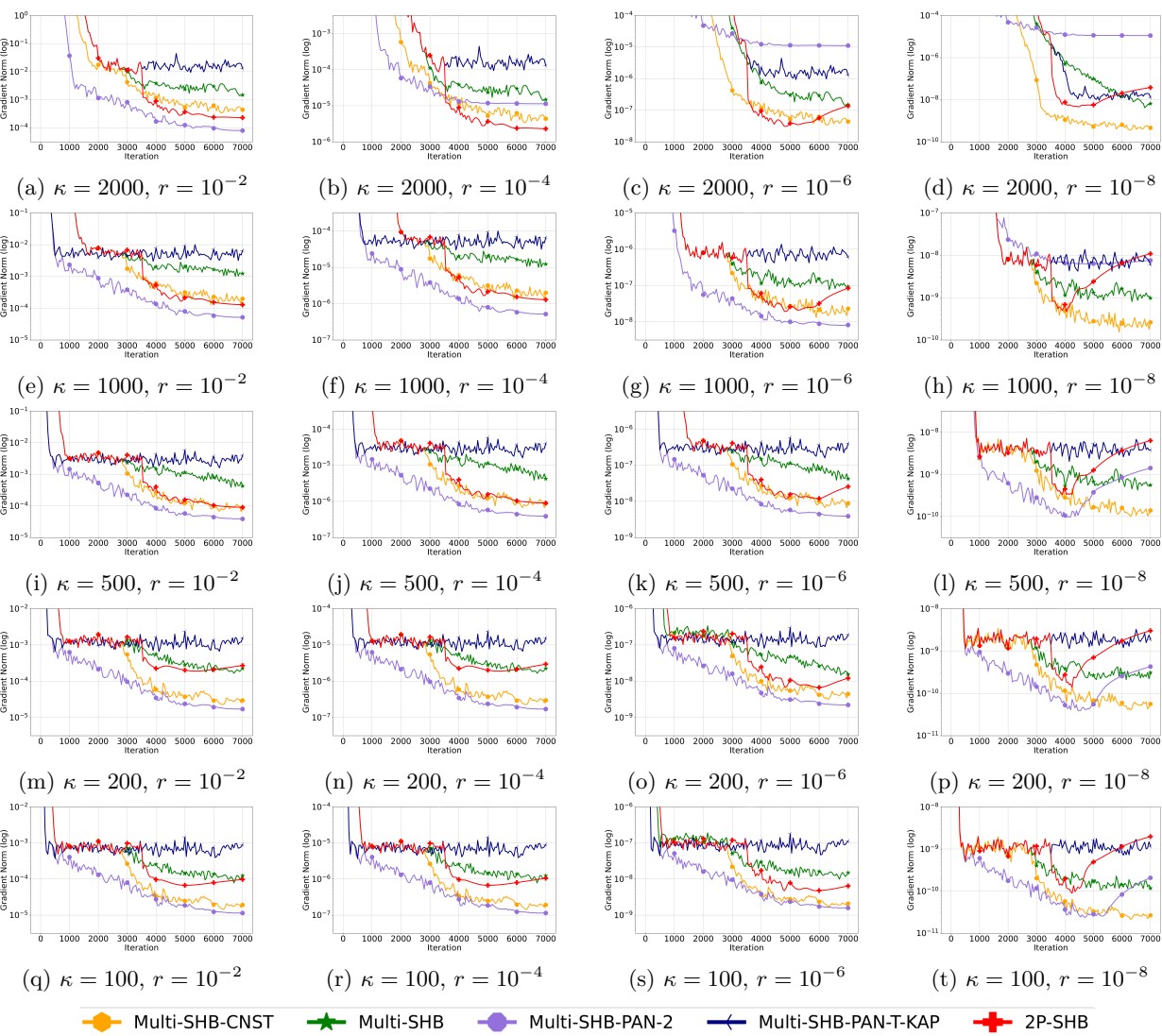

Figure 10: Comparison of `Multi-SHB`, `Multi-SHB-CNST`, `2P-SHB`, `Multi-SHB-PAN-2`, and `Multi-SHB-PAN-T-KAP`. With a sufficiently large batch-size, the method by Pan et al. (2024) is able to avoid the divergence behaviour in Fig. 2. The performance of SHB variants is similar to the method in Pan et al. (2024), and it consistently lies in-between the two extremes.

average result. We will use the full gradient norm as the performance measure and plot it against the number of iterations.

We compare the following methods: Multi-stage SHB (Algorithm 1) (`Multi-SHB`), our heuristic Multi-stage SHB (Algorithm 1) with constant momentum parameter (`Multi-SHB-CNST`), Two-phase SHB (Algorithm 2) with $c = 0.5$ (`2P-SHB`), Multi-stage SHB (Pan et al., 2024) with $C = 2$ (`Multi-SHB-PAN-2`) and $C = T\sqrt{\kappa}$ (`Multi-SHB-PAN-T-KAP`).

We observe that with a sufficiently large batch-size the method by Pan et al. (2024) is able to avoid the divergence behaviour in Fig. 2. Furthermore, the performance of `2P-SHB`, `Multi-SHB`, `Multi-SHB-CNST` is similar to the method in Pan et al. (2024), and it consistently lies in-between the two extremes.

