# OpenReview forum: "(Accelerated) Noise-adaptive Stochastic Heavy-Ball Momentum"
_TMLR — Accepted by TMLR_

### Review · Reviewer_Lpjn · 2024-12-01

**Summary Of Contributions:**

This paper studies the convergence rate of stochastic heavy ball methods (SHB), and provides various results under different settings. The main results are summarized as follows:
1 For general smooth and strongly convex functions, the authors prove a $\mathcal{O}(\exp(- T/\kappa) + \sigma^2/T)$ rate.

2 For smooth and strongly convex quadratic function, the authors prove a $\mathcal{O}(\exp(- T/\sqrt{\kappa}) + \sigma^2)$ accelerated rate for batch size $b>b^*$. Specifically, for interpolation case where $\sigma = 0,$ the Theorem implies a convergence to global optimal.

3 For smooth and strongly convex quadratic function, the authors propose a multi-stage algorithm and provide a  $\mathcal{O}(\exp(- T/\sqrt{\kappa}) + \sigma^2/\sqrt{T})$ accelerated rate. The authors also provide a two-stage algorithm which provides a $\mathcal{O}(\exp(- T/\kappa^{0.7}) + \sigma^2/\sqrt{T}).

**Audience:**

Yes

**Broader Impact Concerns:**

No concerns on the ethical implications of the work.

**Claims And Evidence:**

Yes

**Requested Changes:**

1. While I appreciate the authors' effort to state each theorem precisely, I think many theorems are too "precise" and contain too many notations. Also, sometimes the same parameters are separated into two terms. To name a few:

    - In Theorem 1, $c_2, c_L$ and $\exp( \dots)$ terms are all depend on $T$, could you put all the terms depend on $T$ together, so one can see the dependence on $T$ more clearly?

    - In Theorem 1, I didn't see the utility of $\tau.$ Why not just pick a specific $\tau$?

    - In Theorem  2 and Corollary 2, $\max \{ 3/4, 1-2\sqrt{kappa} \sqrt{\zeta} \}  \geq 3/4,$ so why not directly use $3/4$ in the convergence rate as an upper bound.

    - In Theorem 4, again $C_1, C_3, \kappa^{1/4} \exp(\dots)$ terms both depends on $\kappa$ and $T,$ could you make term depends on $\kappa$ and $T$ together, so one could clearly see the order of $\kappa$ and $T$ in the theorem?

2. For the numerical experiment,

    - In the theorems, your emphasize the for $n \gg \kappa^2,$ only constant batch size guarantees the accelerated convergence. However, in the experiments in Figure 3, you only consider cases where $n \ll \kappa^2$ and pick $b = 0.9n$ as the batch size. I suggest having some experiments showing the convergence in the case of  $n \gg \kappa^2$ and constant $b$  if possible, to strengthen the theoretical results.

    - In both Figure 3 and Figure 4, it is not clear to me how the convergence rate depends on $\kappa.$ Thus, is it possible to make a plot showing the relationship between the convergence rate and $\kappa?$ In particular, under the conditions of Theorem 2 and Theorem 4, is it possible see a $\mathcal(1/\sqrt{\kappa})$ dependence on the convergence rate w.r.t $\kappa?$

3. A few references are out-of-date, for example, Bollapragada et al. (2022) and Lee et al. (2022). Also, There's no Theorem 3.1 in the newest version of Bollapragada et al. (2022). I suggest the authors to double-check the references and cite the latest version if possible.

**Strengths And Weaknesses:**

**Strengths**
1. This paper studies the convergence rate of SHB for smooth and strongly convex functions, and I believe both the problems and results are interesting.
2. I appreciate the authors' effort to state the conditions and implications of each theorem precisely, making the paper more readable.
3. This paper has a rather detailed comparison to related work, which helps the readers to understand the contribution of the paper.

**Weaknesses and Questions**
1. For the results in Theorem 1, it seems that there are two "drawbacks" listed below, could the authors explain more on these two points?

    (1) one need to fix the number of iteration $T,$ and the set the hyperparameters depends on $T.$ This seems to be not common for practical algorithms.

    (2) In the convergence rate, there's a term $\frac{\gamma}{\ln(T/\tau)}$ that also depends on $T$. Moreover, $\lim_{T \rightarrow \infty} \frac{\gamma}{\ln(T/\tau)} = 0,$ since $\lim_{T \rightarrow \infty} \gamma = 1.$ Thus, for large $T, $ it seems to me that the convergence rate is smaller than $\frac{1}{\kappa}.$

2. In Theorem 2, there's a term that depends on the noise is not vanishing in $T.$ While I believe this type  of upper-bound appears in proofs of  stochastic optimization algorithms, I still wonders:

   (1) what is the technical difficulty of obtaining a convergence noise term in $T$ as well as obtaining an accelerated convergence rate? In Theorem 1, you prove a vanishing noise term for general smooth and strongly convex function with non-accelaerated rate.

   (2) The results, in corollary 2 regarding the interpolation case is a bit vague to me. It seems to me that, in practice, it is not common that one can have both strongly convex and interpolation. For example, for linear regression problems, if one considers overparameterized regime, then the problem is not strongly convex; if one considers underparameterized regime or ridge regression, then there's no interpolation in general. Thus could the authors give a practical example for this case?

---

> ### Author Response · Authors · 2025-01-21
>
> We thank the reviewer for their helpful feedback. We have updated the paper (in red) with the following changes requested by the reviewer.
> - For Theorem 1 and Theorem 4, we represent unimportant constants as functions of $\kappa$ and $T$ in the main paper, while providing detailed derivations in the Appendix to help readers focus on the key results.
> - We also included the lines for $\exp\left(\frac{-T}{\sqrt{\kappa}}\right)$ and $\exp\left(\frac{-T}{\kappa}\right)$ in Figure 4 to understand the convergence rate w.r.t $\kappa$. We observe that the larger the batch-size, SHB converges at a rate similar to the $\exp\left(\frac{-T}{\sqrt{\kappa}}\right)$ baseline, while across problems, SGD converges at a rate similar to the $\exp\left(\frac{-T}{\kappa}\right)$ baseline.
> - We have reviewed and updated our references. For Bollapragada et al. (2022), we now refer to the correct theorem in the latest version.
>
> We also address their remaining questions and concerns below.
>
> > Theorem 1, .. need to fix the number of iteration $T$ and the set the hyperparameters depends on $T$
>
> Typically, the number of iterations is fixed before running the algorithm in practice. If $T$ is not known apriori, we can use the "doubling trick'', a standard technique in the literature. Specifically, we first run the method for small $T_0$ iterations (e.g.,$T_0=16$). If continued, the iterations are doubled to $2T_0$ iterations, while adjusting the other hyper-parameters accordingly. This process is repeated iteratively (such that run $j$ of the method requires $2^jT_0$ iterations) until the desired termination criterion is satisfied. We note that "Lu et al, Towards Principled Practical Policy Gradient for Bandits and Tabular MDPs, 2024'' use SGD with exponentially decreasing step-sizes and use the doubling trick to improve the practical performance (see Figure 2).
>
> > Theorem 1,.. for large $T$, it seems that the convergence rate is smaller than $1/\kappa$
>
> The reviewer's understanding is correct. The first term is (slightly) slower than $\exp(-T/\kappa)$.  However, the additional $\gamma/\ln(\frac{T}{\tau})$ term goes to zero a very slow rate. For example, when $\tau=1$, it is roughly equal to $0.1$ even for $T=10^6$. We note that such a dependence also appears in other works that use exponentially decreasing step-sizes [Vaswani, et al 2022, Li et al, 2021].
>
> > In Theorem 1, I didn't see the utility of $\tau$. Why not just pick a specific $\tau$
>
> In practice, we do set $\tau=1$. We kept a general $\tau$ in Theorem 1 because in some cases, setting a specific $\tau$ (for example, one that depends on $\kappa$) can improve the constants in the resulting convergence rate (see Theorem 1 in [Li et al, 2021])
>
> > In Theorem 2, what is the technical difficulty of obtaining a convergence noise term in $T$ as well as obtaining an accelerated convergence rate?
>
> Intuitively, the exponentially decreasing step-sizes in Theorem 1 are too conservative to obtain an accelerated rate. Consequently, in Section 4.3, Theorem 4, we decrease the step-sizes in stages resulting in a multi-stage algorithm that achieves accelerated convergence with a decreasing noise term. We have added this explanation in section 4 in the main paper.
>
> > In practice, it is not common that one can have both strongly convex and interpolation .. could the authors give a practical example for this case?
>
> Linear regression where the targets are realizable by a linear model is an example that satisfies both strong-convexity and interpolation. In particular, consider $y \in \mathbb{R}^n$, $X\in\mathbb{R}^{n\times d}$ and $w^* \in\mathbb{R}^d$ s.t. $d\leq n$ and $y=Xw^*$. $f(w)=\frac{1}{2}||Xw-y||^2$ is strongly-convex with $\mu=\lambda_{\min}[X^T X] > 0$ and satisfies interpolation since $\nabla f_i(w^*)=x_i \left[\langle x_i, w^* \rangle -y_i\right]=0 $ for all $i$.
>
> > In Theorem 2 and Corollary 2, why not directly use 3/4 in the convergence rate as an upper bound.
>
> We keep this notation to demonstrate that different batch-sizes yield different convergence rates under the same condition number $\kappa$ (see Figure 4 for further details).
>
> > In the theorems, your emphasize the for $n \gg \kappa$, only constant batch size guarantees the accelerated convergence. However, in the experiments in Figure 3, you only consider cases where $n \ll \kappa^2$ and pick $0.9n$ as the batch size.
>
> We claim that $b^* = O(\kappa^2)$ is sufficient to guarantee acceleration when $n \gg \kappa^2$. Unfortunately, the constant factor in the big $O$ notation is quite large (equal to $2^6 3^5 = 15552$ according to our theory). In order to see the accelerated rate, we also require $\kappa$ to be relatively large (e.g., in Figure 4, it is difficult to see the difference between the accelerated and non-accelerated rates when $\kappa \leq 32$). Hence, in order to verify this claim, we require $\kappa$ and hence $n \gg C \kappa^2$ to be large. Unfortunately, this is beyond the scope of our current experimental setup.

---

### Review · Reviewer_iV4k · 2024-12-03

**Summary Of Contributions:**

The paper studies the Stochastic Heavy Ball (SHB) method and proves an improved rate for general smooth and strongly convex functions compared to previous works. The rate is obtained using an exponentially decreasing step size; notably, the obtained rate does not need knowledge of the variance.

For smooth, strongly convex quadratic, the authors show that for sufficiently large batch size, SHB with a constant step size enjoys an accelerated rate (up to a neighborhood defined by the noise $\sigma$ that can be made as small as possible through the choice of a parameter $a$), they also show that this need for a large batch size is necessary.

To solve this convergence up to a neighborhood, the authors analyze the multi-stage SHB algorithm and show that by using it they can keep the accelerated rate (always for quadratic) and improve the stochastic rate from $\sigma$ to $\sigma/\sqrt{T}$.

Finally, the authors show that by only using two stages of the previous algorithm, they still get a slightly accelerated rate ($\kappa^{0.7}$ instead of $\kappa^{0.5}$ ) for the bias term and keep the same $\sigma/\sqrt{T}$ stochastic term as before.

**Update:** I have just realized that you have Theorems with the norm squared and Theorems with only the norm, I believe it would be best to stick to one choice.

**Audience:**

Yes

**Claims And Evidence:**

Yes

**Requested Changes:**

I think this is an interesting work, but it needs some slight improvements in terms of writing:

Explain what happens for quadratic functions when you use the same step size schedule as for Th1 and why it prevents having the accelerated bias term.

Explain what is the problem with the multistage SHB that you would consider the two-stage version, is it a problem with the number of stages? I am also curious about what would happen with a three or four-stage version and how can such versions be defined.

You should also discuss the noise term you obtain with the multistage and two-stage versions and compare with your Th2 if you can choose the best $a$ (which seems to give the $\sigma/T$ dependence at the expense of a $T$ factor multiplying the exponential.


Maybe also drop unimportant constants in the main paper and only keep what is important (to keep the focus of the reader on what is really important).

**Update:** You should use one convention (either norm or norm squared ) in your Theorems or mention somewhere that you use both (but it is better to only have one or the other).

And now it seems that based on TH2, the best choice of $a$ can lead to a $\sigma/T^2$ term, can you comment on this?

**Strengths And Weaknesses:**

Strengths:
- The paper has some very nice-looking results.
- Although I did not have time to review the proofs, the results seem logical.
- The definition of the different variance measures that are used are also interesting and more meaningful than what was done in previous works.


Weaknesses:
- The storytelling can be slightly improved: explaining why you need to study the multi-stage SHB (to deal with the problem of convergence to a neighborhood)? Why do you need the 2-stage version of it? why not use the exponentially decreasing step sizes that worked for Th1? For me, all of these things are missing and make it harder to connect the dots.

- There are also some typos: like in the contribution section where the rate for multi-stage SHB is stated with $\sigma /T$ instead of $\sigma /\sqrt{T}$. Also, Algorithm 2 does not specify how to choose $a$.

- The authors also kind of claim that they propose the multistage SHB and at the same time say that previous works already considered it, so which is which? I understood that you analyzed it using a different (more realistic) noise assumption, but you did not invent it. You need to make this clear.

- For quadratic functions the obtained stochastic term $\sigma/\sqrt{T}$ seems to be suboptimal, and the authors neglected to discuss why this is the case.

---

> ### Author Response · Authors · 2025-01-21
>
> We thank the reviewer for their helpful feedback. We have updated the paper (in red) with the following changes requested by the reviewer:
> - For Theorem 1 and Theorem 4, we represent unimportant constants as functions of $\kappa$ and $T$ in the main paper, while providing detailed derivations in the Appendix to help readers focus on the key results.
> - We have added a sentence in Section 4.1 to highlight the difference and connection in the measures of suboptimality between the accelerated and non-accelerated results. In the main paper, we have explicitly clarified the notion of sub-optimality for both sets of results.
>
> We also address their remaining questions and concerns below.
> > Why you need to study the multi-stage SHB? .. Why not use the exponentially decreasing step sizes that worked for Th1?
>
> Intuitively, the exponentially decreasing step-sizes in Theorem 1 are too conservative to obtain an accelerated rate. Consequently, in Section 4.3, Theorem 4, we decrease the step-sizes in stages resulting in a multi-stage algorithm that achieves accelerated convergence with a decreasing noise term. We have added this explanation in Section 4 in the main paper.
>
> > Why do you need the 2-stage version of it? .. what is the problem with the multistage SHB that you would consider the two-stage version, is it a problem with the number of stages?
>
> In the last paragraph of section 4.3, we mention that for a fixed batch-size, the multi-stage SHB algorithm achieves the optimal accelerated convergence rate to the minimizer only for a certain range of $T$. To address this limitation, we propose a simple and practical two-phase SHB algorithm that achieves partially accelerated rates for all values of $T$. Finally, we note that this two-phase algorithm is different from using two stages of our multi-stage algorithm.
>
> > Algorithm 2 does not specify how to choose $a$
>
> We assume that the reviewer is asking about the choice of parameter $c$ (there is no $a$ in Algorithm 2). $c$ can be set to any value in $(0,1)$. For example, in Corollary 3, we analyze the case when $c = \frac{1}{2}$ and in Theorem 11 (Appendix H), we analyze the general case. The optimal value of $c$ depends on the trade-off between the bias and the variance, and requires the knowledge of $\sigma^2$.
>
> > The authors also kind of claim that they propose the multistage SHB and at the same time say that previous works already considered it, so which is which? ..
>
> We noted that previous works utilized the multi-stage approach in conjunction with Nesterov acceleration. While there is concurrent work that uses the multi-stage framework with SHB, the details of the algorithms differ despite both being multi-stage in nature.
>
> > For quadratic functions the obtained stochastic term $\frac{\sigma}{\sqrt{T}}$ seems to be suboptimal ..
>
> As later acknowledged by the reviewer, we are using a different measure of stochasticity (norm vs norm squared) which is the cause for the discrepancy between the results. Given this discrepancy in the choice of sub-optimality, our result is indeed optimal.
>
> > You should also discuss the noise term you obtain with the multistage and two-stage versions and compare with your Th2 if you can choose the best $a$ (which seems to give the dependence at the expense of a factor multiplying the exponential.
>
> In Section 4, we have explained that multi-stage SHB and the two stage variant result in an $O(\exp(-T/\sqrt{\kappa}) + \sigma/\sqrt{T})$ rate. Any choice of $a$ in Theorem 2 results in a worse rate.
>
> > And now it seems that based on TH2, the best choice of can lead to a $O(\sigma/T^2)$ term, can you comment on this?
>
>  We do not understand the reviewer's claim that the best choice of $a$ can result in an $O(\sigma/T^2)$ term. In general, choosing the parameter $a$ involves a trade-off between bias and variance. In particular, if the noise is assumed to be known, the optimal $a$ can be determined (see the result in Corollary 4). Finally, we also note that the variance term (depending on $\sigma^2$) cannot be decreased at a faster rate than $\Omega(\frac{1}{T})$ [Nguyen et al.,2019].

---

> > ### Comment · Reviewer_iV4k · 2025-02-05
> >
> > Sorry for the very late reply:
> >
> > - Algorithm 2 depends on $a$ through the choice of $\alpha$ and $\beta$ from TH2.
> >
> > - In TH2, assuming you know $T$ beforehand, if you choose $a= \mathcal{O}(\frac{log(T)^k}{T^2})$ for some $k$, then it gives $E \|w - w^\star \| = \tilde{\mathcal{O}}(\frac{\sigma}{T})$ for some well-chosen $k$, which in terms of the norm squared gives what I said.

---

> > > ### Author Response · Authors · 2025-02-07
> > >
> > > We thank the reviewer for engaging in a discussion. We provide our answers to the questions and concerns below.
> > >
> > > > Algorithm 2 depends on $a$ through the choice of $\alpha$ and $\beta$ from TH2.
> > >
> > > Yes, the reviewer is correct. In our experiment and the theorem proof in App. H, we choose $a = 1$ to set $\alpha$ and $\beta$ for the first phase of Algorithm 2. We will specify this in the paper.
> > >
> > > > In TH2, assuming you know $T$ beforehand, if you choose $a = O\left( \frac{\log(T)^k}{T^2}\right)$ for some $k$ then it gives $E|w - w^*| = O\left( \frac{1}{T} \right)$ for some well-chosen $k$.
> > >
> > > Indeed, choosing $a = O\left( \frac{\log(T)^k}{T^2}\right)$ in Theorem 2 will result in the second (variance) term having an $O\left( \frac{1}{T} \right)$ dependence. However, the first (bias) term has an $O\left(\frac{1}{\sqrt{a}} \exp(-\sqrt{a} T) \right)$ dependence on $a$. Consequently, choosing $a$ to be $O\left(\frac{1}{T^2}\right)$ will make the first term to be $O(T)$, meaning that the algorithm will diverge. In general, as we mentioned in our previous comment, choosing $a$ involves a trade-off between the bias and the variance.

---

> ### Comment · Reviewer_iV4k · 2025-02-07
>
> Thanks for the answer,
>
> I noticed the term $\frac{1}{\sqrt{a}}$ outside of the exponential, and this is why I said that you need to choose $k$ carefully, but after trying it, it seems we need to take $a \approx \frac{\log(T)^2 k^2}{T^2}$. In general, this leads to an expression of the form $\mathcal{O}\Big(\frac{T}{\log(T) k} \times \frac{1}{T^{k}}\Big)$, so choosing $k=2$ leads to a $\tilde{\mathcal{O}}\Big(\frac{1}{T}\Big)$ in terms of norm (not squared).
>
> Can you comment on this?
>
> The consequence of this is either a better rate is possible (which I highly doubt), or that you have a mistake in TH2 (maybe the $\sqrt{a}$ term inside the exponential should be without the sqrt?), or maybe my reasoning is not right (can you show me where does it fail?).

---

> > ### Author Response · Authors · 2025-02-09
> >
> > We thank the reviewer for the clarification and address their question below.
> >
> > > it seems we need to take $a \approx \frac{\log(T)^2 k^2}{T^2}$. In general, this leads to an expression of the form $O\left(\frac{T}{\log(T) k} \times \frac{1}{T^k}\right)$, so choosing $k=2$ leads to a $O\left(\frac{1}{T}\right)$ in terms of norm. Can you comment on this?
> >
> > If $a$ is chosen as the reviewer suggested, we can indeed obtain a convergence rate of $O\left(\frac{1}{T}\right)$ for $E\|w_T - w^*\|$. However, the reviewer's suggestion does not account for the fact that the batch-size threshold $b^* $ also depends on $a$ as $O \left(\frac{n}{1 + n a}\right)$. Hence, if $a \approx \frac{\log(T)^2 k^2}{T^2}$, $b^*$ will depend on $T$ as $O\left(\frac{n}{1 + \frac{n}{T^2}} \right)$. As $T \to \infty$, $b^* \to n$ (and $\zeta \to 0$) which means that we are asymptotically using the entire dataset and essentially analyzing SHB with a full-batch size. In this case, the $O\left(\frac{1}{T} \right)$ convergence rate (obtained by setting $a$ according to the reviewer's suggestion) is worse than the accelerated linear convergence rate $O\left( \exp\left(-\frac{T}{\sqrt{\kappa}} \right)\right)$ of full-batch SHB in Theorem 2 (obtained by simply setting $a = 1$ and $b = n$). To avoid such confusion in the paper, we will clarify that $a$ is a constant independent of $T$.

---

> > > ### Comment · Reviewer_iV4k · 2025-02-10
> > >
> > > Thanks for the clarification.
> > > You have addressed all my concerns.

---

### Review · Reviewer_B4mw · 2025-01-07

**Summary Of Contributions:**

The authors address the minimization of a $\mu$ strongly convex function. They also suppose that the functional is $L$ smooth and they study the convergence results of Stochastic Heavy Ball-type algorithms.

**Audience:**

Yes

**Broader Impact Concerns:**

Nothing to report.

**Claims And Evidence:**

No

**Requested Changes:**

The paper addresses an interesting and important topic, but significant corrections and clarifications are required. The authors need to:

- Correct the proofs of Theorems 6 and 7.

- Reevaluate and clarify their claims of improved results.

- Ensure rigor and consistency throughout the manuscript, particularly in the handling of constants and notation.

**Strengths And Weaknesses:**

While the work shows potential, three major issues need to be addressed before considering the possibility of publication:

- The proofs of Theorems 6 and 7 are currently incorrect due to the following reasons:

The step size $\eta_k$ depends on $\hat{L}$, making it random. Contrary to the claim made at the top of page 20, the argument used in the proof of Theorem 5 cannot be extended here because the step size cannot be factored out of the expectation. As a result, the subsequent derivations are invalid.

One potential way to address this issue is to assume that $\eta_k$ is $\mathcal{F}_k$-measurable. However, is this assumption realistic? Perhaps an online estimator of $L$ could be employed.

Even if this problem is resolved, the authors cannot perform a case disjunction as done in the current proofs since $\hat{L}$ remains random. Indicator functions must be used, complicating the proof substantially.


- The second significant concern is whether the improvements claimed by the authors are valid. For instance, in Theorem 1:
  The authors claim a convergence rate of
    \[
    O\left(\exp(-T/\kappa) + \frac{\kappa^2}{T}\right),
    \]
    where $\kappa = L/\mu$ is the condition number. They assert that this result is better than Sebbouh et al. (2020, Appendix H), which provides a convergence rate of
    \[
    O\left(\frac{\kappa^2}{T^2} + \frac{\kappa^2}{T}\right).
    \]
    While it is true that the residual term is improved, the primary term $O(1/T)$ appears to have a larger constant in the authors' results. This discrepancy must be clarified, as it undermines the claim of superiority.

- The manuscript suffers from several instances of imprecision and lack of rigor. Examples include:

In Theorem 1:
    What is $\eta$ in the definition of $\lambda_k$?
          The dependencies on constants, especially the condition number $\kappa$, are not fully exposed. For example, $c_2$ and $c_L$ implicitly depend on $\kappa$ but this is not explicitly stated.


In Theorem 2:
    Does $\chi$ depend on $i$? If not, is it the supremum?
        The definition of $\Delta_k = \| w_k - w \|$ is inconsistent with subsequent sections (e.g., Theorem 3).



In Theorem 4:
     The dependence on $\kappa$ is again hidden in constants such as $C_1$ and $C_3$. This needs to be explicitly addressed.

 In the proof of Theorem 2:
    The condition $\rho + \sqrt{\zeta} \sqrt{a} < 1$ is introduced without being stated in the assumptions.
      Additionally, this inequality is only feasible if $\zeta < \frac{1}{4\kappa}$. Is this assumption realistic?

---

> ### Author Response · Authors · 2025-01-21
>
> We thank the reviewer for their helpful feedback. We have updated the paper (in red) with the following changes requested by the reviewer:
> - We have updated Theorem 1 to use the full expression of $||w_k - w^*||^2$ instead of $\Delta_k$.
> - For Theorem 1 and Theorem 4, we represent unimportant constants as functions of $\kappa$ and $T$ in the main paper, while providing detailed derivations in the Appendix to help readers focus on the key results.
> - In Theorem 1, we have clarified that $\lambda_k$ depends on $\eta_0$ (the initial value of the step-size).
>
> We also address their remaining questions and concerns below.
> > The proofs of Theorems 6 and 7 are currently incorrect since the step size $\eta_k$ depends on $\hat{L}$, making it random.
>
> The reviewer's understanding is incorrect. In our proof, $\hat{L}$ is a deterministic estimate of $L$ (there is no randomness) such that $\hat{L}$ is a multiplicative factor $\nu_L$ away from the true $L$. Since there is no randomness, the step-size can be factored out of the expectation and the resulting proof is correct. We have clarified this in Appendix C.1 in the paper.
>
> > whether the improvements claimed by the authors are valid .. Theorem 1 .. is better than Sebbouh et al. (2020) ..
>
> When comparing convergence rates, the improvement we claim is on the bias term. The $O\left(\frac{\sigma^2}{T}\right)$ variance term is already optimal in terms of $T$. The bias term in Theorem 1 decreases exponentially at a rate of $O\left(\exp\left( \frac{-T}{\kappa} \right) \right)$, which is a significant improvement over the bias term in Sebbouh et al. (2020), where it decreases polynomially at an $O \left( \frac{\kappa^2}{T^2} \right)$ rate. Consequently, if $\sigma = 0$ (for example, in the interpolation regime), the $O(1/T^2)$ rate in Sebbouh et al is sub-optimal, whereas Theorem 1 recovers the standard $O(\exp(-T/\kappa)$ rate.
>
> > Does $\chi$ depend on $i$?
>
> By our definition, $\chi := \mathbb{E}_i || \nabla f_i (w^*)||$ does not depend on $i$ as it represents the expectation (over $i$) of the norm of the gradient at the optimum $w^*$.
>
> > Theorem 2: The condition $\rho + \sqrt{\zeta}\sqrt{a} < 1$ is introduced without being stated in the assumptions.
>
> We only use this condition in our induction proof outline and later demonstrate that it is true for the batch-size threshold specified in the theorem.

---

> > ### Comment · Reviewer_B4mw · 2025-02-10
> >
> > Thank you for these modifications. Regarding the determinism of the estimators, I agree that in this case, there is no longer an issue. However, I am curious to know how you manage to obtain such deterministic estimators.
> >
> > As for the gain provided by Theorem 1, I remain unconvinced. Indeed, while you reduce the bias at an exponential rate, this comes at the cost of increased variance. You do achieve an optimal rate of 1/T, but at the expense of a significantly larger constant. That being said, your result remains interesting.

---

> > > ### Author Response · Authors · 2025-02-12
> > >
> > > We thank the reviewer for engaging in a discussion. We address their remaining questions below.
> > >
> > > > Regarding the determinism of the estimators, I agree that in this case, there is no longer an issue. However, I am curious to know how you manage to obtain such deterministic estimators.
> > >
> > > One common approach to estimate $L$ is to use two points $w, w'$ and their gradient evaluations, and define $\hat{L} = \frac{|| \nabla f(w) - \nabla f(w') ||}{|| w - w' ||}$. Since smoothness guarantees that $|| \nabla f(w) - \nabla f(w') || \leq L || w - w' ||$, this is a sensible way to estimate $L$. In the context of this paper, we could form such a deterministic estimator of the smoothness before running the SHB algorithm. In this case, $\frac{1}{\hat{L}} > \frac{1}{L}$ and the discrepancy between $\hat{L}$ and $L$ is captured by $\nu_L$. Our analysis quantifies the dependence of the convergence rate on this misspecification. Alternatively, if $f^*$ is known, we could use a one-point deterministic estimator of the smoothness as $\hat{L} = \frac{f(w) - f^*}{|| \nabla f(w) ||^2}$. For the strong-convexity parameter, using $\hat{\mu} = \lambda$ (where $\lambda$ is the $\ell_2$ regularization parameter) is a common way to estimate $\mu$ for $\ell_2$-regularized convex losses. Since $\hat{\mu} \leq \mu$, the corresponding discrepancy is captured by $\nu_\mu \leq 1$ and we  again analyze the impact of this misestimation on the resulting convergence rate.
> > >
> > > > While you reduce the bias at an exponential rate, this comes at the cost of increased variance. You do achieve an optimal rate of $\frac{1}{T}$, but at the expense of a significantly larger constant.
> > >
> > > We will add a more nuanced comparison to the paper as follows: ''Compared to Sebbouh et al. (2020), the rate in Theorem 1 has the same optimal $\tilde{O}(1/T)$ dependence on the variance term, but results in a worse dependence on the constants. On the other hand, our algorithm results in a better dependence on the bias term: $O(\exp(-T))$ in Theorem 1 versus the $O(1/T^2)$ rate obtained by Sebbouh et al. (2020). Consequently, when using the full dataset (i.e. $b = n$) or under the interpolation setting (i.e. $\sigma = 0$), the rate in Theorem 1 recovers that of deterministic HB (Ghadimi et al 2015), while that in Sebbouh et al. (2020) does not.''

---

### Decision · Action_Editor_wga6 · 2025-02-21

**Recommendation:** Accept as is

**Comment:**

The paper presents new convergence analysis for stochastic momentum in the strongly convex and smooth setting, that scale gracefully as the batch size increases, or as the problems get closer to interpolation. Gracefully, because the rates converge to the faster rates established for the full batch momentum method as we approach interpolation. The authors also provide a lower bound for quadratics that explains why one their results forces the batch size to depend on the condition number. Because of these results, I support accepting this paper. This is also inline with reviewers, who also supported acceptance.

**Audience:**

Given the widespread use of momentum method, if not by itself than as a component of Adam, I believe there will be a significant subset of TMLR's audience that will find these new faster rates interesting.

**Claims And Evidence:**

The paper presents an analysis of the stochastic heavy-ball (or momentum) method in the smooth and strongly convex setting. This analysis is supported by well written proofs, that are readily verifiable.